# Cell shape sensing licenses dendritic cells for homeostatic migration to lymph nodes

Zahraa Alraies [1], Claudia A. Rivera[1], Maria-Graciela Delgado[1,19], Doriane Sanséau[1,19], Mathieu Maurin[1], Roberto Amadio[2], Giulia Maria Piperno[2], Garett Dunsmore[3], Aline Yatim[1], Livia Lacerda Mariano[1], Anna Kniazeva[1], Vincent Calmettes[1], Pablo J. Sáez [4], Alice Williart[5], Henri Popard[5], Matthieu Gratia[1], Olivier Lamiable [6], Aurélie Moreau[7], Zoé Fusilier[1,8], Lou Crestey[1], Benoit Albaud[9], Patricia Legoix[1], Anne S. Dejean[10], Anne-Louise Le Dorze[10], Hideki Nakano [11], Donald N. Cook[11], Toby Lawrence [12,13,14], Nicolas Manel [1], Federica Benvenuti[2], Florent Ginhoux[3,15,16,17], Hélène D. Moreau[1], Guilherme P. F. Nader[5,18,20], Matthieu Piel [5,20] ✉ & Ana-Maria Lennon-Duménil [1,20] ✉

Immune cells experience large cell shape changes during environmental patrolling because of the physical constraints that they encounter while migrating through tissues. These cells can adapt to such deformation events using dedicated shape-sensing pathways. However, how shape sensing affects immune cell function is mostly unknown. Here, we identify a shape-sensing mechanism that increases the expression of the chemokine receptor CCR7 and guides dendritic cell migration from peripheral tissues to lymph nodes at steady state. This mechanism relies on the lipid metabolism enzyme cPLA$_2$, requires nuclear envelope tensioning and is finely tuned by the ARP2/3 actin nucleation complex. We also show that this shape-sensing axis reprograms dendritic cell transcription by activating an IKKβ−NF-κB-dependent pathway known to control their tolerogenic potential. These results indicate that cell shape changes experienced by immune cells can define their migratory behavior and immunoregulatory properties and reveal a contribution of the physical properties of tissues to adaptive immunity.

The success of immune responses largely relies on the motility of immune cells that constantly circulate between peripheral tissues and/or lymphoid organs. To migrate, these cells must exert forces on the environment, which in most cases is achieved through the dynamic reorganization of their actomyosin cytoskeleton in response to external signals[1]. In addition, to reach a particular destination, cells sense biochemical and/or physical cues that guide them through the complex environment of tissues and vessels[2]. Among them, chemokines, which can form gradients recognized by specific cell-surface receptors, are known to play a prominent role[3]. It is well known that various types of

biochemical signals can switch on the expression of these receptors. However, whether chemokine receptor expression responds to the physical stimuli encountered by migrating immune cells is unclear. A chemokine receptor that has attracted a lot of attention is CCR7, which recognizes gradients of CCL19 and CCL21 chemokines[4,5]. CCR7 is required for the migration of distinct types of immune cells to lymph nodes, including dendritic cells (DCs). Once in lymph nodes, DCs present antigens collected in their tissue of residency to T cells[6,7]. Antigen presentation has two distinct outcomes[8]. It leads to T cell activation when DCs present antigens from a tissue that is inflamed because of

infection or tumor growth. Alternatively, it can inactivate self-reactive T cells, a process referred to as 'peripheral immune tolerance', which limits autoimmune reactions at steady state[7,9,10]. Interestingly, a population of CCR7[+] DCs was recently described as residing within tumors[11,12]; it remains unclear whether these cells trigger tolerogenic or antitumoral responses. Regulation of CCR7 expression might thus be critical to define the balance between tolerance and immunity at steady state or during infection and cancer. The main inducers of CCR7 expression identified so far are microbial components and endogenous inflammatory mediators[4,13,14]. Interestingly, seminal studies from the Mellman group have shown that mechanical disruption of cell–cell junctions can also induce the expression of CCR7 in DCs[15]. However, whether this phenomenon contributes to DC migration to lymph nodes remains an open question, as it is unclear whether DCs form such junctions within peripheral tissues. Nonetheless, these results suggest that the physical signals to which DCs are exposed while patrolling peripheral tissues might modify the capacity of DCs to express CCR7 and migrate to lymph nodes in the absence of tissue inflammation[15–17].

In tissues, a major physical signal experienced by motile cells results from the shape changes and deformation of internal organelles imposed by the physical constraints that they experience. We and others have shown that immune cells can adapt and respond to large shape changes when spontaneously migrating through peripheral tissues or dense tumors[18–20]. Such shape changes lead to nuclear deformation events that can activate the lipid metabolism enzyme cytosolic phospholipase 2 (cPLA$_2$), a sensor of nuclear envelope stretching[21–23]. Once activated, cPLA$_2$ uses phospholipids from the nuclear membrane to produce arachidonic acid (AA) that can be converted into leukotriens and prostaglandins[24,25]. AA production by cPLA$_2$ also enhances actomyosin contractility, allowing cells that are physically trapped within a dense tissue to release themselves and keep moving forward[21,23]. Whether cPLA$_2$ activation has any impact on the transcriptional activity of immune cells is unknown.

Here, we subjected DCs to distinct deformation events of amplitudes that fall within the range of shape changes that they experience in vivo. We identified a precise deformation amplitude at which DCs turn on an ARP2/3–cPLA$_2$–NF-κB-dependent shape-sensing mechanism. Activation of this pathway leads to CCR7 upregulation, controls steady-state DC migration to lymph nodes and further reprograms transcription in a different way than microbial sensing. Hence, DCs use the cytoskeleton–lipid metabolism interplay to respond to environmental physical constraints that imprint them with specific immunoregulatory properties.

## Results

### DC motility and CCR7 expression are co-regulated after shape changes

To migrate to lymph nodes, DCs must express the CCR7 chemokine receptor and enhance their intrinsic motility. We have previously observed that confinement-induced stretching of the nuclear envelope increases DC motility by promoting actomyosin contractility in a cPLA$_2$-dependent manner[21]. We hypothesized that such shape change might further trigger CCR7 expression, thereby endowing DCs with the full capacity to reach their next destination. To test this hypothesis, we manipulated the shape of DCs in a controlled manner using a cell-confining device[26]. To define the confinement heights to be used, we imaged DCs patrolling the dermis of mouse ear explants by two-photon microscopy. Timelapse movies showed that DCs expressing yellow fluorescent protein-tagged CD11c (CD11c–YFP) exhibited a minimal cell diameter of, on average, ~2–4 µm and a maximal cell diameter of ~6–10 µm while sampling the skin (Fig. 1a and Supplementary Video 1). We further observed that DCs spent ~35% of their time displaying diameters between ~2 and 4 µm and spent ~50% of their time at diameters of >4 µm (Fig. 1a). These results were in good agreement with data previously obtained by measuring the minimal diameters of the nuclei of bone marrow-derived DCs transferred into epidermal ear sheets[19]. This prompted us choosing confinement heights of 2, 3 and 4 µm to monitor CCR7 expression in DCs experiencing cell shape changes (Fig. 1b). As a cell model, we used unstimulated bone marrow-derived DCs (immature DCs), which express low levels of CCR7 in culture[27,28], similar to DCs patrolling peripheral tissues[29]. Of note, tissue-resident DCs could not be used for our purpose as they spontaneously upregulate CCR7 expression after tissue disruption. Bone marrow was obtained from a mouse model where a reporter gene encoding green fluorescent protein (GFP)-tagged CCR7 (CCR7–GFP) was knocked in to evaluate the dynamics of CCR7 expression in deformed DCs[30]. Strikingly, fluorescence quantification showed that although GFP expression increased in DCs confined at a height of 3 µm for 4–6 h (Extended Data Fig. 1a), it was not substantially modified in cells confined at heights of 2 or 4 µm (Fig. 1c and Supplementary Video 2). GFP upregulation did not result from cell death, which was low at all confinement heights (<8% of dead cells; Extended Data Fig. 1b). A similar sensitivity window was observed when monitoring the migration speed of confined DCs, which increased at a confinement height of 3 µm but not at heights of 2 or 4 µm (Fig. 1d). These experiments indicate that both CCR7 expression and cell motility respond to precise cell shape changes, suggesting that these events might be mechanistically coupled.

We confirmed these results when analyzing endogenous CCR7 expression by real-time quantitative PCR with reverse transcription (RT–qPCR) and immunofluorescence. Only immature DCs confined at a height of 3 µm exhibited a significant increase in CCR7 expression at both mRNA and protein levels (Fig. 1e,f). The specificity of the antibody to CCR7 was verified using *Ccr7*-knockout DCs (Extended Data Fig. 1c). *Ccr7* mRNA expression was also induced when cells were confined for 30 min, collected and analyzed 4 h later (Fig. 1g), suggesting that a 30-min deformation at a height of 3 µm was sufficient to upregulate *Ccr7* expression in DCs. Of note, CCR7 expressed at the surface of

**Fig. 1 | CCR7 upregulation in immature DCs is shape sensitive. a**, Left, migrating CD11c[+] DCs in an ear explant. Arrows highlight cell deformation events. Center, minimum to maximum range of median cell diameter (*n* = 34 cells). Right, average time spent by each cell at their minimum diameter (mean ± s.d.; *n* = 43 cells). **b**, Top, schematic of cells under confinement (created with BioRender. com). Bottom, representative images of live DCs expressing LifeAct–GFP (gray) and DNA (red); scale bar, 10 µm. **c**, Left, representative images of CCR7–GFP-expressing DCs; scale bar, 10 µm. Right, violin plot (median and quartiles) of total GFP intensity. Outliers were calculated using a ROUT test (*Q* = 1%) represented in gray (*N* = 4 independent experiments; left, *n* = 348 cells in 4 µm, *n* = 314 cells in 3 µm, *n* = 211 cells in 2 µm; middle, *n* = 201 cells in 4 µm, *n* = 206 cells in 3 µm, *n* = 176 cells in 2 µm; right, *n* = 215 cells in 4 µm, *n* = 187 cells in 3 µm, *n* = 180 cells in 2 µm). Data were analyzed by Kruskal–Wallis test; ****P < 0.0001; NS, not significant (*P* > 0.999). **d**, Box plot with minimum to maximum range of median speed (*N* = 4 independent experiments; *n* = 96 cells in 4 µm, *n* = 89 cells in 3 µm, *n* = 85 cells in 2 µm). Data were analyzed by Kruskal–Wallis test;

****P < 0.0001. **e**, Left, representative images of DCs showing CCR7 expression (gray) and nuclei (red). A maximum *z* projection is shown; scale bar, 10 µm. Right, box plot with minimum to maximum range (*N* = 2 independent experiments; *n* = 49 nonconfined cells, *n* = 74 cells in 4 µm, *n* = 89 cells in 3 µm, *n* = 35 cells in 2 µm). Data were analyzed by Kruskal–Wallis test; ****P < 0.0001; NS, *P* > 0.999; NC, nonconfined. **f**, Expression of *Ccr7*. Data are shown as mean ± s.d. (*N* = 4 independent experiments). Data were analyzed by Kruskal–Wallis test; ****P < 0.0001. **g**, Expression of *Ccr7* after 30 min of confinement. Data are shown as mean ± s.d. (*N* = 3 independent experiments). Data were analyzed by Kruskal–Wallis test; **P = 0.0021; ****P < 0.0001. **h**, Left, cell trajectories; scale bar, 100 µm. Middle, representation of cell step. Data are representative of two independent experiments (*n* = 33 cells in 4 µm; *n* = 43 cells in 3 µm). **i**, Box plot with minimum to maximum range of mean speed. Data were analyzed by one-way Mann–Whitney test; ****P < 0,0001. Data are representative of two independent experiments (*n* = 33 cells in 4 µm; *n* = 43 cells in 3 µm).

DCs confined at a height of 3 μm was fully functional; only these cells exhibited chemotaxis when CCL19 was added at one side of the confinement device (Fig. 1h,i). Together, these results show that confinement of immature DCs at a height of 3 μm, but not at 2 or 4 μm, upregulates CCR7 expression and endows these cells with the ability to perform chemotaxis and enhances their intrinsic motility. DC confinement at a height of 4 μm was thus chosen as the negative-control condition for all following experiments.

**CCR7 upregulation after shape sensing relies on cPLA₂ activity**

We found that both CCR7 expression and DC motility were induced at the same confinement height (3 μm), suggesting that these processes might be controlled by common mechanisms. We thus investigated whether CCR7 upregulation involves cPLA₂, as we had previously shown that this lipid metabolism enzyme enhances cell motility in response to nuclear deformation[21,23]. Inhibiting the enzymatic activity of cPLA₂ by treatment with the drug AACOCF3 prevented the upregulation of

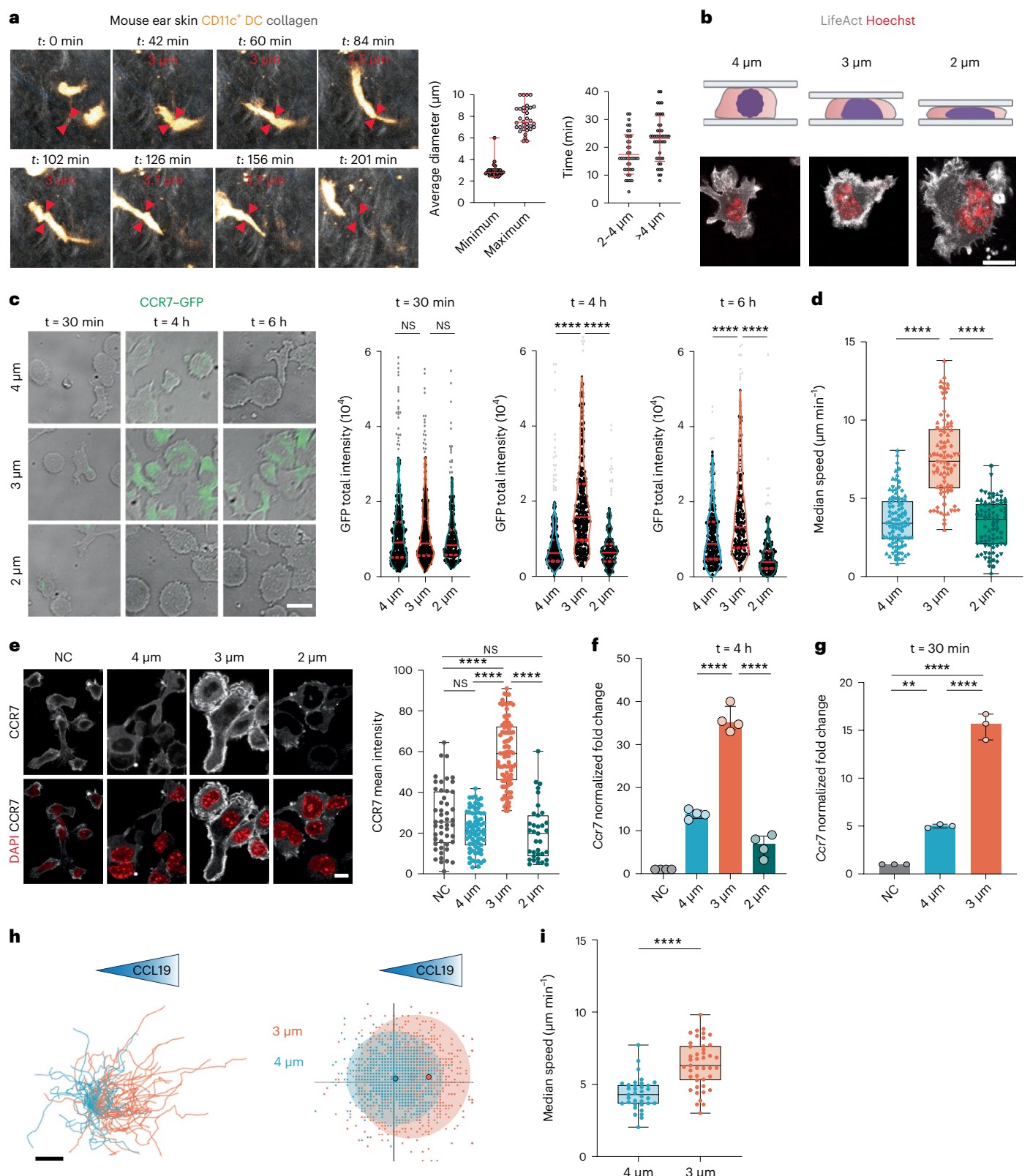

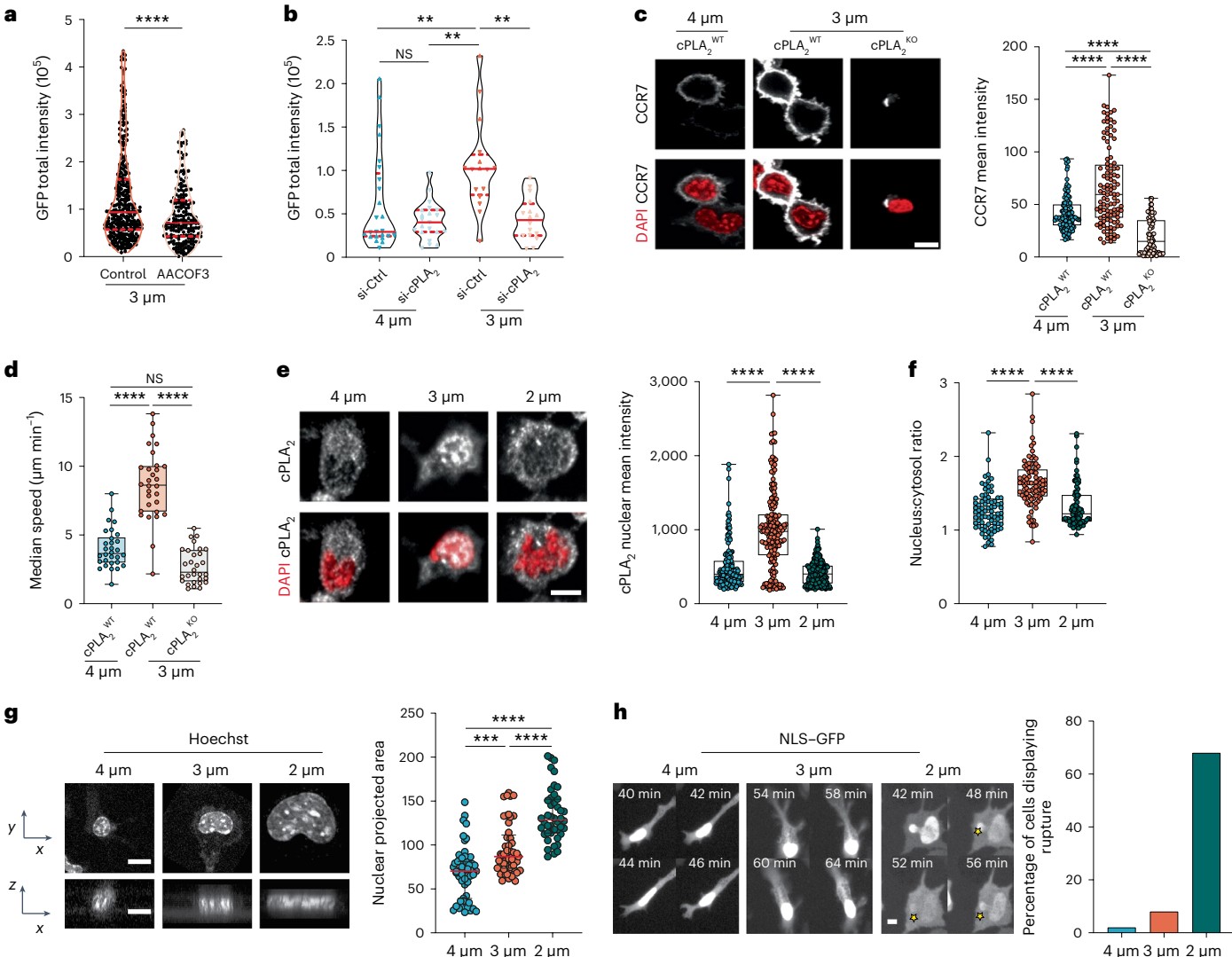

**Fig. 2 | CCR7 upregulation in response to shape sensing depends on cPLA₂ and an intact nuclear envelope. a**, GFP intensity in DCs treated with the cPLA₂ inhibitor AACOF3 (25 µM) or control. The violin plot shows median values and quartiles ($N = 3$, $n = 348$ cells in the 3-µm control, $n = 225$ cells treated with AACOF3). Data were analyzed by one-way Mann–Whitney test; ****$P < 0.0001$. **b**, Expression of CCR7–GFP in *Pla2g4a*-knockdown (si-cPLA₂) and control (si-Ctrl) DCs. The violin plot shows median values and quartiles ($N = 2$, $n = 25$ cells in si-Ctrl at 4 µm (80% of cells), $n = 21$ cells in si-cPLA₂ at 4 µm (75% of cells), $n = 16$ cells in si-Ctrl at 3 µm (75% of cells), $n = 16$ cells in si-cPLA₂ at 3 µm (76% of cells)). Data were analyzed by one-way analysis of variance (ANOVA) with a Kruskal–Wallis multiple comparison analysis; **$P = 0.0059$; **$P = 0.0039$. **c**, Left, representative images of cPLA₂^WT and cPLA₂^KO DCs. CCR7 is in gray, and nuclei are in red; scale bar, 10 µm. Right, box plot with minimum to maximum range of CCR7 intensity ($N = 3$, $n = 114$ cells in cPLA₂^WT at 4 µm, $n = 105$ cells in cPLA₂^WT at 3 µm, $n = 86$ cells in cPLA₂^KO at 3 µm). Data were analyzed by Kruskal–Wallis test; ****$P < 0.0001$. **d**, Box plot showing the minimum to maximum range of median speed of cPLA₂^WT and

cPLA₂^KO DCs ($N = 2$, $n = 30$ cells in all positions). Data were analyzed by Kruskal–Wallis test; ****$P < 0.0001$; NS, $P > 0.999$. **e**, Left, representative images of confined DCs. cPLA₂ is in gray, and nuclei are in red; scale bar, 10 µm. Right, box plot showing the minimum to maximum range of cPLA₂ intensity ($N = 3$, $n = 165$ cells in 4 µm, $n = 178$ cells in 3 µm, $n = 230$ cells in 2 µm). Data were analyzed by Kruskal–Wallis test; ****$P < 0.0001$. **f**, Nucleus-to-cytosolic ratio of cPLA₂ expression. The box plot shows the minimum to maximum range ($N = 3$, $n = 165$ cells in 4 µm, $n = 178$ cells in 3 µm, $n = 230$ cells in 2 µm). Data were analyzed by Kruskal–Wallis test; ****$P < 0.0001$. **g**, Left, projection of cell nuclei; scale bar, 10 µm. Right, box plot showing the median and interquartile range of the nucleus area ($N = 3$, $n = 55$ cells in 4 µm, $n = 49$ cells in 3 µm, $n = 44$ cells in 2 µm). Data were analyzed by Kruskal–Wallis test; ****$P < 0.0001$; ***$P < 0.0006$. **h**, Left, images of DCs transduced with NLS–GFP. Yellow stars indicate nuclear envelope rupture. Right, percentage of DCs displaying rupture events ($n$ = median of 3 independent experiments).

CCR7–GFP expression in DCs confined at a height of 3 µm (Fig. 2a). Similar results were observed in *Pla2g4a*-knockdown DCs, which further exhibited decreased cell motility (Fig. 2b and Extended Data Fig. 2a,b). To strengthen these results, we generated conditional knockout mice for the *Pla2g4a* gene (*Pla2g4a*^fl/fl crossed with *Itgax-cre* (Cd11c-cre) transgenic animals to obtain *Pla2g4a*-knockout (cPLA₂^KO) DCs). These cells did not upregulate CCR7 expression nor increase their motility after confinement at a height of 3 µm (Fig. 2c,d). By contrast, *Pla2g4a* knockdown or knockout did not prevent *Ccr7* upregulation in response

to treatment with the microbial compound lipopolysaccharide (LPS; Extended Data Fig. 2c,d). These results indicate that cPLA₂ is specifically required for CCR7 expression induced by cell shape changes rather than being generally involved in the transcriptional regulation of the *Ccr7* gene. We conclude that upregulation of both cell motility and CCR7 expression in DCs confined at a height of 3 µm relies on the activity of the cPLA₂ enzyme.

Consistently, we observed that cPLA₂ accumulated in the nucleus of DCs confined at a height of 3 µm, but not at 2 or 4 µm (Fig. 2e,f).

cPLA$_2$ is known to translocate to the nucleus[31] and associate with the inner nuclear membrane after activation[22]. Analysis of nuclear shape showed that DCs confined at heights of both 3 and 2 µm gradually increased the nucleus projected area (Fig. 2g), consistent with the nuclei being more deformed than the nuclei of cells confined at 4 µm. These results show that cPLA$_2$ does not accumulate in the nuclei of DCs confined at a height of 2 µm despite their nucleus being extensively stretched. This finding prompted us to hypothesize that confinement at a height of 2 µm might compromise nuclear accumulation of cPLA$_2$ due to loss of nuclear envelope integrity. To test this hypothesis, we transduced DCs with a lentiviral construct expressing nuclear localization signal (NLS)-tagged GFP (NLS–GFP). We observed that most NLS–GFP-expressing DCs confined at a height of 2 µm underwent nuclear envelope rupture followed by repair events, as evidenced by the transient leakage of NLS–GFP signal into the cytoplasm (Fig. 2h and Supplementary Video 3). Nuclear envelope rupture was less frequently observed in DCs confined at a height of 3 or 4 µm (Fig. 2h). Of note, DCs confined at a height of 2 µm did not display any additional sign of damage and were able to upregulate *Ccr7* expression after treatment with microbial LPS (Extended Data Fig. 2e). Together, these results show that coordinated upregulation of DC motility and CCR7 expression relies on nuclear accumulation of cPLA$_2$, which requires an intact nuclear envelope, and point to DCs being equipped with an extremely accurate machinery to detect precise levels of cell shape changes.

## ARP2/3 activity defines the cPLA$_2$ activation threshold after shape sensing

We next asked whether nuclear translocation and activation of cPLA$_2$ resulted from passive stretching of the DC nucleus after confinement or instead required an active cellular response. A good candidate for driving such a response was ARP2/3, as this complex has been shown to nucleate distinct types of actin structures in the perinuclear area of DCs undergoing nucleus deformation[18,32]. Live imaging analysis of LifeAct–GFP distribution in DCs as soon as they were confined at a height of 3 µm indeed revealed a cloud of perinuclear F-actin present in ~45% of cells compared to ~10% of cells under control conditions (Fig. 3a). Of note, this actin cloud was only observed when cells were fixed under confinement and was positioned on top or on either side of the nucleus (Fig. 3a and Extended Data Fig. 3a,b). Treatment of DCs with CK666, which inhibits ARP2/3 activity, led to the disappearance of this actin structure (Fig. 3a, Supplementary Video 4 and Extended Data Fig. 3a), suggesting that ARP2/3 activity is needed for its formation. This result was confirmed using a color-coded $z$ projection to analyze imaged cells. In DCs confined at a height of 3 µm, most actin was found in a cloud located on the top of nuclei (Fig. 3b).

We found that CK666 impaired the upregulation of *Ccr7* expression and the nuclear accumulation of cPLA$_2$ after confinement at a height of 3 µm (Fig. 3c,d). As observed for cPLA$_2$, ARP2/3 inhibition

had no effect on LPS-induced CCR7 upregulation (Extended Data Fig. 3c). Consistent with these results, we further found that DCs lacking Wiscott–Aldrich syndrome protein (WASp; encoded by the gene *Was*), which was recently shown to activate ARP2/3 in the DC perinuclear area[33], behaved similar to cells treated with CK666; they did not upregulate CCR7 expression nor show cPLA$_2$ nuclear translocation after confinement at a height of 3 µm (Fig. 3e,f). Of note, *Was*-knockout (WASp$^{KO}$) DCs exhibited a round morphology, consistent with previous reports by others[34] (Supplementary Video 5). Together, these results suggest that the response of DCs to shape changes requires WASp and ARP2/3 activity to allow nuclear accumulation of cPLA$_2$ and subsequent upregulation of CCR7 expression.

To strengthen these findings, we assessed the response of DCs lacking the ARP2/3 inhibitor Arpin (*Arpin*$^{fl/fl}$ × *Itgax-cre*), which exhibit enhanced ARP2/3 activity[35]. Remarkably, we found that *Arpin*-knockout (Arpin$^{KO}$) immature DCs displayed an increased sensitivity to cell shape changes as they upregulated *Ccr7* when confined at a height of 4 µm instead of 3 µm (Fig. 3g) in a cPLA$_2$-dependent manner (Fig. 3h). Accordingly, Arpin$^{KO}$ cells confined at 4 µm also exhibited cPLA$_2$ nuclear translocation (Fig. 3i). We conclude that the activity of ARP2/3 determines the sensitivity of DCs to confinement, defining a threshold for cPLA$_2$ nuclear accumulation and induction of CCR7 expression in response to cell shape changes. These results are consistent with DCs being equipped with a sensory mechanism that finely tunes their response to shape changes.

## ARP2/3 triggers cPLA$_2$ activation via nuclear envelope tensioning

To investigate the molecular basis of this mechanism, we analyzed whether ARP2/3 controls the folding and tension of the nuclear envelope, as we previously showed that these were critical for confinement-induced cPLA$_2$ activation in HeLa cells[21]. Staining of the nuclear envelope using the lamina-associated polypeptide 2 (LAP2) marker showed that it was less folded in DCs confined at 3 µm than in DCs confined at 4 µm (Fig. 4a). Noticeably, this difference was abrogated when treating DCs with CK666. Quantification of nuclear envelope folding by measuring the excess of the nuclear envelope perimeter (EOP$_{NE}$) showed that EOP$_{NE}$ was significantly lower in DCs confined at 3 µm and increased after inhibition of ARP2/3 (Fig. 4b). These results suggest that branched actin nucleation is required for confinement-induced unfolding of the nuclear envelope. This finding was further confirmed when analyzing the elliptic Fourier decomposition (EFC) ratio of the nuclear envelope, which compares the relative contribution of the first-order ellipse (perfect and smooth) to the contribution of subsequent-order ellipses required to fit shape irregularities. Indeed, CK666 treatment of DCs confined at a height of 3 µm decreased the EFC ratio, indicating that the nuclear envelope of ARP2/3-inhibited DCs exhibits a more irregular shape than that in

**Fig. 3 | ARP2/3 activity tunes the sensitivity of the cPLA2-dependent shape-sensing DC response. a**, Top, representative images of LifeAct–GFP DCs treated with CK666 (30 µM) or untreated. Middle, images of LifeAct–GFP (gray) and nuclei (red); scale bar 10 µm. Bottom, single resliced image; scale bar, 20 µm. Right, median with range of LifeAct nucleus-to-cytosol ratio ($N = 3$; $n = 55$ cells in 4 µm (DMSO), $n = 70$ cells in 3 µm (DMSO), $n = 42$ cells in 3 µm (CK666)). Data were analyzed by ordinary one-way ANOVA; ***$P = 0.0007$; **$P = 0.0016$. **b**, Color-coded $z$ frames of untreated LifeAct DCs and cells treated with 30 µM CK666. **c**, Expression of *Ccr7* (mean ± s.d.; $N = 3$). Data were analyzed by ordinary one-way ANOVA; ****$P < 0.0001$; *$P = 0.0465$; NS, $P > 0.999$. **d**, Left, representative images of DCs treated with 30 µM CK666 or untreated cells. cPLA$_2$ is in gray, and nuclei are in red; scale bar, 10 µm. Right, box plot showing the minimum to maximum range of cPLA$_2$ intensity ($N = 2$; $n = 121$ cells in DMSO (3 µm), $n = 64$ cells in CK666 (3 µm)). Data were analyzed by one-way Mann–Whitney test; ****$P < 0.0001$. **e**, Left, representative images of WASp$^{WT}$/WASp$^{KO}$ DCs. CCR7 is in gray, and nuclei are in red; scale bar, 10 µm. Right, box plot showing the minimum to

maximum range of CCR7 intensity ($N = 2$; $n = 70$ cells in WASp$^{WT}$ at 4 µm, $n = 70$ cells in WASp$^{WT}$ at 3 µm, $n = 38$ cells in WASp$^{KO}$ at 3 µm). Data were analyzed by Kruskal–Wallis test; ****$P < 0.0001$; ***$P < 0.0002$. **f**, Left, representative images of WASp$^{WT}$/WASp$^{KO}$ DCs. cPLA$_2$ is in gray, and the nuclei are in red; scale bar, 10 µm. Right, box plot showing the minimum to maximum range of cPLA$_2$ intensity ($N = 3$; $n = 150$ cells in WASp$^{WT}$ at 4 µm, $n = 145$ cells in WASp$^{WT}$ at 3 µm, $n = 149$ cells in WASp$^{KO}$ at 3 µm). Data were analyzed by Kruskal–Wallis test; ****$P < 0.0001$. **g**, Expression of *Ccr7* in Arpin$^{WT}$/Arpin$^{KO}$ DCs. Data are shown as mean ± s.d. ($N = 3$) and were analyzed by ordinary one-way ANOVA; ****$P < 0.0001$; ***$P = 0.0002$; **$P = 0.0015$; NS, $P > 0.999$. **h**, Expression of *Ccr7* in Arpin$^{WT}$/Arpin$^{KO}$ DCs ($N = 3$). Data were analyzed by ordinary one-way ANOVA; ****$P < 0.0001$; *$P = 0.0134$. **i**, Left, representative images of Arpin$^{WT}$/Arpin$^{KO}$ DCs. cPLA$_2$ is in gray, and nuclei are in red; scale bar, 10 µm. Right, box plot showing the minimum to maximum range of cPLA$_2$ intensity ($N = 2$; $n = 59$ cells in Arpin$^{WT}$ at 4 µm, $n = 87$ cells in Arpin$^{KO}$ at 4 µm, $n = 75$ cells in Arpin$^{WT}$ at 3 µm). Data were analyzed by ordinary one-way ANOVA; ****$P < 0.0001$; **$P = 0.0043$.

untreated confined DCs (Fig. 4c). Collectively, these data strongly suggest that ARP2/3-dependent actin nucleation is needed to unfold the nuclear envelope of DCs in response to cell shape changes.

Nuclear envelope tension can be measured using a membrane tension sensor targeted to the endoplasmic reticulum (ER) membrane (ER Flipper-TR) detected by fluorescence lifetime imaging (FLIM)[36,37]. Unfortunately, DCs failed to take up this sensor probe. We therefore turned to HeLa cells, which were initially used to show that increasing nuclear envelope tension led to cPLA$_2$ activation[22]. Remarkably, we observed that treating HeLa cells with CK666 diminished nuclear

envelope tension, as shown by a decreased lifetime of FLIM signal (Fig. 4d). To provide further evidence of nuclear envelope tension being needed for cPLA$_2$ activation in DCs, we used DCs lacking lamin A/C (*Lmna*-knockout (LmnA/C$^{KO}$) cells), which cannot build up nuclear envelope tension in response to deformation[21,38]. We found that LmnA/C$^{KO}$ DCs did not upregulate CCR7 expression nor translocate cPLA$_2$ into the nucleus when confined at a height of 3 µm (Fig. 4e,f). These results highlight that ARP2/3 facilitates unfolding and tensioning of the nuclear envelope in response to cell shape sensing in DCs, thereby tuning cPLA$_2$ activation.

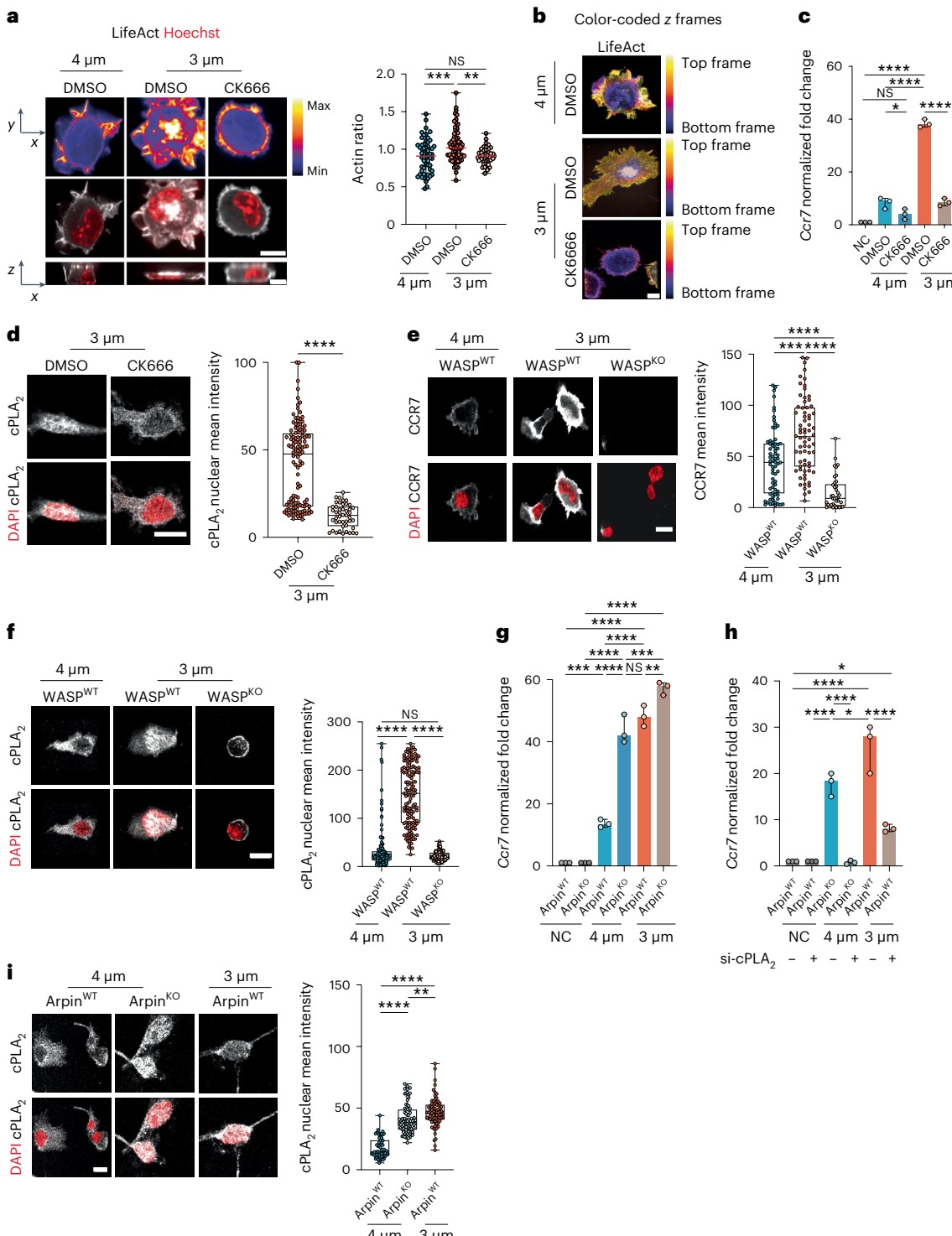

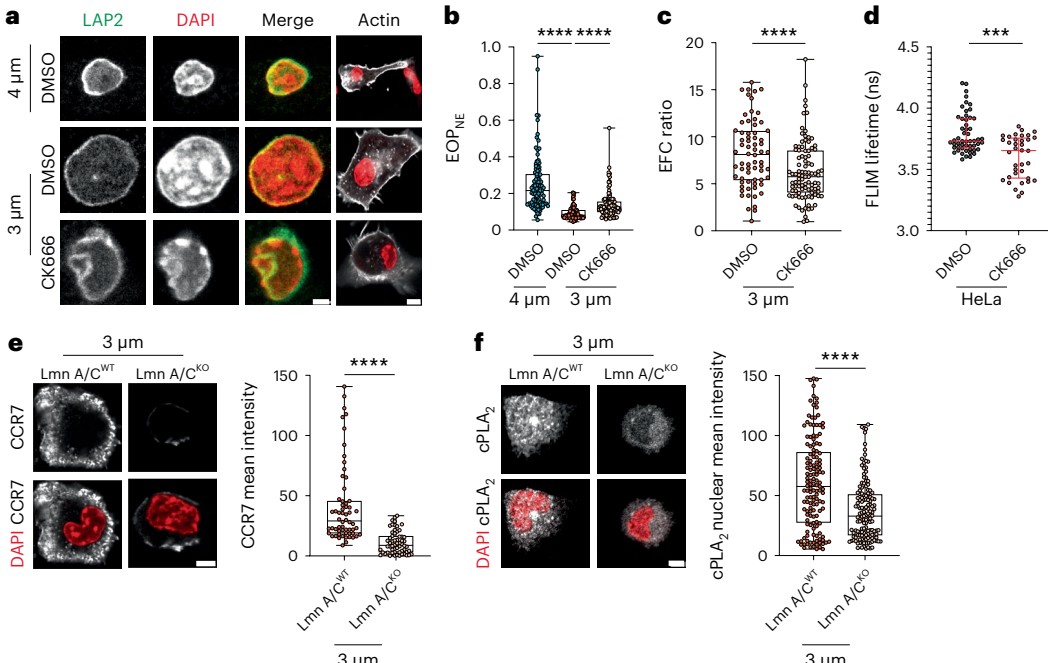

**Fig. 4 | ARP2/3 activity mediates DC shape sensing by unfolding the nuclear envelope and increasing its tension. a**, Representative images of DCs treated with 30 μM CK666 or untreated. LAP2 and DAPI are in gray on the left. Merged images of DAPI (red) and LAP2 (green) and of DAPI (red) and phalloidin (gray) are shown on the right; scale bars, 2 μm (left merge) and 10 μm (right merge). Images are representative of $N = 2$ samples. **b**, $EOP_{NE}$ of confined DCs treated with 30 μM CK666 or untreated DCs. Box plots show the minimum to maximum range ($N = 3$; $n = 100$ cells in 4 μm (DMSO), $n = 93$ cells in 3 μm (DMSO), $n = 97$ cells in 3 μm (CK666)). Data were analyzed by ordinary one-way ANOVA; ****$P < 0.0001$. **c**, EFC of confined DCs treated with 30 μM CK666 or untreated DCs. Box plots show the minimum to maximum range ($N = 3$; $n = 68$ cells in 3 μm (DMSO), $n = 97$ cells in 3 μm (CK666)). Data were analyzed by one-way Mann–Whitney test; ****$P < 0.0001$. **d**, Nuclear envelope tension sensor FLIM measurements in

HeLa cells after treatment with 30 μM CK666. Data are shown as median values with the interquartile range ($N = 3$; $n = 53$ cells (control), $n = 47$ cells (CK666)). Data were analyzed by one-way Mann–Whitney test; ***$P < 0.0001$. **e**, Left, representative images of DCs from LmnA/C^WT and LmnA/C^KO DCs. CCR7 is in gray, and the nuclei are in red; scale bar, 5 μm. Right, box plots showing the minimum to maximum range of CCR7 intensity ($N = 3$; $n = 54$ cells (LmnA/C^WT), $n = 60$ cells (LmnA/C^KO)). Data were analyzed by ordinary one-way ANOVA; ****$P < 0.0001$. **f**, Left, representative images of DCs from LmnA/C^WT and LmnA/C^KO DCs. cPLA_2 is in gray, and the nuclei are in red; scale bar, 5 μm. Right, box plot showing the minimum to maximum range of cPLA_2 intensity ($N = 3$; $n = 141$ cells (LmnA/C^WT), $n = 144$ cells (LmnA/C^KO)). Data were analyzed by unpaired $t$-test with a Welch's correction; ****$P < 0.0001$.

## Shape sensing tunes steady-state migration of DCs to lymph nodes

So far, we have shown that WASp–ARP2/3-dependent actin remodeling in response to shape changes defines the activation threshold of cPLA_2 and the capacity of DCs to upregulate the two processes required for migration to lymph nodes, specifically cell motility and CCR7 expression. These data strongly suggest that (1) WASp–ARP2/3-dependent cPLA_2 activation in response to cell shape changes might license DCs for migration to lymph nodes in the absence of inflammation, and (2) by restraining the activation of this shape-sensing pathway, Arpin might act as a negative regulator of this process in vivo. Such a regulatory mechanism could limit the number of DCs that acquire CCR7 expression and migrate to lymph nodes as a result of shape sensing during environment patrolling, thereby increasing their time of tissue residency.

To assess the physiological relevance of shape sensing by ARP2/3 and cPLA_2, we evaluated whether cPLA_2, WASp and Arpin deficiencies altered the number of migratory DCs present in skin-draining lymph nodes at steady state by flow cytometry. Conventional skin DCs can be divided into the following two main subtypes based on surface marker expression: type 1 conventional DCs (cDC1s) and type 2 conventional DCs (cDC2s)[39]. Although both populations can migrate from the skin to the lymph nodes, the migration rates of cDC2s have been found to be more elevated than the migration rates of cDC1s at steady state[39]. Strikingly, we observed that the numbers of migratory cDC2s found in inguinal lymph nodes were significantly decreased in WASp^KO and cPLA_2^KO mice (Fig. 5a–d). No such difference was detected for migratory cDC1s, which, as expected from prior findings, were less represented in

these secondary lymphoid organs under homeostatic conditions. Conversely, Arpin^KO mice displayed enhanced numbers of migratory cDC2s in lymph nodes compared to their wild-type counterparts (Fig. 5e). Of note, analysis of DC numbers in the skin of these animals showed no significant difference (Extended Data Fig. 4a–d), excluding the possibility that the differences observed in lymph node cDC2 numbers could result from altered cDC2 development and/or survival in the skin of WASp^KO, cPLA_2^KO or Arpin^KO mice. Hence, ARP2/3 activity controlled by WASp and Arpin finely tunes the number of DCs that migrate to lymph nodes at steady state, possibly by controlling the cPLA_2 activation threshold and downstream CCR7 expression in response to shape sensing. Of note, although WASp deficiency was not found to decrease DC motility in confinement ex vivo[34], we cannot exclude that in vivo, it could contribute to DC migration to lymph nodes by other means than activating cPLA_2 (refs. [34,40]). Together, these data suggest that steady-state migration of DCs to lymph nodes might result, at least in part, from the events of shape changes that they experience while patrolling the complex environment of the skin, suggesting a contribution of cell mechanics to this process.

## IKKβ and NF-κB activity control the ARP2/3–cPLA_2 shape-sensing axis

The cDC2s that migrate from the skin to the lymph nodes at steady state were shown to display a specific transcriptional profile enriched for NF-κB- and type I interferon (IFN)-related genes[39]. NF-κB activation downstream of the IKKβ (*Ikbkb*) kinase was further described as required for the migration of DCs from the skin to the lymph nodes

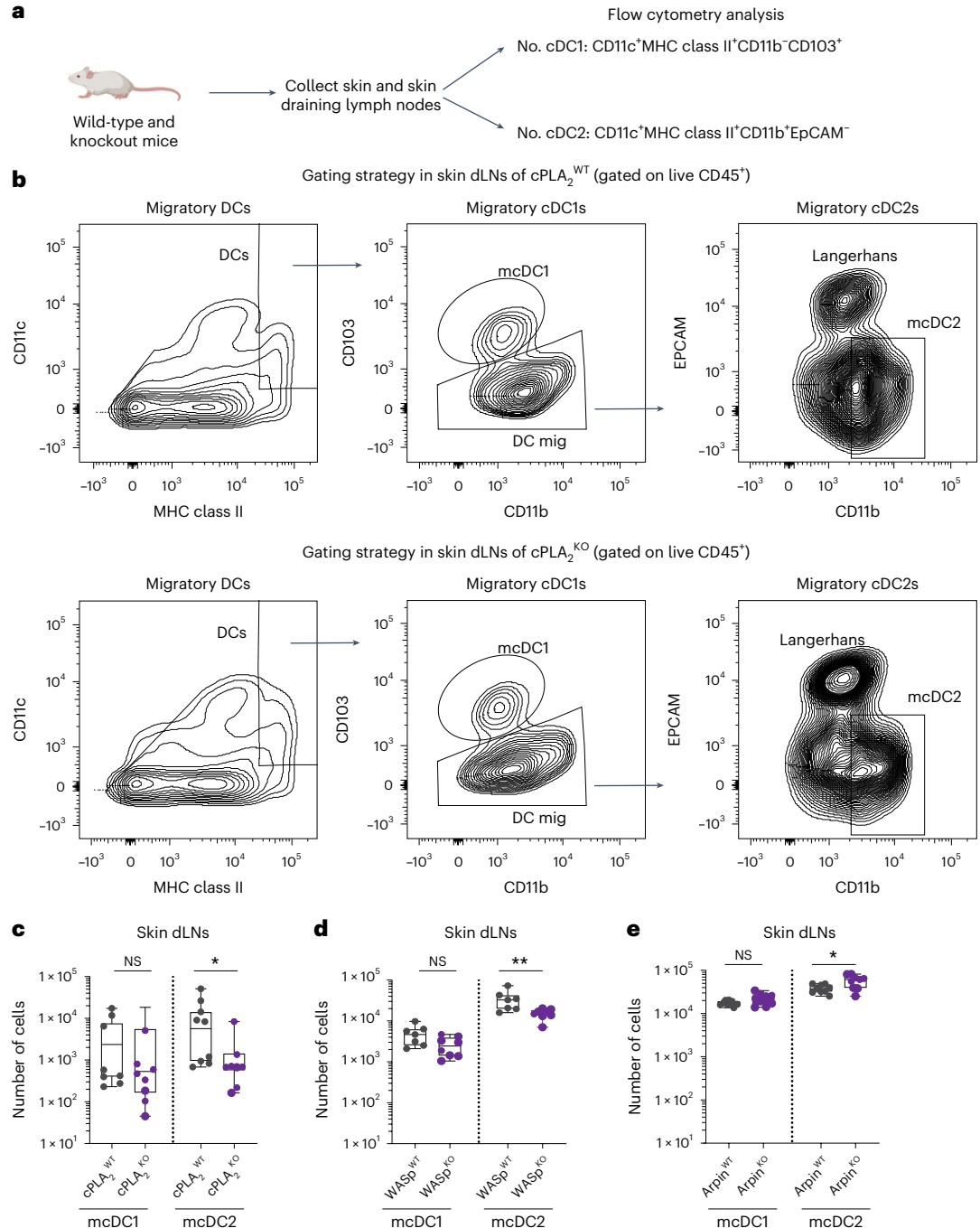

**Fig. 5 | Steady-state migration of skin migratory cDC2s is cell shape sensitive.** **a**, Schematic representation of the work flow. The picture of the mouse was created using BioRender.com. **b**, Gating strategy to quantify DCs in skin draining lymph nodes (dLNs). Top, representative example of the data in cPLA$_2$$^{WT}$ inguinal lymph nodes. Bottom, representative example of the data obtained in cPLA$_2$$^{KO}$ inguinal lymph nodes. Briefly, after gating on live cells, immune cells were identified as CD45$^{high}$; CD11c and MHC class II were then used to differentiate lymph node-resident DCs (MHC class II$^{low}$CD11c$^{high}$) from migratory (mig) DCs (MHC class II$^{high}$CD11c$^{high}$). Among migratory DCs, migratory cDC1s (mcDC1) were identified as CD11b$^{low}$CD103$^{high}$, and migratory cDC2s (mcDC2) were identified as CD11b$^{high}$EPCAM$^{low}$. **c**, Box plots showing the minimum to maximum range

in log scale of the number of migratory DCs in cPLA$_2$$^{WT}$ and cPLA$_2$$^{KO}$ mice ($N$ = 3 independent experiments, where each dot is one mouse). Data were analyzed by one-way Mann–Whitney $U$-test; *$P$ = 0.0221. **d**, Box plots showing the minimum to maximum range in log scale of the number of migratory DCs in WASp$^{WT}$ and WASp$^{KO}$ mice ($N$ = 3 independent experiments, where each dot is one mouse). Data were analyzed by ordinary one-way Mann–Whitney $U$-test; **$P$ = 0.0059. **e**, Box plots showing the minimum to maximum range in log scale of the number of migratory DCs in Arpin$^{WT}$ and Arpin$^{KO}$ ($N$ = 3 independent experiments, where each dot is one mouse). Data were analyzed by ordinary one-way Mann–Whitney $U$-test; *$P$ = 0.04.

at steady state and after inflammation[41]. Of note, this pathway is the only pathway described so far as being implicated in the homeostatic migration of skin DCs. We therefore investigated whether the ARP2/3–cPLA$_2$ shape-sensing axis relies on IKKβ-dependent NF-κB activation

and whether it triggers global transcriptional reprogramming of DCs. To this end, we compared the transcriptomes of cPLA$_2$$^{WT}$ and cPLA$_2$$^{KO}$ DCs confined at a height of 3 μm by bulk RNA sequencing (RNA-seq); nonconfined cells were used as negative controls.

Principal-component and clustering analyses revealed that although nonconfined nonstimulated cPLA$_2$$^{WT}$ and cPLA$_2$$^{KO}$ samples clustered together, this did not apply to confined cPLA$_2$$^{WT}$ and cPLA$_2$$^{KO}$ cells, showing that they display important differences in their gene expression profiles (Fig. 6a). These results indicate that cPLA$_2$ impacts the transcriptomes of confined DCs but has no major effect on nonconfined cells at steady state. More specifically, we observed that ~5,000 and ~4,600 genes were up- and downregulated, respectively, in cPLA$_2$$^{WT}$ DCs confined at a height of 3 μm compared to under nonconfined conditions (Fig. 6b). Comparison of cPLA$_2$$^{WT}$ and cPLA$_2$$^{KO}$ DCs showed that more than half of the genes upregulated by confinement relied on cPLA$_2$ (~3,600 genes; Fig. 6b). Consistent with our findings, *Ccr7* upregulation was found to be fully lost in confined cPLA$_2$$^{KO}$ DCs, which showed decreased *Ccr7* mRNA levels after confinement (Fig. 6c). Strikingly, among the 103 genes following the same expression pattern as *Ccr7* were 2 genes encoding the major subunits of ARP2/3 (*Actr2* and *Actr3*), the gene encoding IKKβ (*Ikbkb*) and several IFN-stimulated genes (Fig. 6d). Consistently, RNA-seq analysis of DCs confined at a height of 3 μm and treated with CK666 showed that the genes that were upregulated in expression after confinement in a cPLA$_2$-dependent manner, including the 103 genes behaving similar to *Ccr7*, were also dependent on ARP2/3 activity (Fig. 6e,f). Of note, CK666 treatment showed additional effects on gene expression similar to the cPLA$_2$$^{KO}$ condition (Fig. 6f), indicating that ARP2/3 regulates transcription in response to confinement by additional means than activating cPLA$_2$.

Importantly, direct comparison of the transcriptional profiles of our confined bone marrow-derived DCs with the transcriptional profile described for skin migratory cDC2s revealed that they exhibited similar signatures (genes following the *Ccr7* expression pattern; Fig. 6g). These results suggest that the ARP2/3–cPLA$_2$ shape-sensing axis imprints DCs with a similar transcriptional program as the program displayed by skin cDC2s migrating to lymph nodes at steady state, confirming the in vivo contribution of this mechanical pathway. These results prompted us to investigate whether cPLA$_2$ and IKKβ–NF-κB were part of the same signaling route. We found that (1) confinement at a height of 3 μm did not lead to *Ccr7* upregulation in DCs treated with the IKKβ inhibitor BI605906 or in DCs lacking the *Ikbkb* gene (Fig. 6h,i), and (2) NF-κB nuclear translocation was compromised in confined DCs lacking cPLA$_2$ (Fig. 6j). By contrast, IKKβ inhibition had no effect on nuclear accumulation of cPLA$_2$ in confined DCs (Fig. 6k). These data strongly suggest that cPLA$_2$ acts upstream of IKKβ and NF-κB to trigger upregulation of CCR7 expression after shape sensing in DCs. Together, our data support a model where shape sensing through the ARP2/3–cPLA$_2$ axis activates the IKKβ–NFκB pathway and thereby licenses DCs to migrate to lymph nodes at steady state.

## Shape sensing endows DCs with specific immunoregulatory properties

DC migration to lymph nodes at steady state helps maintain peripheral tolerance[7,9,42]. This is in sharp contrast to microbe-induced DC migration to lymph nodes that leads to activation of T cells capable of fighting these infectious agents. Our results show that ARP2/3 and cPLA$_2$ are required for CCR7 upregulation in response to confinement but not in response to LPS (Extended Data Figs. 2d and 3c). Similarly, CK666-treated DCs clustered together, suggesting minor differences in their transcriptional profiles whether or not they were activated with LPS (Extended Data Fig. 5a). This suggests that the shape-sensing pathway described here might be specifically involved in steady-state, rather than microbe-induced, migration of DCs to lymph nodes. This scenario would be particularly appealing as no specific mechanism has been identified so far for the triggering of homeostatic DC migration, the IKKβ–NF-κB pathway being required for DC migration to lymph nodes both at steady state and after inflammation[41]. To test this hypothesis, we compared the transcriptomes of DCs expressing or not expressing cPLA$_2$ either confined at a height of 3 μm or treated with LPS. Strikingly, principal-component and clustering analyses revealed that cPLA$_2$ had no substantial impact on the global gene expression pattern induced by microbial stimulation (Fig. 7a and Extended Data Fig. 5b), confirming the specific requirement of cPLA$_2$ for mechanical reprogramming of DCs. Interestingly, while some genes associated with the cPLA$_2$ pathway were upregulated under both conditions, others, including the genes encoding cPLA$_2$ and the prostaglandin E$_2$ (PGE$_2$) receptor, were more induced in confined DCs than in LPS-treated cells (Fig. 7b). This result prompted us to investigate whether production of PGE$_2$ from AA could contribute to DC transcriptional reprogramming by confinement. We found that the addition of PGE$_2$ to confined cPLA$_2$$^{KO}$ DCs led to upregulation of CCR7 expression (Fig. 7c), showing that PGE$_2$ can compensate cPLA$_2$ deficiency. This was not observed when inhibiting IKKβ (Fig. 7c). Accordingly, PGE$_2$ was found to induce NF-κB nuclear translocation (Fig. 7c). Together, these data suggest that DC transcriptional reprogramming after shape sensing involves prostaglandin production downstream of cPLA$_2$, which in turn activates IKKβ and NF-κB in a paracrine manner.

To better understand the specificity of DC reprogramming in response to shape sensing, we compared the pathways induced by DC confinement or by LPS (Fig. 7d). One of the most striking differences observed was related to the 'regulation of helper T cell differentiation' pathway that was exclusively induced by LPS (Fig. 7d,e). Mounting an efficient T cell response against a microbial threat typically requires the following three signals from DCs: specific antigenic peptide presented on major histocompatibility complex (MHC; signal 1), co-stimulation (signal 2) and cytokines (signal 3). The two latter signals ensure T cell activation rather than tolerization as well as proper proliferation.

**Fig. 6 | cPLA$_2$ reprograms DC transcription in an IKKβ–NF-κB-dependent manner. a–d**, Bulk RNA-seq analysis of cPLA$_2$$^{WT}$ and cPLA$_2$$^{KO}$ DCs; NC NS, nonconfined nonstimulated. **a**, Multidimensional scaling (MDS) of the samples. Sample groups under different confinement conditions are represented by different forms. Each dot represents a biological replicate; dim, dimension; FC, fold change. **b**, Pie charts showing proportions of differentially expressed genes in cells under confined conditions compared to nonconfined cells (false discovery rate < 0.05 and log$_2$ (fold change) of <−1.0 or >1.0). Top, number of upregulated genes in confinement in cPLA$_2$$^{WT}$ and cPLA$_2$$^{KO}$ DCs (5,695 genes). Bottom, number of downregulated genes in confinement in both cPLA$_2$$^{WT}$ and cPLA$_2$$^{KO}$ DCs (4,666 genes). **c**, *Ccr7* estimated gene counts in the different samples/conditions. **d**, Heat map of genes harboring a similar expression pattern as *Ccr7*. **e**, Heat map of the genes moving with *Ccr7* expression in response to treatment with CK666 (30 μM). **f**, Heat map of all differentially expressed genes in response to treatment with CK666 (30 μM) in confined and nonconfined DCs. **g**, Heat map showing normalized expression of some IFN-stimulated genes in the CD11b$^+$ cDC2 subset from the dermis and their migratory counterparts in the cutaneous draining lymph node of healthy mice (GSE49358 microarray study from Tamoutounour et al.[39]). **h**, *Ccr7* expression in response to treatment with BI605906 (30 μM). Data are shown as mean ± s.d. ($N = 2$, $n = 6$) and were analyzed by ordinary one-way ANOVA; ****$P < 0.0001$. **i**, *Ccr7* expression in cells from *Ikbkb*$^{WT}$ or *Ikbkb*$^{KO}$ mice. Data are shown as mean ± s.d. ($N = 2$, $n = 6$) and were analyzed by ordinary one-way ANOVA; ****$P < 0.0001$. **j**, Top, representative images of cPLA$_2$$^{WT}$ and cPLA$_2$$^{KO}$ DCs. NF-κB (P65) is in gray, and the nuclei are in red; scale bar, 10 μm. Bottom, box plot showing the minimum to maximum range of NF-κB (P65) intensity ($N = 2$, $n = 55$ cells in 4 μm (cPLA$_2$$^{WT}$), $n = 63$ cells in 3 μm (cPLA$_2$$^{WT}$), $n = 34$ cells in 4 μm (cPLA$_2$$^{KO}$)). Data were analyzed by Kruskal–Wallis test; ****$P < 0.0001$. **k**, Left, representative images of DCs treated with the IKKβ inhibitor BI605906 (30 μM) and nontreated control DCs. cPLA$_2$ is in gray, and nuclei are in red; scale bar, 12 μm. Right, box plot showing the minimum to maximum range of cPLA$_2$ intensity ($N = 2$, $n = 29$ cells (DMSO), $n = 23$ cells (BI605906)). Data were analyzed by ordinary one-way ANOVA; *$P = 0.0175$.

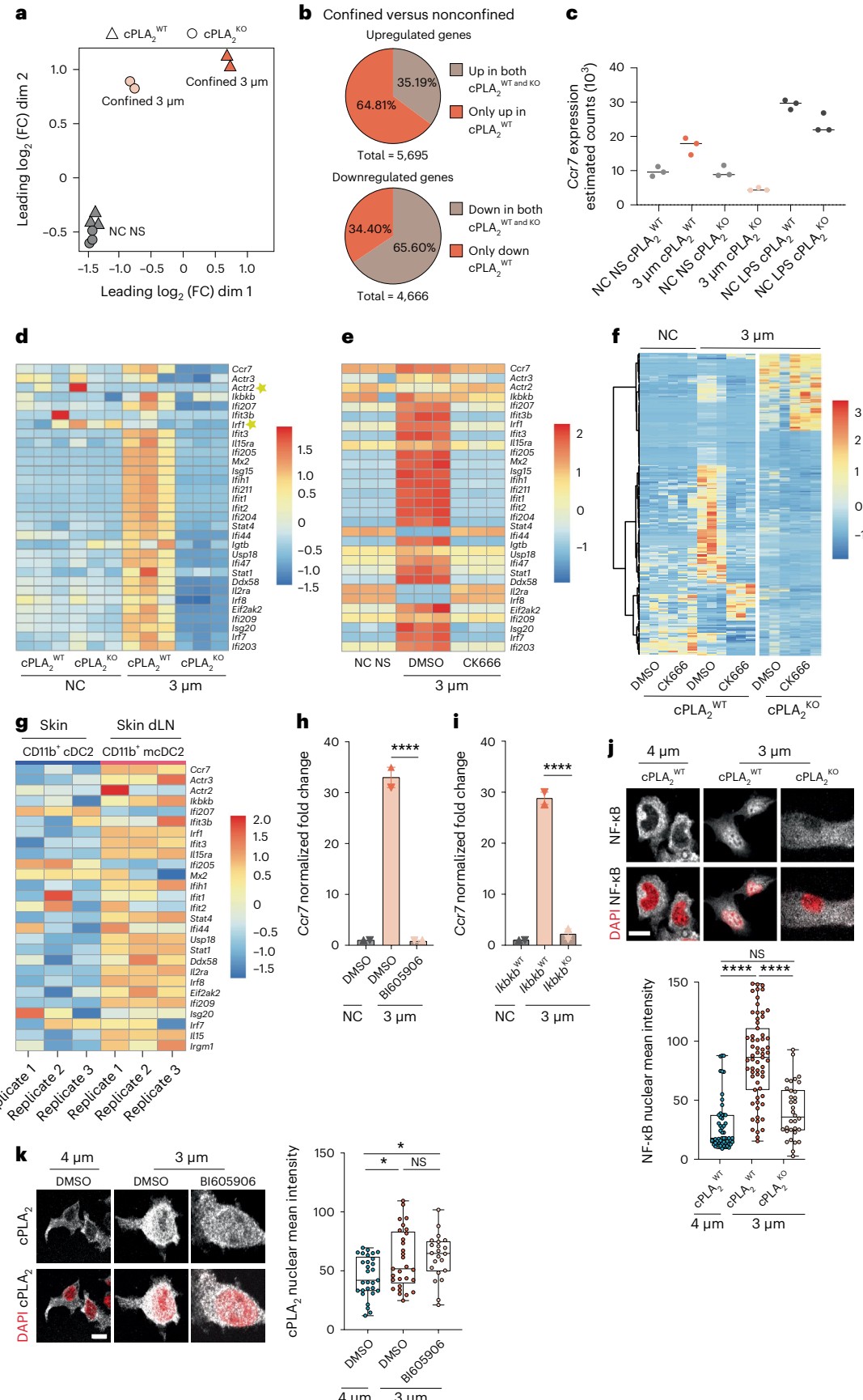

Accordingly, we observed that genes encoding MHC class I, MHC class II and key co-stimulatory molecules (CD80 and CD86) were expressed at lower levels in confined DCs than in LPS-treated cells (Extended Data Fig. 5c,d). Confined DCs also expressed fewer stimulatory cytokines (interleukin-2 (IL-2), IL-12, IL-15 and IL-27), resulting in lower secretion levels (Extended Data Fig. 5c,e). This suggests that confined DCs might be less potent than LPS-treated cells for T lymphocyte activation. To test this hypothesis, we loaded the two types of DCs with the class II ovalbumin (OVA) antigenic peptide and incubated them with OT-II transgenic T cells. OT-II cell stimulation with DCs that experienced confinement led to both lower T cell activation (quantified as upregulation of CD69) and lower proliferation (quantified by CFSE dilution) than stimulation with LPS-treated cells (Fig. 7f). Cell viability was comparable under both conditions (Extended Data Fig. 5f). These data confirmed that confined DCs were less potent than LPS-treated cells in activating T lymphocytes. In support of these results, when applying a threshold to the 3,000 genes upregulated in confined DCs in an ARP2/3- and cPLA$_2$-dependent manner, we obtained a list of 467 upregulated genes mostly related to cytokines/chemokines and dominated by the pathway of the tolerogenic cytokine IL-10 (see Extended Data Fig. 6a,b for the top 30 genes). Together, our data suggest that the ARP2/3–cPLA$_2$ shape-sensing pathway can not only tune steady-state migration of DCs to lymph nodes but also further endow them with specific immunoregulatory properties.

To gain insights into the mechanisms accounting for this specificity, we compared the transcription factor binding sites found in the promoters of the genes enriched in DCs confined at a height of 3 µm or treated with LPS. This analysis allows us to infer the nature of the transcription factors differentially implicated in the two types of DC responses. We found that DC confinement activated IRF1-, STAT1-, STAT3- and STAT5A-dependent gene transcription better than LPS (Fig. 7g). These transcription factors are known to be involved in IFN signaling and/or activation of NF-κB[43], in good agreement with our results showing that these two pathways are enriched following shape sensing (Fig. 6d). Noticeably, IRF1, STAT3 and STAT5A have been implicated in the acquisition of tolerogenic properties by DCs[44,45]. None of these transcription factors were activated in confined cPLA$_2^{KO}$ DCs (Fig. 7g). Together, our results suggest that the interplay between the cytoskeleton and the lipid metabolism enzyme cPLA$_2$ transcriptionally reprograms DCs in response to precise shape changes, endowing them with the ability to reach lymph nodes in an immunoregulatory state that might facilitate the tolerogenic function proposed for these cells at steady state.

## Discussion

Here, we show that DCs are equipped with an exquisitely sensitive shape-sensing mechanism that defines their migratory behavior and immune phenotype. This mechanism requires ARP2/3 nuclear envelope tensioning and translocation of the cPLA$_2$ lipid metabolism enzyme into the nucleus. cPLA$_2$ in turn activates NF-κB, leading to upregulation of CCR7 expression and DC migration to lymph nodes at steady state. These findings might explain why, despite two decades of research, the signals responsible for homeostatic DC migration from the periphery to the lymph nodes have remained largely unknown. Rather than sensing (bio)chemical signals, DCs might sense the physical constraints they encounter during environment patrolling through this ARP2/3–cPLA$_2$–NF-κB shape-sensing pathway. Interestingly, although this mechanical pathway shares signaling players, such as NF-κB, with the pathway induced by microbial sensing, it does not involve the direct engagement of DC receptors such as Toll-like receptors (TLRs) and NOD-like receptors (NLRs) and leads to an overlapping but distinct transcriptional program in these cells.

Previous work from us and others suggests that a specific threshold of nuclear deformation can promote nuclear envelope tensioning and cPLA$_2$ insertion into the nuclear membrane, leading to its activation[21,23]. Here, we provide direct evidence for this threshold being tuned by the actin cytoskeleton by showing that (1) ARP2/3 activity promotes nuclear envelope unfolding and tensioning in response to confinement, and (2) confinement induces the appearance of a large actin cloud in the vicinity of the nucleus, probably as a result of cortical actin flow directed toward the cell center, as previously shown in fibroblasts[46]. We propose that flowing cortical actin, when pressed against the nucleus after confinement, exerts forces, potentially through the LINC complex, to unfold the nuclear lamina and tense the nuclear membrane. The increased nuclear/cytosolic ratio of cPLA$_2$ observed in DCs confined at a height of 3 µm could be due to its insertion into the tensed inner nuclear envelope. Of note, we do not exclude that ARP2/3-dependent forces might also open nuclear pores to enhance cPLA$_2$ translocation from the cytoplasm to the nucleus, similar to what had been shown for YAP/TAZ[47], thereby contributing to nuclear accumulation of the enzyme.

We postulate that self-activation of the ARP2/3–cPLA$_2$ axis originates from the successive events of deformation that DCs undergo as they move through the complex environment of tissues. These deformation events would induce a 'cellular massage' and thereby activate shape sensing in DCs. Notably, the ARP2/3–cPLA$_2$ pathway is induced at a precise amplitude of cell deformation compatible with the shape changes observed in DCs patrolling the skin (Fig. 1a)[19,48]. Yet, the physical constraints imposed by the environment are likely to vary in distinct tissues[49] or pathological contexts, such as the tumor environment[50]. Determining whether the range of sensitivity of DCs to deformation is adapted to each tissue would be of the utmost importance to understand how the ARP2/3–cPLA$_2$ shape-sensing pathway controls both DC tissue exit to lymph nodes and their immunoregulatory properties. Nonetheless, our results emphasize that the physical properties of tissues indeed contribute to shaping immunity. The ability of immune cells to integrate these physical properties with biochemical environmental cues should be investigated.

**Fig. 7 | cPLA$_2$-dependent transcriptional reprogramming in response to shape sensing shapes DC properties. a**, MDS of samples. **b**, Heat map of cPLA$_2$-related genes. **c**, Top, representative images of untreated control cPLA$_2^{KO}$ DCs and cPLA$_2^{KO}$ DCs treated with 2.5 µg ml$^{-1}$ PGE$_2$ for 30 min in the presence or absence of BI605906 (30 µM) before confinement. CCR7 is in gray, and nuclei are in red; scale bar, 10 µm; Gaussian blur of 0.5 µm. Right, box plot showing the minimum to maximum range of CCR7 intensity ($N = 3$; $n = 87$ cells in 3 µm (control), $n = 99$ cells in 3 µm (PGE$_2$), $n = 87$ cells in 3 µm (PGE$_2$ + BI605906)). Data were analyzed by Kruskal–Wallis test; ****$P < 0.0001$; **$P = 0.0026$. Bottom, representative images of nonconfined untreated DCs or DCs treated with 2.5 µg ml$^{-1}$ PGE$_2$ for 30 min in the presence or absence of BI605906 (30 µM). NF-κB (P65) is in gray, and nuclei are in red; scale bar, 10 µm; Gaussian blur of 0.5 µm. Right, box plot showing the minimum and maximum range of NF-κB intensity ($N = 3$; $n = 133$ cells (control), $n = 191$ cells (PGE$_2$), $n = 138$ cells (PGE$_2$ + BI605906)). Data were analyzed by Kruskal–Wallis test; ****$P < 0.0001$. **d**, Pathway analysis of differentially expressed genes in confined versus LPS-treated DCs; arrow width corresponds to the enrichment score from the significantly enriched ($P < 0.05$) genes. **e**, Heat map of genes in the 'regulation of helper T cell differentiation' pathway marked in blue in **d**. **f**, Antigen presentation assays with T cells incubated with DCs activated with LPS or confined at a height of 3 µm. Left, percentage of CD69$^+$CD4$^+$ T cells among live cells after 18 h of incubation with OVA peptide II (OVAp). Right, Percentage of CFSE$^-$CD4$^+$ OT-II T cells among live cells after 3 days of incubation with OVA peptide II. Data are shown as mean ± s.e.m. of duplicate measures of each independent experiment ($N = 5$). Data were analyzed by multiple unpaired Student's $t$-tests; ***$P = 0.0001$; ****$P < 0.0001$. **g**, Transcription factor analysis of the different conditions. Transcription factor activity estimation used the TRUUST database, which predicts transcription factor activity and assigns a score. Arrow thickness corresponds to the enrichment score of each transcription factor.

Importantly, the ARP2/3–cPLA$_2$ shape-sensing pathway is the first pathway identified so far to be specifically involved in homeostatic DC migration to lymph nodes, a process essential for the maintenance of tolerance. We show that this shape-sensing axis has no impact on DC transcriptional reprogramming by microbial components such as LPS. Even though confined and LPS-treated DCs share a considerable part of their transcriptional program, LPS treatment led to activation of specific pathways associated with efficient T cell activation, which were less induced by confinement. Consistently, confined DCs were less

efficient in activating T cells than LPS-treated cells. These results are in agreement with the tolerogenic function proposed for homeostatic DC migration to lymph nodes[41,51]. Furthermore, several of the transcription factors specifically activated in confined DCs have been described as associated with immune tolerance[44], and examining genes that were most upregulated after shape sensing in an ARP2/3–cPLA$_2$-dependent manner revealed IL-10 signaling as the predominant pathway. However, we acknowledge that these results only constitute indirect evidence for shape sensing, leading to tolerogenic DCs, and that additional

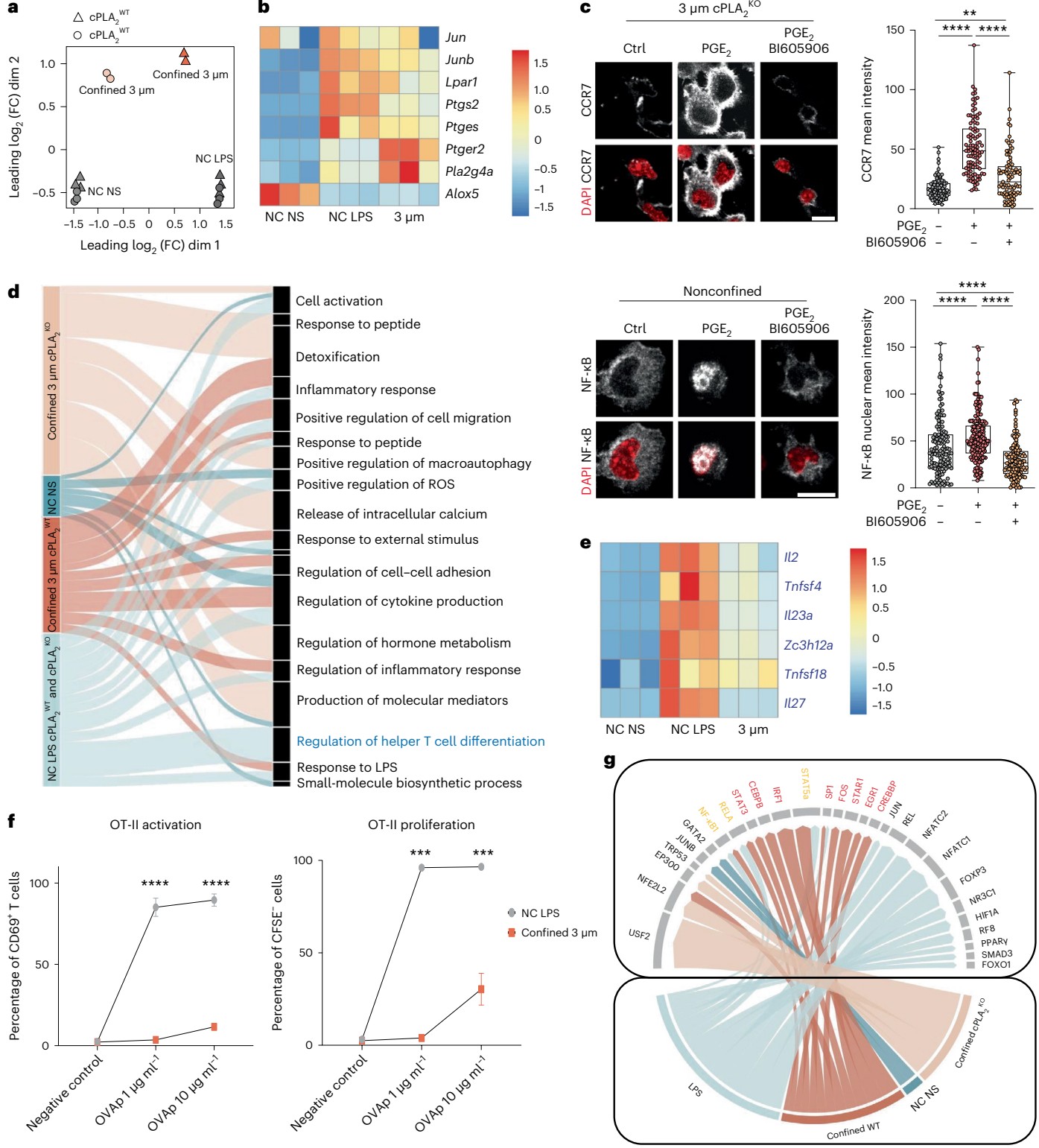

functional and in vivo studies will be required to directly visualize cPLA$_2$ activation in tissue-patrolling DCs and to show that these cells can protect mice from autoimmunity.

Interestingly, it has recently been shown that spleen CCR7$^+$ tolerogenic DCs can also emerge after internalization of apoptotic bodies, a process referred to as efferocytosis[52]. Along the same line, efferocytosis can trigger a tolerogenic phenotype in CCR7$^+$ tumor-infiltrating DCs by regulating cholesterol metabolism[53]. Future studies aimed at determining whether and how the here described shape-sensing pathway functionally interacts with these efferocytosis-dependent pathways should shed light on how the balance between tolerance and immunity is critically tuned by distinct tissue environmental properties.

## Online content

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

[1]INSERM U932, Immunity and Cancer, Institut Curie, PSL University, Paris, France. [2]Cellular Immunology, International Centre for Genetic Engineering and Biotechnology (ICGEB), Trieste, Italy. [3]INSERM U1015, Gustave Roussy Cancer Campus, Villejuif, France. [4]Cell Communication and Migration Laboratory, Institute of Biochemistry and Molecular Cell Biology, Center for Experimental Medicine, University Medical Center Hamburg-Eppendorf, Hamburg, Germany. [5]CNRS UMR144, Institut Curie, PSL Research University, Paris, France. [6]Malaghan Institute of Medical Research, Wellington, New Zealand. [7]Center for Research in Transplantation and Translational Immunology, UMR 1064, INSERM, Nantes Université, Nantes, France. [8]INSERM U932, Immunity and Cancer, Institut Curie, Paris-Cité University, Paris, France. [9]Platform NGS-ICGEX, Institut Curie, Paris, France. [10]INSERM UMR1291, CNRS UMR5051, Institut Toulousain des Maladies Infectieuses et Inflammatoires (INFINITy), Université Toulouse III, Toulouse, France. [11]Immunity, Inflammation, and Disease Laboratory, National Institute of Environmental Health Sciences (NIEHS), National Institutes of Health (NIH), Research Triangle Park, NC, USA. [12]Centre d'Immunologie de Marseille-Luminy, INSERM, CNRS, Université Aix-Marseille, Marseille, France. [13]Centre for Inflammation Biology and Cancer Immunology, School of Immunology and Microbial Sciences, King's College London, London, UK. [14]Henan Key Laboratory of Immunology and Targeted Therapy, School of Laboratory Medicine, Xinxiang Medical University, Xinxiang, China. [15]Singapore Immunology Network (SIgN), Agency for Science, Technology and Research (A*STAR), 8A Biomedical Grove, Immunos, Singapore, Singapore. [16]Department of Immunology and Microbiology, Shanghai Institute of Immunology, Shanghai Jiao Tong University School of Medicine, Shanghai, China. [17]Translational Immunology Institute, SingHealth Duke-NUS Academic Medical Centre, Singapore, Singapore. [18]Department of Pathology and Laboratory Medicine, Children's Hospital of Philadelphia and University of Pennsylvania Perelman School of Medicine, Philadelphia, PA, USA. [19]These authors contributed equally: Maria-Graciela Delgado, Doriane Sanséau. [20]These authors jointly supervised this work: Guilherme P. F. Nader, Matthieu Piel, Ana-Maria Lennon-Duménil. ✉e-mail: matthieu.piel@curie.fr; ana-maria.lennon@curie.fr

## Methods

### Mice

For animal care, we strictly followed the European and French National Regulation for the Protection of Vertebrate Animals used for Experimental and other Scientific Purposes (Directive 2010/63; French Decree 2013-118, Authorization DAP number 43530-2023051216135493 v2 given by National Authority). Experiments were performed on 8- to 14-week-old male or female mice. Mice were maintained under specific pathogen-free conditions at the animal facility of Institut Curie, in accordance with institutional guidelines, and housed in a 12-h light/12-h dark environment with free access to water (osmotic water) and food (MUCEDOLA, 4RF25SV Aliment Pellets 8 × 16 mm irradiated 2.5 Mrad and DietGel Recovery, 2 oz/DietGel Boost, 2 oz). Littermates or age-matched mice were used as controls for all experiments involving knockout animals. Breeder mice were previously backcrossed to C57BL/6 mice for seven generations. C57BL6/J mice were obtained from Charles River (000664). CCR7–GFP mice were obtained from Jackson Laboratory (027913) and were bred in our animal facility[30]. *Itgax-cre* mice were bred in our animal facility[54]. *Arpin-* and *Pla2g4a*-conditional-knockout mice were generated by Centre d'Immunophénomique using CRISPR–Cas9 technology and were crossed in our animal facility with *Itgax-cre* mice. *Ikbkb*-knockout mice were from the laboratory of T. Lawrence (Centre d'Immunologie de Marseille-Luminy, Marseille, France), as described in Baratin et al.[41]. WASp[KO] mice on a C57BL/6 (CD45.2) genetic background were from the laboratory of F.B. Experiments were performed using homozygous WASp[−/−] mice as knockout animals. *Lmna*[fl/fl]*Vav1-cre*[+/−] (lamin A/C CK[55,56])mice were obtained from the laboratory of N.M. YFP–CD11c mice were from the laboratory of S. Amigorena (Institute Curie, U932, Paris, France)[57].

### Cells

DCs were obtained following a protocol adapted by the Ricciardi-Castagnoli and Amigorena labs. Using this differentiation protocol >90% of the cells recovered after 10 days of culture are positive for the CD11c marker, in addition to MHC class II[58,59]. In brief, both whole legs from 6- to 8-week-old mice were flushed to obtain bone marrow. Cells were maintained in culture for 10 days in IMDM (Sigma-Aldrich) containing decomplemented and filtered 10% fetal bovine serum (FBS; Biowest), 20 mM L-glutamine (Gibco), 100 U ml$^{-1}$ penicillin–streptomycin (Gibco), 50 µM 2-mercaptoethanol (Gibco) and 50 ng ml$^{-1}$ granulocyte–macrophage colony-stimulating factor containing supernatant obtained from transfected J558 cells tested by enzyme-linked immunosorbent assay. At days 4 and 7 of culture, cells were detached using PBS-EDTA (5 mM) and replated at 0.5:10$^6$ cells per ml of medium. The obtained cells were then used at day 10 or 11 as immature cells. This differentiation protocol also promotes the differentiation of granulocytes and macrophages, but they can be separated from DCs based on their adhesion. Granulocytes are eliminated during differentiation because they are nonadherent, whereas macrophages are much more adherent than the other two cell types and stick to the bottom of the plate. The semiadherent fraction of cells, which corresponds to DCs, was thus recovered at day 10 or 11.

**DC maturation.** Day 10 cells were stimulated with 100 ng ml$^{-1}$ LPS (*Salmonella enterica* serotype Typhimurium; Sigma) for 25 min, washed three times with complete medium, replated in fresh medium, left overnight in the incubator and used.

### Six-well plate confiner

Confinement was performed using a six-well plate confiner as previously described[26]. To make the polydimethylsiloxane (PDMS; RTV615) pillars at a certain height, 12-mm glass coverslips were sonicated in methanol, washed in ethanol, plasma treated and placed on the top of a PDMS mixture on top of wafer molds that contained the pillars at the desired height. The PDMS mixture was composed of PDMS A:cross-linker B at a ratio of 1:10 (wt/wt). After adding the coverslips to the wafers, the wafers were baked at 95 °C for 15 min and carefully removed using isopropanol. The wafers were then washed again with isopropanol, dried and plasma treated for 2 min. Next, the molds were incubated with nonadhesive pLL-PEG (SuSoS, PLL (20)-g [3.5]-PEG (2)) at 0.5 mg ml$^{-1}$ in 10 mM HEPES (pH 7.4) buffer for 1 h at room temperature. The coverslips were washed well with water to remove all remaining PEG and incubated in cell medium for 2 h before confinement. To perform the confinement steps, we used a modified version of a classical six-well plate cover; large PDMS pillars were stuck on the coverlid. Six-well plates with a glass bottom were used for cell imaging (MatTek, P06G-1.5-20-F).

### Live imaging experiments

Live timelapse recordings were acquired by Nikon video microscopy with a ×20/0.75-NA dry objective for 4–6 h at 37 °C with 5% CO$_2$ atmosphere, by confocal microscopy (Leica DMi8, SP8 scanning head unit) with a ×40/1.3-NA oil objective with a resolution of 1,024 × 1,024 pixels or by inverted spinning desk confocal microscopy with a ×60/1.4-NA oil objective. All microscopes were controlled by Meta Morph software. Images were processed using ImageJ software[60] (NIH, http://rsb.info.nih.gov/ij/index.html), and analyses were performed using the same software. The following drugs and reagents were used for live imaging experiments: NucBlue (Hoechst33342; DNA marker; Thermo Fischer, R37605), AACOCF3 (selective phospholipase A$_2$ inhibitor; Tocris, 1462-5), CK666 (selective ARP2/3 inhibitor; Tocris, 3950), BI605906 (selective IKKβ inhibitor; Tocris, 53001) and PGE$_2$ (Stem Cell, 72372).

For GFP quantification, imaging was performed using both phase contrast and GFP channels. Images corresponding to each time point of interest were collected for each condition, the outline of each cell was drawn by hand on the *trans* images, and the mean GFP signal was calculated on the corresponding GFP image using a homemade macro. Of note, only cells with an intensity higher than the background were plotted.

### Lentivirus transduction

Transduced DCs were obtained by transfection of bone marrow-derived DCs. Bone marrow-derived DCs were plated at a concentration of 1 million cells per 2 ml of medium. On day 4, 40 ml of fresh pTRIP-SFFV-GFP-NLS lentivector supernatant was loaded in Ultra-Clear Centrifuge tubes (Beckman Coulter) and ultracentrifuged at 100,000$g$ in a SW32 rotor (Beckman Coulter) for 90 min at 4 °C and resuspended in 400 µl of DC medium. Ultracentrifuged virus (200 µl) was used to infect one well of cells in the presence of 8 µg ml$^{-1}$ protamine. Cells were then left for 48 h, washed to remove the viral particles and incubated in new medium until day 10 of culture. For the NLS–GFP construct, pTRIP-SFFV-EGFP-NLS was generated introducing the SV40 NLS sequence (PKKKRKVEDP) by overlapping PCR at the C terminus of GFP in pTRIP-SFFV.

### Immunofluorescence microscopy

After removing the confinement, samples were fixed directly with 4% paraformaldehyde for 30 min, permeabilized with 0.2% Triton X-100 and incubated overnight with primary antibodies at 4 °C. The next day, samples were washed with PBS and incubated with the corresponding secondary antibodies for 1 h, washed three times with PBS and mounted with Fluoromount solution. Imaging was performed using a confocal microscope (Leica DMi8, SP8 scanning head unit) with a ×40/1.3-NA oil objective and a resolution of 1,024 × 1,024 pixels. The following primary antibodies were used for the immunofluorescence stainings: anti-CCR7 (Abcam, ab32527; 1:100), anti-cPLA$_2$ (Cell Signaling (Ozyme), 2832; 1:100), anti-NF-κB P65 (Cell Signaling, mAB 8242; 1:200), anti-LAP2 (BD Biosciences, 611000; 1:200) and Alexa Fluor-coupled phalloidin (Invitrogen; 1:300). The following secondary antibodies were used:

Alexa Fluor 647 goat anti-rabbit (Invitrogen, A21244; 1:300) and Alexa Fluor 488 goat anti-mouse Fab₂ (Invitrogen, A11017; 1:300).

**Calculation of fluorescence intensity.** Images were acquired as *z* stacks with a step size of 0.33 µm for each position. Using a homemade macro, the plane of the nucleus was used to quantify the florescence intensity by making a mask on both the nucleus (DAPI channel) and the cell (phalloidin channel).

### Transfection and short interfering RNA (siRNA)
Bone marrow-derived DCs at day 7 ($3 \times 10^6$) were transfected with 100 µl of Amaxa solution (Lonza) containing siRNA (control or target specific) following the manufacturer's protocol. Cells were further cultured for 48 h. The following SMARTpool siRNAs were used: ON-TARGETplus *Pla₂g4a* siRNA (Dharmacon, L-009886-00-0010) and ON-TARGETplus Non-Targeting Control Pool (Dharmacon, D-001810-10-20).

### RT–qPCR
After removing the confinement, lysis buffer was added directly to recover the confined cells. RNA extraction was performed using an RNeasy Micro RNA kit (Qiagen) according to the manufacturer's protocol. cDNA was produced using a high-capacity cDNA synthesis kit (Thermo Fisher), according to the manufacturer's protocol, starting from 1 µg of RNA. qPCR experiments were performed using Taqman Gene Expression Assays (Applied Biosystems) on a Lightcycler 480 (Roche) using the settings recommended by the manufacturer. The following primers were used: Mm99999130_s1 for *Ccr7*, Mm01284324_m1 for *Pla₂g4a* and Mm99999915 for *Gapdh* (endogenous control). The expression of each gene of interest was assessed in immature nonconfined cells. Samples were run in triplicate for each condition. Data were subsequently normalized to *Gapdh* values. Values obtained in control immature cells were used as a base unit equal to 1, thus allowing for display of the data as 'fold greater' than immature cells. Fold change was calculated using the formula $2^{-\Delta\Delta C_t}$.

### Flow cytometry
After confining the cells, the confiner was removed, and cells were recovered directly by gently washing with medium. Cells were resuspended in buffer (PBS supplemented with 1% bovine serum albumin and 2 mM EDTA). After blocking with Fc antibody (BD Biosciences, 553142; 1:1,000) and processing with a live/dead staining kit (Thermo Fischer, L34966) for 15 min, cells were stained with the desired antibodies for 20 min (at 37 °C for CCR7 staining and at 4 °C all other antibodies). Cells were then washed three times and resuspended in staining buffer. Flow cytometry was performed on an LSRII flow cytometer (BD Biosciences), and data were analyzed using FlowJo software version 10. Mean florescence values of each condition were plotted with GraphPad Prism version 8. The following antibodies were used: anti-CD80 (BD Biosciences, 553769, clone 16-10A1; 1:100), anti-MHC class II (Ozyme, BLE107622, clone M5/114.15.2; 1:400), anti-CD86 (Biolegend, 105037, clone GL-1; 1:200) and anti-CD11c (BD Biosciences, 550261, clone HL3; 1:200) with the corresponding isotypes to each antibody.

For the T cell presentation assay, OVA peptide (Cayla, vac-isq) was added to each condition with the corresponding concentration. Cells were then confined, recovered and counted. T cells were purified from OT-II mice, stained with CFSE and added to the recovered DCs at a ratio of 10:1, respectively. Cells were plated in round-bottom, 96-well plates. OT-II T cell activation was analyzed 18 h later, and after 3 days, proliferation was measured by flow cytometry. Cells were stained in 2 mM EDTA and 5% FBS in PBS following the same protocol as described earlier. Flow cytometry was performed on an LSRII flow cytometer (BD Biosciences), and data were analyzed using FlowJo software version 10. Percentage values were plotted with GraphPad Prism version 8. The following antibodies were used: anti-CD4 (BD Biosciences, 553051, clone RM4-5; 1:100), anti-CD69 (eBioscience, 48-0691-82, clone H1.2F3; 1:300) and anti-TCR (BD Biosciences, 553190, clone MR9-4; 1:300).

### RNA-seq
After removing the confinement, lysis buffer was added directly to recover the confined cells. RNA extraction was performed using an RNeasy Micro RNA kit (Qiagen), according to the manufacturer's protocol. RNA-seq libraries were prepared from 300 ng to 1 µg of total RNA using an Illumina TruSeq Stranded mRNA Library Preparation kit and an Illumina Stranded mRNA Prep Ligation kit, which allowed strand-specific RNA-seq. First, poly(A) selection using magnetic beads was performed to focus sequencing on polyadenylated transcripts. After fragmentation, cDNA synthesis was performed, and the resulting fragments were used for dA-tailing and were ligated to the TruSeq indexed adapters (for the TruSeq kit) or RNA Index Anchors (for the mRNA Ligation kit). PCR amplification was performed to generate the final indexed cDNA libraries (with 13 cycles). Individual library quantification and quality assessment was performed using a Qubit fluorometric assay (Invitrogen) with a dsDNA High-Sensitivity Assay kit and LabChip GX Touch using a High-Sensitivity DNA chip (PerkinElmer). Libraries were then pooled at equimolar concentrations and quantified by qPCR using a KAPA library quantification kit (Roche). Sequencing was performed on a NovaSeq 6000 instrument from Illumina using paired-end 2 × 100 bp sequencing to obtain around 30 million clusters (60 million raw paired-end reads) per sample.

### RNA-seq data analysis
Library read repartition (for example, for potential ribosomal contamination), inner distance size estimation, gene body coverage and strand specificity were performed using FastQC, Picard-Tools, Samtools and RseQC. Reads were mapped using STAR[61] on the mm39 genome assembly.

Gene expression was estimated as described previously[62] using Mouse FAST DB v2021_2 annotations. Only genes expressed in at least one of the two compared conditions were analyzed further. Genes were considered expressed if their fragments per kilobase per million mapped reads (FPKM) value was greater than the FPKM of 98% of the intergenic regions (background). Analysis at the gene level was performed using DESeq2 (ref. 63). Genes were considered differentially expressed if the fold change value was ≥1.5 and the *P* value was ≤0.05. Pathway analyses and transcription factor network analyses were performed using WebGestalt 0.4.4 (ref. 64), merging results from upregulated and downregulated genes only as well as all regulated genes. Pathways and networks were considered significant with a false discovery rate of ≤0.05. Graphics (heat map, MDS, scatter and volcano plots) were generated using R v4.2.1 with the help of pheatmap, ggplot2 and EnhancedVolcano[65] packages, respectively. Heat maps were created using the *z* score of EdgeR-normalized counts.

### Analysis of microarray data
GSE49358 microarray data[39] were downloaded from Gene Expression Omnibus (GEO) and annotated using the 'mogene10stprobeset.db' R package (v. 2.7). cPLA signature genes were displayed as heat maps using tidyverse (v. 1.3.0), scales (v. 1.1.1) and readxl (v. 1.3.1).

### In vivo analysis of DC subsets
For the in vivo analysis of DC subsets, both male and female wild-type and knockout littermates at the age of 8–12 weeks (*n* = 6 to 9 mice per experiment) were used. To collect cells from the skin, a section of skin (1 cm²) was cut and transferred to a 1-ml Eppendorf tube containing 0.25 mg ml⁻¹ liberase (Sigma, 5401020001) and 0.5 mg ml⁻¹ DNase (Sigma, 10104159001) in 1 ml of RPMI (Sigma). The skin was cut using scissors and incubated for 2 h at 37 °C. To collect cells from lymph nodes, inguinal lymph nodes were removed and transferred to an Eppendorf tube containing 500 µl of RPMI. DNase and collagenase were

added at 0.5 mg ml$^{-1}$ and 1 mg ml$^{-1}$, respectively. The lymph nodes were further cut with scissors and incubated for 20 min at 37 °C. Cells were resuspended in PBS supplemented with 0.5% bovine serum albumin and 2 mM EDTA at 4 °C, filtered and stained following the same protocol as described earlier. The following antibodies were used: anti-CCR7 (Biolegend, 120114, clone 4B12; 1:50), anti-CD11c (eBioscience, 25-0114-81, clone N418; 1:800), anti-CD326 (Biolegend, 118217, clone G8.8; 1:800), anti-CD86 (BD Pharmingen, 553692, clone GL-1; 1:800), anti-CD11b (Biolegend, 101237, clone M1/70; 1:500), anti-MHC class II (eBioscience, 56-5321-80, clone I-A/I-E-M5/114.15.2; 1:250), anti-CD45 (eBioscience, 61-0451-82, clone 30-F11; 1:100), anti-CD8a (BD Bioscience, 553035, clone 53-6.7; 1:100) and anti-CD103 (eBioscience, 46-1031-80, clone 2E7; 1:100). Counting beads were used to normalize the number of cells to the number of beads. Flow cytometry was performed on an LSRII flow cytometer (BD Biosciences), and data were analyzed using FlowJo software version 10. Percentage values were plotted with GraphPad Prism version 8.

## Calculation of EOP$_{NE}$

To calculate the EOP, we made a mask on the nucleus, then from the RIO function in ImageJ, we obtained the values of nuclear 'area' and 'perimeter' ($P$). We then calculated $R_0$, which is the radius of a circle defined by the area of the nucleus. The ratio between $P - 2\pi R_0$ and $2\pi R_0$ was calculated as the EOP$_{NE}$. EOP values close to 1 indicate a highly folded (and thus less tensed) envelope, whereas EOP values close to 0 indicate a smooth surface with fewer folds (and thus a more tensed envelope).

## Calculation of EFC

Based on the work by Tamashunas et al.[66], we fit contour with sum of successive Fourier ellipses (total number of ellipses for good fit of 25). We first made a mask of all the nucleus that needed to be analyzed and then automatically compared the relative contributions of the first-order ellipse to those of the later-order ellipses. The larger the EFC ratio, the more regular/smooth the shape. The EFC ratio is the sum of major and minor axes of the first ellipse divided by the sum of the major and minor axes of all others

## Multiphoton imaging of immune cells in mouse ear skin ex vivo

CD11c–EYFP mice were injected intravenously with 20 µl of CD31 (Thermo Fisher, 5278509) 5 min before being killed. Mouse ears were retrieved by cutting them with scissors and placing them in RPMI at 37 °C. Hair was removed with forceps. Ears were mechanically opened in two sheets with two forceps and were maintained in RPMI. The thick part of the posterior sheet was cut and prepared for imaging by piercing a piece of tape with several holes using a 29-G needle. Using a tissue, the epidermis of the posterior ear sheet was dried and placed on the pierced tape. The sheet was flattened as much as possible without damaging the dermis and placed at the bottom of a fluorodish, and 1 ml of preheated imaging medium was added. The tissue was imaged with a Biphoton microscope (×20; CD11c (orange), CD31 (blue) and fibrillar collagen (gray)).

## Calculation of the minimum and maximum diameter of DCs in ear skin

We selected several cells from the field of view and duplicated their corresponding movies. The cell masks were created using the thresholding function in ImageJ. Subsequently, we duplicated the movie with the mask and applied the 'skeletonize' function in ImageJ. This function tags all the pixels/voxels in a skeleton image and then counts all junctions, triple and quadruple points and branches. It automatically generated a region of interest with all the different frames of the cells. For the other mask's movie, we applied the 'distance map' function, which generates a Euclidean distance map. Each pixel in the image was replaced with a gray value equal to the pixel's distance from the nearest background pixel. The region of interest with the different frames was measured on the distance map of each frame for one cell. Measurements of the average, minimum and maximum lengths of each frame were calculated. We then calculated the average minimum (or maximum) diameter for one cell over several frames.

## Calculation of the time spent with a minimum cell diameter

For each cell, we duplicated the corresponding movie to track the cell over a minimum time scale of 40 min. We measured and recorded the minimum diameter of the cell at each time point throughout the entire duration of the movie. Subsequently, we calculated and plotted the time spent at a minimum diameter of 2–4 µm or at higher than 4 µm.

## FLIM measurements for nuclear envelope tension

HeLa Kyoto EMBO cells were cultured in DMEM supplemented with 10% FBS and 1% penicillin–streptomycin. ER Flipper-TR probe from Spirochrome (SC021) was added 30 minutes before the start of the experiment. For the FLIM experiments involving CK666, cells were plated in Fluorobright (Gibco) supplemented with 10% FBS, 1% penicillin–streptomycin, 1× GlutaMAX (Gibco) and 25 mM HEPES for 30 min and incubated with 30 µM CK666 and 1 µM ER Flipper-TR for 30 min.

## Statistics and data handling

Analyses were performed using GraphPad Prism 8 software. No statistical methods were used to predetermine sample sizes, but our sample sizes are similar to those reported in previous publications[21]. Data distribution was assumed to be normal (unless stated otherwise), but this was not formally tested. Mice were chosen randomly for the experiments. Experimental conditions were organized randomly. Data collection and analysis were not performed blind to the conditions of the experiments. No data points nor animals were excluded from the analysis.

## Reporting summary

Further information on research design is available in the Nature Portfolio Reporting Summary linked to this article.

## Data availability

RNA-seq data have been deposited in the GEO under the accession code GSE207653. Raw data quantification used to generate figures are available on FigShare at https://doi.org/10.6084/m9.figshare.25236715 (ref. 67). GSE49358 microarray data were accessed at https://www.ncbi.nlm.nih.gov/geo/query/acc.cgi?acc=GSE49358. The confinement protocol is deposited on Protocol Exchange at https://doi.org/10.21203/rs.3.pex-2616/v1 (ref. 68). Source data are provided with this paper.

## Code availability

Scripts of the macros used to analyze microscopy images are available at https://github.com/Zalraies/Alraies-at-al-2024n.

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

## Acknowledgements

This work was supported by the European Union's Horizon 2020 Research and Innovation Program under Marie Skłodowska-Curie grant agreement number 666003 (to Z.A.). This work also received support under the program 'Investissements d'Avenir' launched by the French Government (ANR-10-IDEX-0001-02 PSL and LabEx DCBIOL). This work has received support under the program France 2030 launched by the French Government and by la Fondation de la Recherche Médicale (SMC202006012351; to A.-M.L.-D.) and from ERC Synergy (101071470–SHAPINCELLFATE; to A.-M.L.-D. and M.P.). H.D.M. received support from ANR-20-CE15-0023 (InfEx). C.A.R. received support from ARC Foundation for Cancer Research. P.J.S. received support from the Human Frontier Science Program (RGP0032-2022) and Forschungszentrum Medizintechnik Hamburg (04fmthh2021). High-throughput sequencing was performed by the ICGex NGS platform of the Institut Curie supported by grants from the Agence Nationale de la Recherche ANR-10-EQPX-03 (Equipex) and ANR-10-INBS-09-08 (France Génomique Consortium) from the Agence Nationale de la Recherche ('Investissements d'Avenir' program), by the ITMO-Cancer Aviesan (Plan Cancer III) and by the SiRIC-Curie program (SiRIC grants INCa-DGOS-465 and INCa-DGOS-Inserm_12554). Data management, quality control and primary analysis were performed by the Bioinformatics Platform of the Institute Curie. We acknowledge the Cell and Tissue Imaging (PICT-IBiSA) platform, members of the French National Research Infractructure France-BioImaging (ANR-10-INBS-04) and the flow cytometry and mouse facilities at Institut Curie. We acknowledge the NeurImage facility of the Institute of Psychiatry and Neuroscience of Paris, where imaging experiments were performed. We thank C. Canet-Jourdan for help with the acquisition of video microscopy imaging.

## Author contributions

Z.A. designed, performed and analyzed most experiments and actively contributed to manuscript preparation by building figures and editing the manuscript. C.A.R. performed most of the flow cytometry experiments and analyses and antigen presentation experiments, helped edit the manuscript and generated the gating strategy figures. M.-G.D. and D.S. performed fluorescence-activated cell sorting (FACS) analyses of lymph nodes of *Arpin* mice and DCs derived from the bone marrow of these mice. M.-G.D. helped with animal management and developed Extended Data Fig. 7. M.M. helped Z.A. with the setup of the microscopy experiments and wrote macros for image analysis. R.A. and G.M.P. performed FACS analyses of lymph nodes from WASp[KO] mice and sent bone marrow samples from these mice. G.D. performed the RNA-seq data analysis and generated RNA-seq figures. A.Y. helped with DC confinement for antigen presentation experiments. L.L.M. performed FACS analyses of the lymph nodes from cPLA$_2$[KO] mice and, together with Z.F., performed single-cell RNA-seq experiments on their lymph nodes (data not shown). A.K. and V.C. performed experiments on the role of PGE$_2$. P.J.S. helped Z.A. with chemotaxis under confinement experiments and performed the analyses for these experiments. A.W. and H.P. performed the experiments and analyses of HeLa cell nuclear envelope tension. A.W. analyzed the EFC data. M.G. helped Z.A. with DC transduction of the NLS–GFP construct. O.L. performed the reanalysis of published data on skin and lymph node cDC2s and generated heat maps. A.M. helped perform cytokine measurement experiments. L.C. performed imaging experiments of DCs in mouse ear skin and generated live-cell videos. B.A. and P.L. prepared RNA-seq libraries and performed RNA-seq steps. A.S.D. and A.-L.L.-D. performed experiments that were not included in the manuscript to generate preliminary data but helped answer some reviewer questions. H.N. and D.N.C. provided bone marrow from the CCR7–GFP mice that they generated. T.L. provided bone marrow from *Ikbkb*-knockout mice and provided feedback throughout the study. N.M. provided bone marrow from *Lmna*-knockout mice and provided feedback throughout the study. F.B. provided bone marrow from WASp[KO] mice. F.G. supervised G.D. during the RNA-seq experiment analyses. H.D.M. gave essential conceptual feedback for this study, suggested key experiments, helped with mouse management, helped write several parts of the manuscript and trained Z.F. and L.C. for work with DCs. G.P.F.N. supervised Z.A. during the confinement experiments and helped with study conceptualization. M.P. and A.-M.L.-D. developed the initial hypothesis, supervised the study and wrote the manuscript. All authors contributed to manuscript preparation.

## Competing interests

The authors declare no competing interests.

## Additional information

**Extended data** is available for this paper at https://doi.org/10.1038/s41590-024-01856-3.

**Correspondence and requests for materials** should be addressed to Matthieu Piel or Ana-Maria Lennon-Duménil.

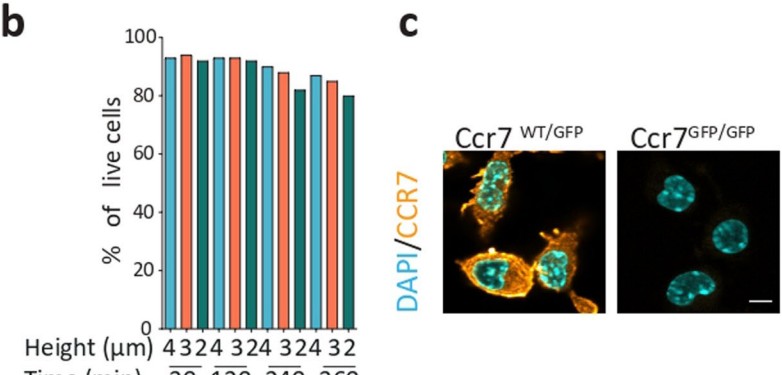

**a**

1. Choose the time point of interest
2. Draw each cell's outline and add to ROI manager
3. Measure the mean GFP intensity per cell
4. Total GFP Intensity = (mean intensity - median of the background) x cell area

**Extended Data Fig. 1 | GFP quantification approach.** (**a**) Quantification approach to quantify GFP intensity in each cells. (**b**) Quantification of cell viability in different confinement conditions and during different time points, using propidium iodide (**c**) representative images of immature DCs confined for 4 h at the height of 3 μm from C57BL/6 J (WT) or GFP/GFP mice (*Ccr7*KO), CCR7 was visualized using immunostaining (in orange) and the nucleus stained with DAPI (in cyan) scale bar 5 μm.

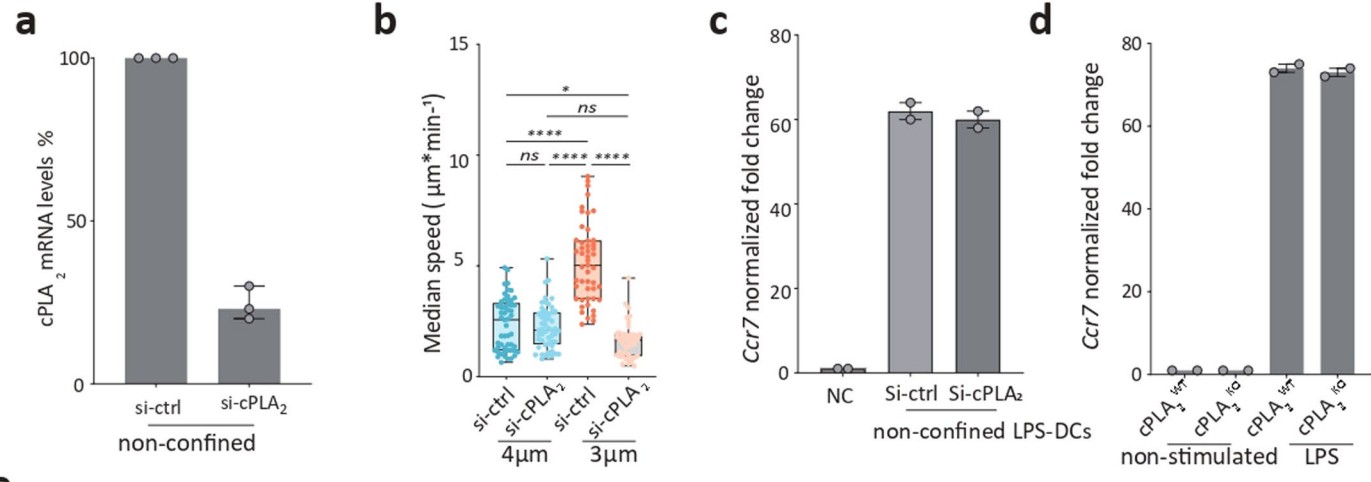

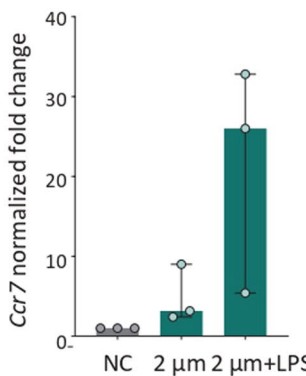

**Extended Data Fig. 2 | cPLA₂ activity is not necessary for LPS-induced *Ccr7* up regulation.** (**a**) median with interquartile range of cPLA₂ ^gene expression upon its knock down^, N = 3, n = 9 (**b**) Box plot with min to max range of median speed of cPLA₂ KD or control DCs. N = 3, n = 53 cells in cPLA₂ Ctrl 4μm, n = 53 cells in cPLA₂ KD 4μm, n = 49 cells in cPLA₂ Ctrl 3 μm, n = 44 cells in cPLA₂ KD 3 μm. one way ANOVA with Kruskal-Wallis multiple analysis test, ****:P < 0.0001, *: P = 0.0315, ns: not significant P = 0.0813. (**c**) RT-qPCR data of *Ccr7* gene expression shows no difference in *Ccr7* expression in cPLA₂ KD or control cells activated with the microbial component LPS, median with interquartile range of 2 independent experiments, n = 6 (**d**) RT-qPCR data of *Ccr7* gene expression in cPLA₂ WT and KO DCs activated with LPS ^activated with LPS^. median with interquartile range of 3 independent experiments. (**e**) RT-qPCR data of *Ccr7* gene expression in cells confined at 2 μm activated with LPS after confinement, median with interquartile range of 3 independent experiments, n = 9.

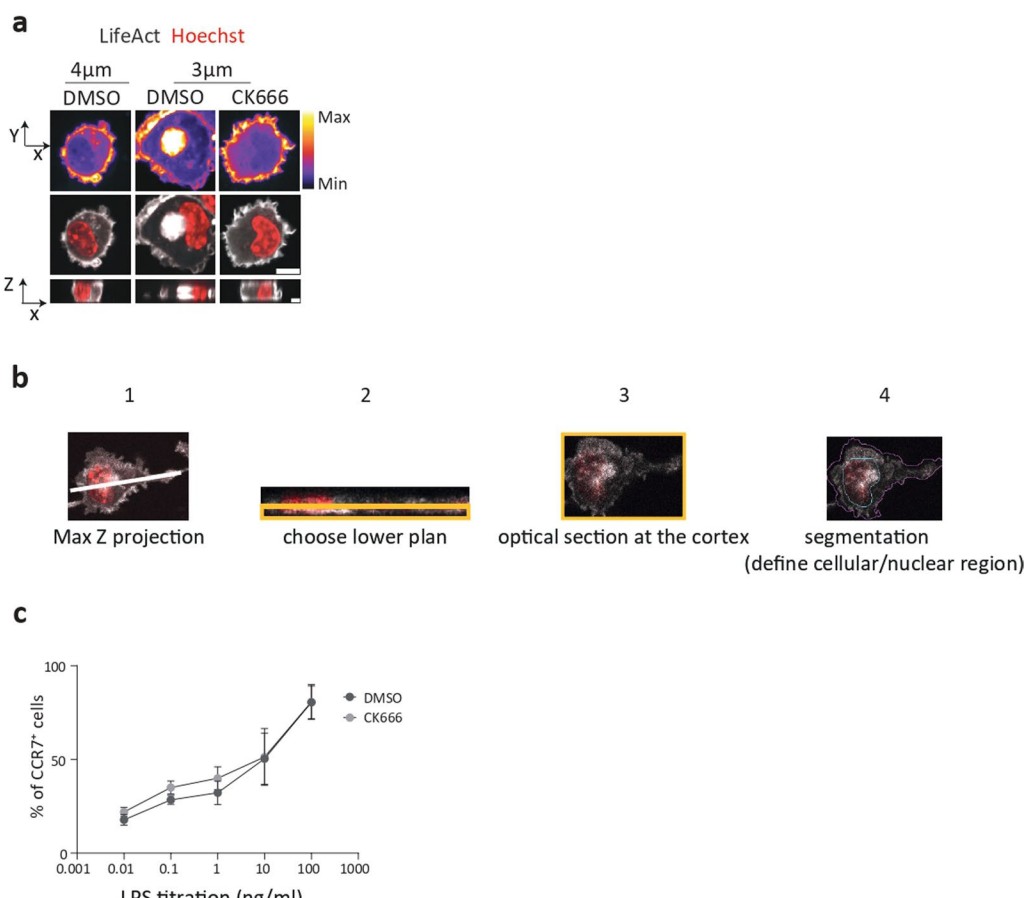

**Extended Data Fig. 3 | Arp2/3 branched actin is not important for CCR7 upregulation upon LPS.** (**a**) Second example: Upper panel: representative XY images of DCs expressing LifeAct-GFP (false colors), Middle panel: XY images of LifeAct-GFP DCs (in grey) stained with NucBlue (DNA, red) treated or not with CK666 (30 μM), scale bar 10 μ. lower panel: XZ view, single confocal frame from resliced images, scale bar 20 μ. (**b**) Quantification approach of LifeAct-GFP intensity in cells under confinement: 1- choose Z-stack lower plan (since cells don't always touch the upper plan). 2⁻ optical section at the surface cortex. 3-Segmentation to define cell and nuclear contour at the surface. 4- Measurement of LifeAct-GFP ratio: Actin Ratio= Nuclear surface mean actin intensity/cell surface mean actin intensity (ratio <1 => actin is mostly cortical). (**c**) FACS analysis of CCR7+ DCs activated by different concentration of LPS and treated with CK666 (30 μM) or DMSO. N = 3 independent experiments.

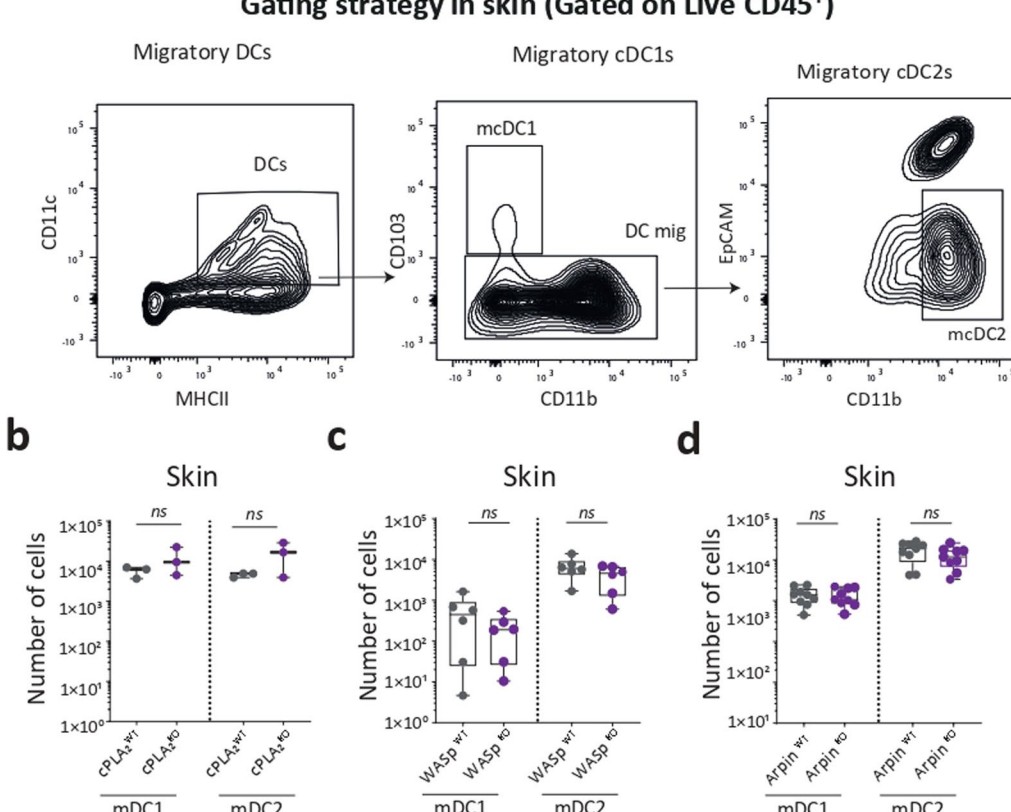

**Extended Data Fig. 4 | Gating strategy for DC quantification in the skin.**
(**a**) Gating strategy to quantify DCs in the skin of different mice at steady-state: after gating on live cells, immune cells were identified as CD45high; CD11c and MHCII were then used to differentiate lymph node-resident DCs (MHCIIlow, CD11chigh) from migratory DCs (MHCIIhigh, CD11chigh). Among migratory cDCs, mDC1s were identified as CD11bhigh, CD103low and mDC2s as CD11bhigh, EPCAMlow. (**b**) Box plots with min to max range in Log scale of the number of DCs in the skin of cPLA2WT and cPLA2KO mice, N = 2 independent experiments, n = 3 mice (where each dot is a mouse). Mann-Whitney *U*-test ns: non-significant. (**c**) Box plots with min to max range in Log scale of the number of migratory DCs in the skin of WASpWT and WASpKO mice, N = 2 independent experiments, n = 6 mice (where each dot is a mouse). Mann-Whitney *U*-test, ns: non-significant. (**d**) Box plots with min to max range in Log scale showing the number of migratory DCs in the skin of ArpinWT and KO mice, N = 3 independent experiments, n = 9 mice (where each dot is a mouse). Mann-Whitney *U*-test, ns: non-significant.

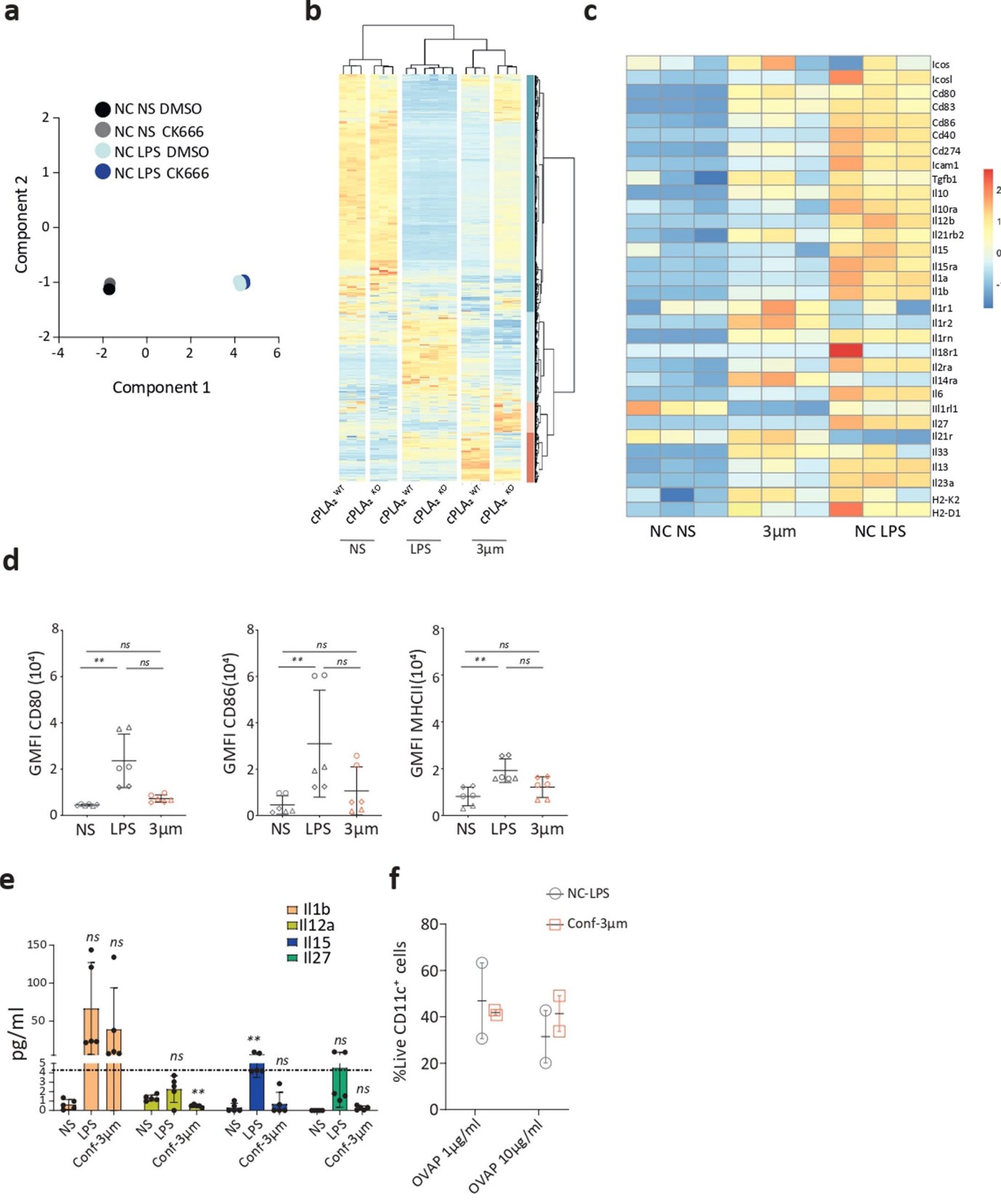

**Extended Data Fig. 5 | See next page for caption.**

**Extended Data Fig. 5 | Transcriptional changes in DC in response to confinement are cPLA2-dependent.** (**a**) Multidimensional scaling (MDS) of non-confined DCs stimulated or not with LPS and treated or not with CK666 (**b**) Heatmap of all the differentially expressed genes in cPLA$_2$ WT and KO cells in all the different conditions (**c**) Heat map of examples of cytokines and costimulatory and MHC-I genes in DCs in response to confinement or LPS (**d**) FACS analysis of some immune-activating genes expressed by DCs not-stimulated (NS) controls or in response to LPS or confinement at 3 μm height. Graphs of Mean with SD showing geometric mean intensity of CD80, CD86, and MHCII. N = 3 each condition was done in duplicates, Kruskal-Wallis test CD80 plot: ** p = 0.003, ns: not significant p = 0.1547. CD86 plot: **: p = 0.0083, ns: not significant p = 0.4684. MHCII plot: **: p = 0.0041, ns: not significant p = 0.2886 (**e**) cytokine secretion analysis by luminex of some cytokines in the supernatant of LPS or confined DCs 48h after the confinement or the LPS activation. Graph of Mean with SD of 5 independent experiments, each condition was done in triplicates. 2 way ANOVA test **: p = 0.0039, ns: not significant p = 0.1186, p = 0.3014, p = 0.3329, p = 0.7488, p = 0.1212, p = 0.0922 respectively(**f**) percentage of live DCs CD11c high after 48 hour of confinement and incubation with OTII T-cells. Graph: mean with SEM, n = 6, N = 2 independent experiments, each condition was done in duplicates.

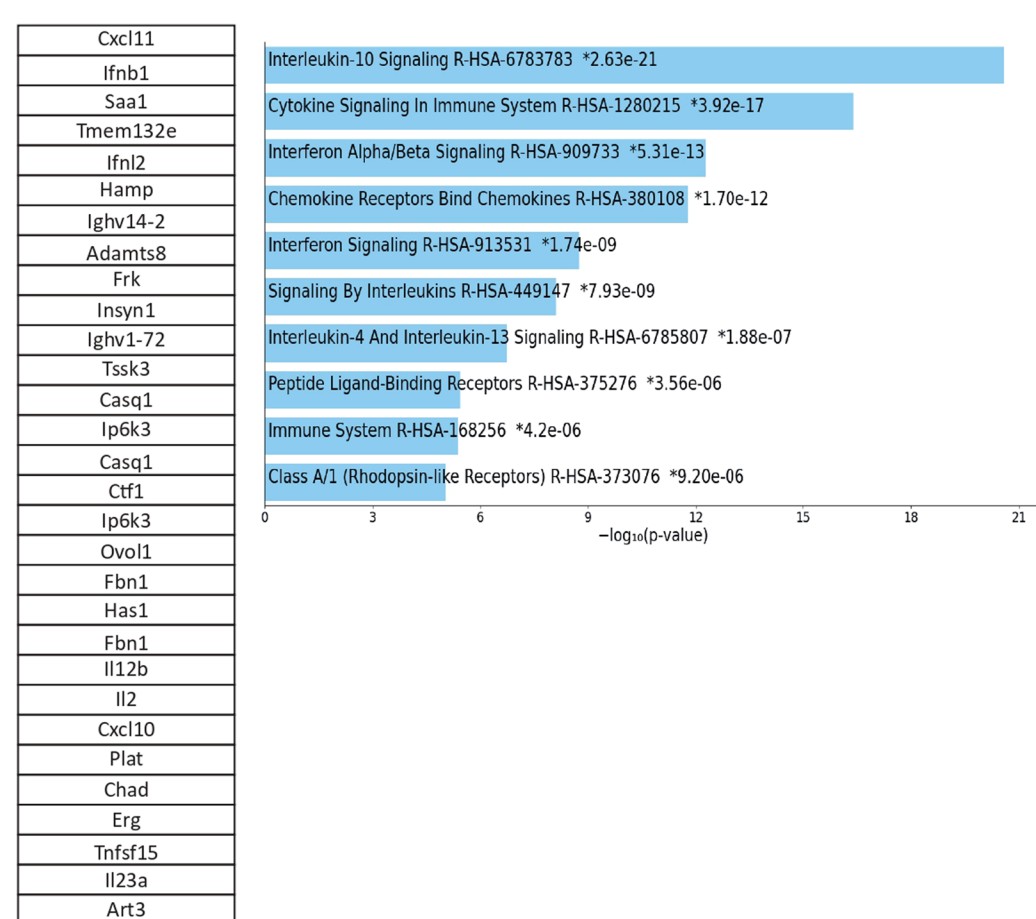

**Extended Data Fig. 6 | The Arp2/3 cPLA$_2$ pathway induces a tolerogenic signature in the confined DCs. (a, b)** statistical analysis used to determine genes with log FC > 1 and P value < 0.05 is detailed in the methods section. **(a)** Example of the top 30 genes that are upregulated in the confined DCs which their expression depends on cPLA$_2$ and Arp2/3 activity. **(b)** Reactom pathway analysis of the genes upregulated in DCs in response to shape sensing in an Arp2/3- cPLA$_2$- dependent manner.

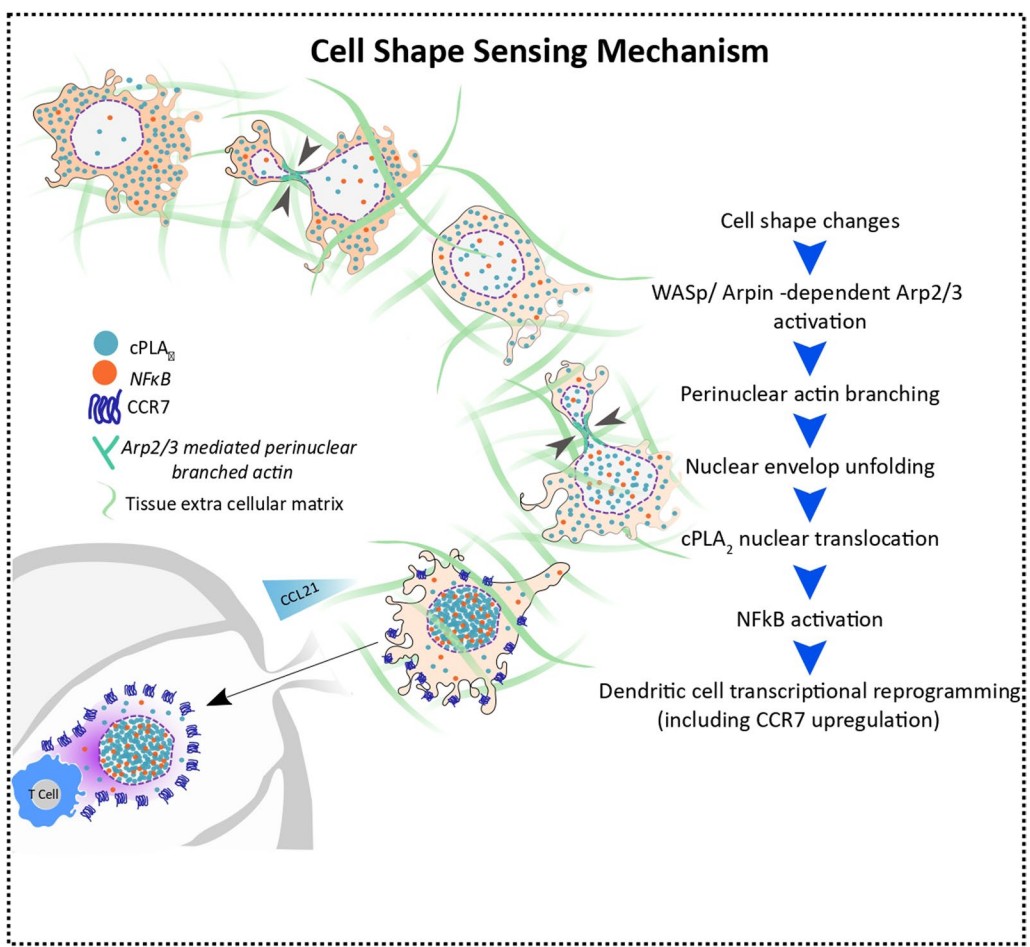

**Extended Data Fig. 7 | A model figure.** A model figure representing cell shape sensing pathway in patrolling dendritic cells, created with Inkscape.

| | |
|---|---|

# Reporting Summary

## Statistics

For all statistical analyses, confirm that the following items are present in the figure legend, table legend, main text, or Methods section.

| n/a | Confirmed | |
|---|---|---|
| ☐ | ☒ | The exact sample size (*n*) for each experimental group/condition, given as a discrete number and unit of measurement |
| ☐ | ☒ | A statement on whether measurements were taken from distinct samples or whether the same sample was measured repeatedly |
| ☐ | ☒ | The statistical test(s) used AND whether they are one- or two-sided<br>*Only common tests should be described solely by name; describe more complex techniques in the Methods section.* |
| ☒ | ☐ | A description of all covariates tested |
| ☐ | ☒ | A description of any assumptions or corrections, such as tests of normality and adjustment for multiple comparisons |
| ☐ | ☒ | A full description of the statistical parameters including central tendency (e.g. means) or other basic estimates (e.g. regression coefficient) AND variation (e.g. standard deviation) or associated estimates of uncertainty (e.g. confidence intervals) |
| ☐ | ☒ | For null hypothesis testing, the test statistic (e.g. *F*, *t*, *r*) with confidence intervals, effect sizes, degrees of freedom and *P* value noted<br>*Give P values as exact values whenever suitable.* |
| ☒ | ☐ | For Bayesian analysis, information on the choice of priors and Markov chain Monte Carlo settings |
| ☒ | ☐ | For hierarchical and complex designs, identification of the appropriate level for tests and full reporting of outcomes |
| ☒ | ☐ | Estimates of effect sizes (e.g. Cohen's *d*, Pearson's *r*), indicating how they were calculated |

*Our web collection on statistics for biologists contains articles on many of the points above.*

## Software and code

Policy information about availability of computer code

| | |
|---|---|
| Data collection | All microscopes were controlled by Meta Morphe software for data collection |
| Data analysis | Image analysis was performed using ImagJ software (version ImageJ2)., For image analysis custom made codes from ImageJ were used and they are available on Guithub (https://github.com/Zalraies/Alraies-at-al-2024).For FACS: data were analyzed using FlowJo software (version 10). All statistical analysis was done by Graph pad prism (version 8). For RNA seq data: Gene expression was estimated as described previously [PMID: 34495298] using Mouse FAST DB v2021_2 annotations, Pathway analyses and transcription factor network analysis were performed using WebGestalt 0.4.4 (92)[PMID: 31114916], Microarray data were annotated using the 'mogene10stprobeset.db' R package (v. 2.7). cPLA2 signature genes were displayed as heatmaps using tidyverse (v. 1.3.0), scales (v. 1.1.1), readxl (v. 1.3.1). Heat map, MDS, scatter and volcano plots were generated using R v4.2.1 with the help of pheatmap, ggplot2, and EnhancedVolcano [PMID:27207943] packages respectively. The heat maps were created using the Z-score of EdgeR normalized counts. |

For manuscripts utilizing custom algorithms or software that are central to the research but not yet described in published literature, software must be made available to editors and reviewers. We strongly encourage code deposition in a community repository (e.g. GitHub). See the Nature Portfolio guidelines for submitting code & software for further information.

## Data

Policy information about availability of data

All manuscripts must include a data availability statement. This statement should provide the following information, where applicable:

- Accession codes, unique identifiers, or web links for publicly available datasets
- A description of any restrictions on data availability
- For clinical datasets or third party data, please ensure that the statement adheres to our policy

> Raw RNA-seq data are available at GEO under the number: GSE207653: https://www.ncbi.nlm.nih.gov/geo/query/acc.cgi
> Raw data quantification used to generate the figures are available on FigShare: DOI: 10.6084/m9.figshare.25236715
> The GSE49358 microarray data (Tamoutounour et al., 2013) were downloaded from GE: https://www.ncbi.nlm.nih.gov/geo/query/acc.cgi?acc=GSE49358
> Confinement protocol is deposited on protocol exchange, DIO: https://doi.org/10.21203/rs.3.pex-2616/v1

## Research involving human participants, their data, or biological material

Policy information about studies with human participants or human data. See also policy information about sex, gender (identity/presentation), and sexual orientation and race, ethnicity and racism.

| | |
|---|---|
| Reporting on sex and gender | NA |
| Reporting on race, ethnicity, or other socially relevant groupings | NA |
| Population characteristics | NA |
| Recruitment | NA |
| Ethics oversight | NA |

Note that full information on the approval of the study protocol must also be provided in the manuscript.

# Field-specific reporting

Please select the one below that is the best fit for your research. If you are not sure, read the appropriate sections before making your selection.

☒ Life sciences  ☐ Behavioural & social sciences  ☐ Ecological, evolutionary & environmental sciences

For a reference copy of the document with all sections, see nature.com/documents/nr-reporting-summary-flat.pdf

# Life sciences study design

All studies must disclose on these points even when the disclosure is negative.

| | |
|---|---|
| Sample size | No statistical methods were used to pre-determine sample sizes but our sample sizes are similar to those reported in previous publications (Lomakin et al., Science 2020). |
| Data exclusions | No data were excluded |
| Replication | Statistical analysis was carried out using Prism 8 (GraphPad Software). Statistical analysis was conducted on data from three or more biologically independent experimental replicates. Data distribution was assumed to be normal (unless stated otherwise), but this was not formally tested. Statistical significance was calculated between two groups by Mann-Whitney test. Ordinary Mann-Whitney with multiple comparison was used to calculate statistical significance between multiple groups. Analyses were performed using GraphPad Prism 8 software. |
| Randomization | Experimental conditions were organized randomly, mice were chosen randomly for the experiments. |
| Blinding | Data collection and analysis were not performed blind to the conditions of the experiments |

# Behavioural & social sciences study design

All studies must disclose on these points even when the disclosure is negative.

| | |
|---|---|
| Study description | Briefly describe the study type including whether data are quantitative, qualitative, or mixed-methods (e.g. qualitative cross-sectional, |

| Study description | *quantitative experimental, mixed-methods case study).* |
|---|---|
| Research sample | *State the research sample (e.g. Harvard university undergraduates, villagers in rural India) and provide relevant demographic information (e.g. age, sex) and indicate whether the sample is representative. Provide a rationale for the study sample chosen. For studies involving existing datasets, please describe the dataset and source.* |
| Sampling strategy | *Describe the sampling procedure (e.g. random, snowball, stratified, convenience). Describe the statistical methods that were used to predetermine sample size OR if no sample-size calculation was performed, describe how sample sizes were chosen and provide a rationale for why these sample sizes are sufficient. For qualitative data, please indicate whether data saturation was considered, and what criteria were used to decide that no further sampling was needed.* |
| Data collection | *Provide details about the data collection procedure, including the instruments or devices used to record the data (e.g. pen and paper, computer, eye tracker, video or audio equipment) whether anyone was present besides the participant(s) and the researcher, and whether the researcher was blind to experimental condition and/or the study hypothesis during data collection.* |
| Timing | *Indicate the start and stop dates of data collection. If there is a gap between collection periods, state the dates for each sample cohort.* |
| Data exclusions | *If no data were excluded from the analyses, state so OR if data were excluded, provide the exact number of exclusions and the rationale behind them, indicating whether exclusion criteria were pre-established.* |
| Non-participation | *State how many participants dropped out/declined participation and the reason(s) given OR provide response rate OR state that no participants dropped out/declined participation.* |
| Randomization | *If participants were not allocated into experimental groups, state so OR describe how participants were allocated to groups, and if allocation was not random, describe how covariates were controlled.* |

# Ecological, evolutionary & environmental sciences study design

All studies must disclose on these points even when the disclosure is negative.

| Study description | *Briefly describe the study. For quantitative data include treatment factors and interactions, design structure (e.g. factorial, nested, hierarchical), nature and number of experimental units and replicates.* |
|---|---|
| Research sample | *Describe the research sample (e.g. a group of tagged Passer domesticus, all Stenocereus thurberi within Organ Pipe Cactus National Monument), and provide a rationale for the sample choice. When relevant, describe the organism taxa, source, sex, age range and any manipulations. State what population the sample is meant to represent when applicable. For studies involving existing datasets, describe the data and its source.* |
| Sampling strategy | *Note the sampling procedure. Describe the statistical methods that were used to predetermine sample size OR if no sample-size calculation was performed, describe how sample sizes were chosen and provide a rationale for why these sample sizes are sufficient.* |
| Data collection | *Describe the data collection procedure, including who recorded the data and how.* |
| Timing and spatial scale | *Indicate the start and stop dates of data collection, noting the frequency and periodicity of sampling and providing a rationale for these choices. If there is a gap between collection periods, state the dates for each sample cohort. Specify the spatial scale from which the data are taken* |
| Data exclusions | *If no data were excluded from the analyses, state so OR if data were excluded, describe the exclusions and the rationale behind them, indicating whether exclusion criteria were pre-established.* |
| Reproducibility | *Describe the measures taken to verify the reproducibility of experimental findings. For each experiment, note whether any attempts to repeat the experiment failed OR state that all attempts to repeat the experiment were successful.* |
| Randomization | *Describe how samples/organisms/participants were allocated into groups. If allocation was not random, describe how covariates were controlled. If this is not relevant to your study, explain why.* |
| Blinding | *Describe the extent of blinding used during data acquisition and analysis. If blinding was not possible, describe why OR explain why blinding was not relevant to your study.* |

Did the study involve field work? ☐ Yes ☒ No

# Field work, collection and transport

| Field conditions | *Describe the study conditions for field work, providing relevant parameters (e.g. temperature, rainfall).* |
|---|---|
| Location | *State the location of the sampling or experiment, providing relevant parameters (e.g. latitude and longitude, elevation, water depth).* |
| Access & import/export | *Describe the efforts you have made to access habitats and to collect and import/export your samples in a responsible manner and in* |

| Access & import/export | *compliance with local, national and international laws, noting any permits that were obtained (give the name of the issuing authority, the date of issue, and any identifying information).* |
|---|---|
| Disturbance | *Describe any disturbance caused by the study and how it was minimized.* |

# Reporting for specific materials, systems and methods

We require information from authors about some types of materials, experimental systems and methods used in many studies. Here, indicate whether each material, system or method listed is relevant to your study. If you are not sure if a list item applies to your research, read the appropriate section before selecting a response.

## Materials & experimental systems

| n/a | Involved in the study |
|---|---|
| ☐ | ☒ Antibodies |
| ☐ | ☒ Eukaryotic cell lines |
| ☒ | ☐ Palaeontology and archaeology |
| ☐ | ☒ Animals and other organisms |
| ☒ | ☐ Clinical data |
| ☒ | ☐ Dual use research of concern |
| ☒ | ☐ Plants |

## Methods

| n/a | Involved in the study |
|---|---|
| ☒ | ☐ ChIP-seq |
| ☐ | ☒ Flow cytometry |
| ☒ | ☐ MRI-based neuroimaging |

## Antibodies

| Antibodies used | For DC in LN FACS analysis: Anti-CCR7 (Biolegend #120114, clone 4B12, dilution: 1/50), anti-CD11c (eBioscience #25-0114-81, clone: N418, dilution: 1/800), anti-CD326 (Biolegend #118217, clone: G8.8, dilution: 1/800), anti-CD86 (BD pharmingen #553692, clone: GL1, dilution: 1/800), anti-CD11b (Biolegend #101237, clone: M1/70, dilution : 1/500), anti-MHCII (eBioscience #56-5321-80, clone: I-A/I-E - M5/114.15.2, dilution: 1/250), anti-CD45 (eBioscience #61-0451-82, clone: 30-F11, dilution: 1/100), anti-CD8a (BD bioscience #553035, clone: 53-6.7, dilution: 1/100), anti-CD103 (eBioscience #46-1031-80, clone: 2E7, dilution: 1/100). <br> For T cell FACS analysis: anti-CD4 (BD # 553051, clone RM4-5, dilution: 1/100), anti-CD69 (eBioscience # 48-0691-82, clone H1.2F3, dilution: 1/300), anti-TCR (BD # 553190, clone MR9-4, dilution: 1/300). <br> For DC FACS analysis: anti-CD80 (BD # 553769, clone 16-10A1, dilution: 1/100), anti-MHCII (Ozym # BLE107622, clone M5/114.15.2, dilution 1/400), anti-CD86 (Biolegend # 105037, clone GL-1, dilution: 1/200), anti-CD11c (BD # 550261, clone HL3, dilution: 1/200), Fc antibody (BD #553142, dilution 1/1000). <br> For IF: anti-CCR7 (abcam #ab32527, dilution : 1/100), anti-cPLA2 (cell signalling (Ozyme) #2832, dilution 1/100), anti-NF-κB P65 (cell signalling #mAB 8242, dilution 1/200), anti-Lap2 (BD Bioscience # 611000, dilution 1/200), Alexa Fluor-coupled Phalloidin (Invitrogen, dilution 1/300). <br> For secondary antibodies the following were used: Alexa 647 goat anti rabbit (Invitrogen, A21244, 1/300) and Alexa 488 goat anti mouse Fab2 (Invitrogen, A11017, 1/300). |
|---|---|
| Validation | All antibodies were validated on KO cells |

## Eukaryotic cell lines

Policy information about cell lines and Sex and Gender in Research

| Cell line source(s) | HeLa EMBL (Kyoto) were from Valerie Doyen, J588 cell lines were from S. Amigorena (Institut Curie). Both cell lines are originally commercially available on American Type Culture Collection ATCC. |
|---|---|
| Authentication | No authentication is done |
| Mycoplasma contamination | All cell lines tested negative for mycoplasma contamination |
| Commonly misidentified lines (See ICLAC register) | No misidentified cell lines were used in the study |

## Palaeontology and Archaeology

| Specimen provenance | *Provide provenance information for specimens and describe permits that were obtained for the work (including the name of the issuing authority, the date of issue, and any identifying information). Permits should encompass collection and, where applicable, export.* |
|---|---|
| Specimen deposition | *Indicate where the specimens have been deposited to permit free access by other researchers.* |
| Dating methods | *If new dates are provided, describe how they were obtained (e.g. collection, storage, sample pretreatment and measurement), where* |

| Dating methods | *they were obtained (i.e. lab name), the calibration program and the protocol for quality assurance OR state that no new dates are provided.* |

☐ Tick this box to confirm that the raw and calibrated dates are available in the paper or in Supplementary Information.

| Ethics oversight | *Identify the organization(s) that approved or provided guidance on the study protocol, OR state that no ethical approval or guidance was required and explain why not.* |

Note that full information on the approval of the study protocol must also be provided in the manuscript.

# Animals and other research organisms

Policy information about studies involving animals; ARRIVE guidelines recommended for reporting animal research, and Sex and Gender in Research

| Laboratory animals | For animal care, we strictly followed the European and French National Regulation for the Protection of Vertebrate Animals used for Experimental and other Scientific Purposes (Directive 2010/63; French Decree 2013-118, Authorization DAP number 43530-2023051216135493 v2 given by National Authority). Experiments were performed on 8 to 14 weeks-old male or female mice. Mice were maintained under specific pathogen-free conditions at the animal facility of Institut Curie, in accordance with institutional guidelines, housed in a 12h light:12h dark environment with free access to water (osmotic water) and food (MUCEDOLA - 4RF25SV Aliment Pellets 8*16mm irradiated 2,5 Mrad, and DietGel® Recovery, 2 oz / DietGel® Boost, 2 oz) . Littermates or age-mated mice were used as controls for all experiments involving knockout animals; breeder mice were previously backcrossed to C57BL6 for 7generations.C57BL6/J mice were obtained from Charles River, catalog #000664. CCR7-GFP mice were obtained from Jackson laboratory (stock# 027913), bred in our animal facility, original paper: Nakano H et al., 2013. CD11c-Cre mice: bred in our animal facility (Caton et al., 2007). Arpin and cPLA2 conditional knockout mice were generated by CIPHE (Centre d'Immunophénomique) Marseille, France. Both mice were generated using CRISPR cas-9 technique to generate cPLA2 and Arpin Flox mice that were crossed in our animal facility with Cd11c Cre mice.  Ikkb knockout mice were from Toby Lawernce laboratory described in Baratin et al., 2015. WASp KO mice on a C57BL/6 (CD45.2) genetic background were from Federica Benvenuti lab mice. Experiments were performed using homozygous WASp −/− as KO. Lmna fl/fl Vav1-Cre+/− (lamin A/C CK) mice were obtained from the lab of Nicolas Manel. YFP CD11c mice: From the lab of  S. Amigorena (original paper: Lindquist et al, 2004). |
| Wild animals | The study didn't involve wild animals |
| Reporting on sex | Males and females were used |
| Field-collected samples | No field collected samples were used in this study |
| Ethics oversight | For animal care, we strictly followed the European and French National Regulation for the Protection of Vertebrate Animals used for Experimental and other Scientific Purposes (Directive 2010/63; French Decree 2013-118, Authorization DAP number 43530-2023051216135493 v2 given by National Authority). |

Note that full information on the approval of the study protocol must also be provided in the manuscript.

# Clinical data

Policy information about clinical studies
All manuscripts should comply with the ICMJE guidelines for publication of clinical research and a completed CONSORT checklist must be included with all submissions.

| Clinical trial registration | *Provide the trial registration number from ClinicalTrials.gov or an equivalent agency.* |
| Study protocol | *Note where the full trial protocol can be accessed OR if not available, explain why.* |
| Data collection | *Describe the settings and locales of data collection, noting the time periods of recruitment and data collection.* |
| Outcomes | *Describe how you pre-defined primary and secondary outcome measures and how you assessed these measures.* |

# Dual use research of concern

Policy information about dual use research of concern

## Hazards

Could the accidental, deliberate or reckless misuse of agents or technologies generated in the work, or the application of information presented in the manuscript, pose a threat to:

| No | Yes | |
|----|-----|---|
| ☒ | ☐ | Public health |
| ☒ | ☐ | National security |
| ☒ | ☐ | Crops and/or livestock |
| ☒ | ☐ | Ecosystems |
| ☒ | ☐ | Any other significant area |

## Experiments of concern

Does the work involve any of these experiments of concern:

| No | Yes | |
|----|-----|---|
| ☒ | ☐ | Demonstrate how to render a vaccine ineffective |
| ☒ | ☐ | Confer resistance to therapeutically useful antibiotics or antiviral agents |
| ☒ | ☐ | Enhance the virulence of a pathogen or render a nonpathogen virulent |
| ☒ | ☐ | Increase transmissibility of a pathogen |
| ☒ | ☐ | Alter the host range of a pathogen |
| ☒ | ☐ | Enable evasion of diagnostic/detection modalities |
| ☒ | ☐ | Enable the weaponization of a biological agent or toxin |
| ☒ | ☐ | Any other potentially harmful combination of experiments and agents |

# Plants

| | |
|---|---|
| Seed stocks | *Report on the source of all seed stocks or other plant material used. If applicable, state the seed stock centre and catalogue number. If plant specimens were collected from the field, describe the collection location, date and sampling procedures.* |
| Novel plant genotypes | *Describe the methods by which all novel plant genotypes were produced. This includes those generated by transgenic approaches, gene editing, chemical/radiation-based mutagenesis and hybridization. For transgenic lines, describe the transformation method, the number of independent lines analyzed and the generation upon which experiments were performed. For gene-edited lines, describe the editor used, the endogenous sequence targeted for editing, the targeting guide RNA sequence (if applicable) and how the editor was applied.* |
| Authentication | *Describe any authentication procedures for each seed stock used or novel genotype generated. Describe any experiments used to assess the effect of a mutation and, where applicable, how potential secondary effects (e.g. second site T-DNA insertions, mosiacism, off-target gene editing) were examined.* |

# Flow Cytometry

## Plots

Confirm that:

☒ The axis labels state the marker and fluorochrome used (e.g. CD4-FITC).

☒ The axis scales are clearly visible. Include numbers along axes only for bottom left plot of group (a 'group' is an analysis of identical markers).

☒ All plots are contour plots with outliers or pseudocolor plots.

☒ A numerical value for number of cells or percentage (with statistics) is provided.

## Methodology

| | |
|---|---|
| Sample preparation | cells were recovered directly by gently washing with medium.Cells were re-suspended in the buffer (PBS BSA 1% EDTA 2mM). After blocking with Fc antibody (BD #553142) and live/dead staining kit (Thermo # L34966) for 15 min, cells were stained with the desired antibodies for 20 min Cells were then washed three times and re-suspended in the staining buffer |
| Instrument | Flow cytometry was performed on LSRII (BD) |
| Software | FlowJo software version 10 |
| Cell population abundance | cells of interest were highly abudant in the culture and were defined by a gating strategy and checked for vaiability using live dead staining whenever applicable |
| Gating strategy | Gating strategy to quantify DCs in skin-draining lymph nodes and skin of mice at steady-state: after gating on live cells, immune cells were identified as CD45high; CD11c and MHCII were then used to differentiate lymph node-resident DCs |

(MHCII low, CD11chigh) from migratory DCs (MHCIIhigh, CD11chigh). Among migratory cDCs, mDC1s were identified as CD11bhigh, CD103low and mDC2s as CD11bhigh, EPCAMlow.

☒ Tick this box to confirm that a figure exemplifying the gating strategy is provided in the Supplementary Information.

