## [Peer Review File · Nature Immunology]

Peer Review Information

Journal: Nature Immunology

Manuscript Title: Cell Shape Sensing Licenses Dendritic Cells for Homeostatic Migration to Lymph Nodes

Corresponding author name(s): Dr Ana-Maria Lennon-Dumenil; Dr Matthieu Piel

Reviewer Comments & Decisions:

Decision Letter, initial version:
--

10th May 2023

Dear Dr. Lennon-Dumenil,

We have now finished reviewing your manuscript entitled "A Shape Sensing Mechanism driven by Arp2/3 and cPLA2 licenses Dendritic Cells for Migration to Lymph Nodes in Homeostasis", reference number NI-A35561-T.

Although the editors thought that the manuscript was interesting enough to send out for in-depth review, the reviewers were not completely in favor of publishing the paper in Nature Immunology.

As you will see, reviewer 1 really likes the paper but reviewer 2 points to some serious deficits that sorting would be requisite for publication in our journal. These two reviewers are expert in cell migration, biophysics and mechanosensing. We also had a third reviewer who is more of an immunologist and expert in dendritic cells and their migration. Unfortunately, this reviewer did not supply a report or respond to some 5 queries I sent them. Normally I would want to replace this reviewer but given the existing negative report we have from reviewer 2 we decided to not waste more of your time and instead make a decision now. We do appreciate that you might think you can revise the paper according to the present reviews, and if you are able to do so you would be welcome to appeal our decision down the line. However, please note that any successful appeal revised manuscript would still need to be seen by a dendritic cell expert, which might result in yet further criticisms and revision expectations, a point which has further contributed to our decision here. We are very sorry that this peer review has not gone well and apologize for the missing dendritic cell expert report, but we cannot strong this decision along any further as it is not in anybody's interests to do so. As noted, if you wish to appeal the decision with a major revision we will be happy to take another look.

We realize that this is disappointing. I hope that you continue to consider Nature Immunology for your

results most significant for the immunology community and wish you well in your future investigations.

Sincerely,

Nick Bernard, PhD
Senior Editor
Nature Immunology

Reviewers' comments:

Reviewer #1 (Remarks to the Author):

Alraies et al. describe the involvement of the recently characterized "nuclear ruler" pathway in homeostatic lymph node migration of DCs. They propose that "cellular massage" of DCs in the periphery leads to activation of NFkB and CCR7 upregulation. This mechanical stimulation pathway depends on cPLA2 activity whereas LPS induced DC stimulation does not. Thus, nuclear confinement of peripheral DCs could lead to increased DC/autoantigen trafficking to the lymph node antagonizing autoimmunity under homeostatic conditions.

This is a fascinating paper that is, as far as I can judge, technically well done. It contains orthogonal experimental approaches to validate key points, is nicely quantitative and well written. Most of all, the concept that non-chemical cues could modulate auto-immunity is new and intriguing. Some of the phenotypes of cPLA2 KO animals should perhaps be revisited now, taking the possibility of auto-immune effects into account. With some minor modifications, I think the manuscript would be suited for publication in Nature Immunology.

Comments:

-To gain more information about how cPLA2 might affect the transcription of CCR7, it would be nice if they authors could distinguish whether the metabolic products of the enzyme (e.g., arachidonic acid, prostaglandins, leukotrienes, etc.) can induce CCR7 in the absence of functional cPLA2 protein. From the current data, it is not entirely obvious whether cPLA2's enzymatic activity is required for CCR7 regulation or some other function of the protein. cPLA2 has been, for example, reported to interact with a splice variant of TIP60 (PLIP). PLIP has been shown to modulate cPLA2's nuclear localization (e.g., PMID: 11416127) and TIP60 is known to affect a range of transcriptional responses, incl. NFkB. The authors could try to reconstitute CCR7 expression in cPLA2 KO cells by exposing them to AA, PGE2, 5(S) HPETE/5(S)HETE, etc. and check whether pharmacologic inhibitors of cPLA2 mimic the genetic KO effects. Or directly check whether CCR7 expression is PLIP dependent.

-As evidence of nuclear translocation of cPLa2, nuclear fluorescence is currently taken as a proxy. Is it possible to measure the cytoplasmic/nucleoplasmic cPLA2 ratio instead? This could give clearer evidence for a confinement-induced translocation mechanism.

-Discussion/2nd paragraph: "Of note, membrane tension is more likely..." I found this sentence confusing. The inner nuclear membrane is also continuous with the outer nuclear membrane just like the outer membrane is continuous with the ER. So, membrane continuity alone, at least for me, does

not explain why the inner nuclear membrane should get more easily stretched than the outer nuclear membrane. Maybe reword?

Reviewer #2 (Remarks to the Author):

In the manuscript by Alraies et al, microengineered PDMS confiners are used to study the migration and transcriptional adaptation of immature dendritic cells in response to mechanical deformation. Based on their previous work, the authors hypothesized that the nucleus acts as a shape sensor to measure cellular confinement. Along these lines, they confirmed their previous findings that stretching of the nuclear envelope triggers mechanosensitive recruitment of cPLA2 to the nucleus and promotes cellular motility. In addition to previously published data, the authors now show that CCR7 is upregulated in response to confinement in a cPLA2-dependent manner. Interestingly, they show that environmental constriction alone is not sufficient, but that the elicited perinuclear Arp2/3-dependent actin polymerization is required for the cellular response to confinement. Unfortunately, they do not provide a mechanistic explanation for this very interesting observation. The critical, innovative parts of the paper are Figures 4-6, which purport to show that shape sensing by the Arp2/3-cPLA2-CCR7 axis plays a critical role in the homeostatic migration of dendritic cells to draining lymph nodes, with implications for their tolerogenic function. They claim that cellular deformations as a result of surveillance of dense tissue induce transcriptional reprogramming that ultimately determines the fate of cells migrating to lymph nodes. The mechanisms of steady-state DC migration are not well understood and the mechanistic model proposed in the manuscript is highly innovative. However, as presented, the data do not support it.

1. The authors claim that (peri)nuclear Arp2/3 activity is required for confinement-mediated cPLA2 recruitment to the nucleus. However, the spatio-temporal dynamics of the actin cloud and its precise characterization with respect to the nucleus remain unclear. Does the actin cloud shown in Figure 3a deform the nuclear envelope, possibly affecting its permeability to cPLA2? Alternatively, does intra- rather than perinuclear actin mediate cPLA2 entry?
2. The authors do not provide direct evidence for perinuclear actin-dependent transcriptional reprogramming in confinement. Does Arp2/3 inhibition and WASp-deficiency block transcriptional reprogramming in confined DCs?
3. A key message of the work is that the identified shape sensing mechanism is crucial for the homeostatic migration of DCs to the lymph node. However, the data shown in Fig. 5 do not provide any definitive evidence for the proposed mechanisms of shape sensing and the associated upregulation of CCR7 by mechanical constraints in vivo, as the corresponding knockouts have far-reaching consequences for cellular motility. For example, it is known that WASp-deficient cells migrate to the lymph node at a slower rate despite a CCR7 expression comparable to WT (<https://pubmed.ncbi.nlm.nih.gov/15494425/>).
4. The authors show convincingly that the transcriptional profile of mechanically stimulated DCs differs from that of DCs stimulated with LPS and that these cells are less efficient in activating T cells. Based on these data, they hypothesize that mechanical stimulation is responsible for immunoregulatory properties of tissue-patrolling DCs in the steady state which allows them to transport self-antigen to the lymph node in a tolerogenic state. Unfortunately, no functional in vivo data are shown to support

this innovative hypothesis.

Minor points:

1. The intravital microscopic images are of poor quality and do not allow any conclusions to be drawn about the morphology of the DCs and their nuclei. What is the physiological range of nucleus confinement in vivo?
2. The cell morphology of the representative images is very heterogeneous. For example, the WASp-KO cell in Figure 3e is very small and round and shows no protrusions typical of DCs. Are these cells still alive.
3. Please provide more than 2 biological replicates. E.g. in Fig. 1g; 3f,h

Author Rebuttal to Initial comments

See inserted PDF

General comment from the authors:

We thank the referees for their valuable comments on our study and their thoughtful suggestions, which greatly improved our manuscript. We have addressed each of their points, as outlined below, incorporating new experiments and analyses. Main revisions include: (i) a detailed study of the mechanistic link between Arp2/3 branched actin cytoskeleton and cPLA₂ activation and of its impact on transcriptional reprogramming of dendritic cells (DCs); several new figures are dedicated to this point raised by ref.#2 (fig. 3a,b, entire fig. 4, fig. 6e, f, Supplementary fig.5 a, Supplementary movie 3), (ii) a clarification of the role of cPLA₂ and the downstream products of the enzyme in CCR7 upregulation (fig. 7c, point raised by ref.#1), and (iii) a more accurate analysis and precise quantifications of the events of DC deformation *in vivo* to better link these events to our *ex vivo* experiments (fig. 1a, Supplementary movie 1 point raised ref.#2).

In addition, we have corrected and clarified our manuscript in accordance with both referees' recommendations, and updated our abstract, introduction and discussion based on these changes and recent literature advances. **All manuscript changes are highlighted in yellow.**

We hope that referees will find our revised manuscript now well-suited for publication in *Nature Immunology*.

Reviewer #1 (Remarks to the Author):

Alraies et al. describe the involvement of the recently characterized “nuclear ruler” pathway in homeostatic lymph node migration of DCs. They propose that “cellular massage” of DCs in the periphery leads to activation of NFκB and CCR7 upregulation. This mechanical stimulation pathway depends on cPLA₂ activity whereas LPS induced DC stimulation does not. Thus, nuclear confinement of peripheral DCs could lead to increased DC/autoantigen trafficking to the lymph node antagonizing autoimmunity under homeostatic conditions. This is a fascinating paper that is, as far as I can judge, technically well done. It contains orthogonal experimental approaches to validate key points, is nicely quantitative and well written. Most of all, the concept that non-chemical cues could modulate auto-immunity is new and intriguing. Some of the phenotypes of cPLA₂ KO animals should perhaps be revisited now, taking the possibility of auto-immune effects into account. With some minor modifications, I think the manuscript would be suited for publication in Nature Immunology.

Authors: we thank the referee for providing a positive assessment of our work and for her/his thoughtful comments. We are hopeful that our research will indeed contribute emphasizing the role of cell shape sensing and physical properties of tissues at the forefront of the immunology field and will open new opportunities for therapeutics intervention and drug discovery not only in the context of autoimmunity but as well in cancer, as CCR7+ DCs have recently been highlighted as playing key roles in tumor immunity (Zilionis et al. Immunity 2019, Maier et al. Nature 2020, Di Pilato et al. Cell 2021).

Comments:

-To gain more information about how cPLA₂ might affect the transcription of CCR7, it would be nice if they authors could distinguish whether the metabolic products of the enzyme (e.g., arachidonic acid, prostaglandins, leukotrienes, etc.) can induce CCR7 in the absence of functional cPLA₂ protein. From

the current data, it is not entirely obvious whether cPLA₂'s enzymatic activity is required for CCR7 regulation or some other function of the protein. cPLA₂ has been, for example, reported to interact with a splice variant of TIP60 (PLIP). PLIP has been shown to modulate cPLA₂'s nuclear localization (e.g., PMID: 11416127) and TIP60 is known to affect a range of transcriptional responses, incl. NFκB. The authors could try to reconstitute CCR7 expression in cPLA₂ KO cells by exposing them to AA, PGE₂, 5(S) HPETE/5(S)HETE, etc. and check whether pharmacologic inhibitors of cPLA₂ mimic the genetic KO effects. Or directly check whether CCR7 expression is PLIP dependent.

Authors: We thank the referee for bringing this point, which is indeed very important. We now have incorporated several new experiments to address it, as described below.

1- We pharmacologically inhibited cPLA₂ with AACOCF₃, which phenocopied the phenotype of cPLA₂ mRNA knock down and gene knock out in terms of CCR7 upregulation by confinement at 3 μm-height (fig. 2a, page 6). This result shows that the enzymatic activity of cPLA₂ is indeed required for the here-described response of DCs to cell shape changes.

2- We investigated the role of Prostaglandin E₂ (PGE₂, a cPLA₂ end-product) in this response. Indeed, it had already been shown that its precursor Arachidonic Acid (AA) renders DCs tolerogenic (Kumar et al. Plos One 2014 <https://doi.org/10.1371/journal.pone.0111759>), so we chose to focus on downstream products, which correspond to either PGEs (Prostaglandins) or LTs (Leukotriens). We decided to focus on PGEs rather than LTs as we found that several mRNAs encoding for enzymes involved in PGE synthesis and PGE receptor 2 were more abundant in confined DCs, suggesting that this pathway is specifically upregulated in these cells. Accordingly, we observed that addition of PGE₂ restored CCR7 expression in confined cPLA₂ knock out cells (fig. 7c, page 12), highlighting the role of this cPLA₂ downstream product in sensing of shape changes by DCs.

3- Our RNAseq analysis revealed that the gene coding for TIP60 (*Kat5*) was expressed at similar levels in cPLA₂^{WT} and cPLA₂^{KO} DCs. As suggested by the referee, we thus used an inhibitor of TIP to investigate its role in CCR7 upregulation upon DC confinement. Unfortunately, this drug was very toxic on DCs even at low concentrations and incubation times (TIP inhibitor Abcam: ab255734, 20, 30, 35 μM incubated either 2 hours or 20 minutes before confinement, more than 80% of DCs were dead after 4 hours of confinement with the drug). We were therefore not able to add these experiments in our manuscript.

-As evidence of nuclear translocation of cPLA₂, nuclear fluorescence is currently taken as a proxy. Is it possible to measure the cytoplasmic/nucleoplasmic cPLA₂ ratio instead? This could give clearer evidence for a confinement-induced translocation mechanism.

Authors: We thank the referee for this very relevant suggestion. As predicted by the referee, this analysis confirmed confinement-induced nuclear translocation of cPLA₂ (fig. 2f).

-Discussion/2nd paragraph: "Of note, membrane tension is more likely..." I found this sentence confusing. The inner nuclear membrane is also continuous with the outer nuclear membrane just like the outer membrane is continuous with the ER. So, membrane continuity alone, at least for me, does not explain why the inner nuclear membrane should get more easily stretched than the outer nuclear membrane.

Authors: We apologize for this potentially confusing sentence in the discussion. Membrane tension is in general a difficult point to address in the complex context of cellular membranes. Based on what is

known, mostly for the plasma membrane, membrane tension is actively generated due to friction of lipids with membrane bound and transmembrane proteins on which some forces are applied from various structures such as the actin cytoskeleton. This is because there is always a large reservoir of membrane (this is well quantified for the plasma membrane), and tension never comes from a lack of reservoir but from the amount of friction that opposes lipid movements. The question of nuclear envelope tension is rendered more complex due to the presence of two nuclear membranes. Similar to the plasma membrane, tension in the outer and/or inner nuclear membrane will depend on the amount of friction with other structures, preventing the membrane to flow freely from the large ER reservoirs – having that in mind, it is usually assumed that the inner nuclear membrane is more likely to get more tensed than the outer, because of a larger friction at the level of nuclear pores, although there is so far no available studies on this question. We think that this discussion goes too far relative to the scope of the present article as there is no strong evidence for this phenomenon and it has no direct implication for our findings, it is rather important for future studies getting deeper into the mechanism of cPLA₂ activation. We thus modified this part of the discussion and removed the notion of inner versus outer membrane tension.

Reviewer #2 (Remarks to the Author):

In the manuscript by Alraies et al, microengineered PDMS confiners are used to study the migration and transcriptional adaptation of immature dendritic cells in response to mechanical deformation. Based on their previous work, the authors hypothesized that the nucleus acts as a shape sensor to measure cellular confinement. Along these lines, they confirmed their previous findings that stretching of the nuclear envelope triggers mechanosensitive recruitment of cPLA₂ to the nucleus and promotes cellular motility. In addition to previously published data, the authors now show that CCR7 is upregulated in response to confinement in a cPLA₂-dependent manner. Interestingly, they show that environmental constriction alone is not sufficient, but that the elicited perinuclear Arp2/3-dependent actin polymerization is required for the cellular response to confinement. Unfortunately, they do not provide a mechanistic explanation for this very interesting observation.

Authors: We thank the referee for her/his positive global assessment of our work and for pointing to the importance of the mechanism through which Arp2/3 allows shape sensing by DCs. In response to this insightful criticism, we have performed different sets of new experiments to provide such mechanism (see below).

The critical, innovative parts of the paper are Figures 4-6, which purport to show that shape sensing by the Arp2/3-cPLA₂-CCR7 axis plays a critical role in the homeostatic migration of dendritic cells to draining lymph nodes, with implications for their tolerogenic function. They claim that cellular deformations as a result of surveillance of dense tissue induce transcriptional reprogramming that ultimately determines the fate of cells migrating to lymph nodes. The mechanisms of steady-state DC migration are not well understood, and the mechanistic model proposed in the manuscript is highly innovative. However, as presented, the data do not support it.

Authors: we respectfully disagree with the referee on figures 1-3 not representing an innovation as neither the impact of cell shape changes on the expression of a chemokine receptor such as CCR7 (fig. 1), nor its dependency on cPLA₂ (fig. 2), nor the requirement of Arp2/3 for cPLA₂ activation (fig. 3) had ever been highlighted before.

Yet, to better link the cell deformation events observed in the dense environment of tissues to our *ex vivo* confinement experiments, we now provide new imaging experiments to precisely quantify the cell shape changes that DCs undergo while patrolling the ear skin. We found that dermal DCs spend approximately a third of their time with their cell body constricted to minimal diameters ranging between 2 and 4 μm (fig. 1a, page 5), further validating the confinement heights used in our experiments.

These new results combined to our results showing that homeostatic DC migration to skin-draining lymph nodes is indeed decreased in cPLA₂ or WASp KO mice (both being required for shape sensing), but increased in ARPIN KO mice, which exhibit enhanced sensitivity to mechanical deformation, strongly suggest that the here-described shape-sensing pathway can drive homeostatic cDC2 migration (fig. 5). This conclusion is further supported by the strong similarities observed when comparing the transcriptomic profiles of migratory cDC2s purified from skin-draining LNs at steady-state and the one induced *ex vivo* in DCs that have activated the Arp2/3-cPLA₂ pathway in response to confinement (fig. 6g).

Altogether, these results highlight that this mechanical axis is very likely to contribute to steady-state DC migration *in vivo*. The text has been modified all along the manuscript for this message to appear more clearly.

1. The authors claim that (peri)nuclear Arp2/3 activity is required for confinement-mediated cPLA2 recruitment to the nucleus. However, the spatio-temporal dynamics of the actin cloud and its precise characterization with respect to the nucleus remain unclear. Does the actin cloud shown in Figure 3a deform the nuclear envelope, possibly affecting its permeability to cPLA2? Alternatively, does intra- rather than perinuclear actin mediate cPLA2 entry?

Authors: We thank the reviewer for pointing to this interesting aspect of the molecular mechanism. We provide a new set of experiments to address this critical referee's comment, as described below.

- We performed better resolved live-imaging experiments of LifeAct-DCs under confinement with or without the Arp2/3 inhibitor CK666. These data non-ambiguously show that a central actin pool appears in DCs confined at 3 but not 4 μm -height as soon as confinement is applied (fig.3a, b, Supplementary figS.3a) and is usually lost upon removal of the confinement. Orthogonal projections showed that this pool is cortical (part of the actin pool that forms below the plasma membrane). Its positioning with respect of the nucleus can vary, being detected either at the top or on the side of this organelle. We never observed this actin pool inside of the nucleus, whether we used LifeAct-GFP expressing mice or phalloidin staining on DCs fixed under confinement. Importantly, we found that the confinement induced cortical actin pool was lost when treating the cells with the Arp2/3 inhibitor CK666, suggesting that its nucleation requires Arp2/3 (fig. 3a, b Supplementary figS.3a and Supplementary movie S4, pages 7,8). These results confirm the cortical positioning of this Arp2/3-dependent perinuclear branched actin pool.

- We developed a method to fix DCs without removing confinement. This helped us observing that the perinuclear actin pool is lost when confinement is released. This method was then used to image both perinuclear actin and the nuclear envelope stained with the inner nuclear envelope marker LAP2 and treated or not with the Arp2/3 inhibitor CK666. These experiments show that DCs confined at 3 μm -height exhibit a smoother nuclear envelope than DCs confined at 4 μm -height or treated with CK666, which often displayed a folded nuclear envelope that exhibits wrinkles (fig. 4a, page 8). These results

are consistent with the Arp2/3-dependent perinuclear actin pool being required for nuclear envelop unfolding and increase in tension, which was previously shown to lead to cPLA₂ activation (Enyedi et al. Cell 2016). We hypothesized that the cortical actin accumulation we observed at the cell center could be due to a centripetal cortical actin flow, as observed before in various contexts (Yolland et al. Nat Cell Biol. 2019). Such a flow could be responsible for pulling on the nuclear envelop and unfolding it (such an effect of retrograde flow of actin filaments exerting forces on the nucleus has also been proposed before (see Gomes et al. Cell 2005). We have now discussed these points in the article (page 14) to propose a mechanism by which actin unfolds the nuclear envelop (describing this mechanism in further details is of great interest but out of the scope of the present article).

- We therefore attempted to measure nuclear envelop tension in confined DCs using a membrane tension sensor targeted to the ER membrane, ER Flipper-TR, detected by Fluorescence Lifetime Imaging (FLIM) (Goujon et al. J Am Chem Soc. 2019, Colom et al. Nat Chem. 2018). Unfortunately, this could not be done as DCs did not accumulate this sensor intracellularly. We thus turned to HeLa cells, which have also been previously shown to activate cPLA₂ in response to nuclear envelope stretching and can successfully accumulate the tension probe (Lomakin et al. Science 2020). These experiments showed that treatment of HeLa cells with CK666 indeed decreased the tension of their nuclear envelop (fig. 4d and page 9), consistent with branched actin being important to build up nuclear membrane tension, similar to what was shown for the plasma membrane (Venkova et al. elife 2022, Diz-Muñoz A et al. PLOS biology 2010).

- To provide a formal proof for this model, we analyzed the response to confinement of Lamin (Lmn) A/C KO DCs. Indeed, it has been shown by us and others that cells cannot build nuclear envelop tension in response to confinement when they lack this key nucleoskeleton component (Lomakin et al. Science 2020). We found that Lmn A/C KO DCs do not activate cPLA₂ nor upregulate CCR7 expression when confined at 3 μm-height (fig. 4e, f and page 9) confirming that this response of DCs does indeed depend on nuclear envelop tension. This conclusion is further supported by our data showing that confinement of DCs at 2 μm-height, which leads to nuclear envelop rupture (fig. 2h) and thus probably to a decrease in tension, prevents cPLA₂ activation and upregulation of CCR7 expression in response to shape sensing.

Altogether, these new results allowed us building a model where confinement triggers the formation of an Arp2/3-dependent perinuclear branched actin pool, which helps unfolding and tensing up the nuclear envelop for activation of cPLA₂ and downstream upregulation of CCR7. An entire figure (new Figure 4) is now dedicated to these findings.

Of note, as mentioned in our discussion (page 14), we do not exclude that the increment of nuclear envelop tension through Arp2/3 might also help the opening of nuclear pores to increase cPLA₂ nuclear translocation. Unfortunately, we failed assessing this point experimentally as the drug that blocks active import was extremely toxic to DCs (Importazole, Selleckchem S8446 tested at 30 and 50 μM for 20 minutes or just before confinement, more than 90% of the cells were dead by the end of the experiment).

2. The authors do not provide direct evidence for perinuclear actin-dependent transcriptional reprogramming in confinement. Does Arp2/3 inhibition and WASp-deficiency block transcriptional reprogramming in confined DCs?

Authors: We thank the referee for pinning down this aspect. To address this question, we conducted a new bulk RNAseq experiment using the inhibitor of Arp2/3 CK666 on cPLA₂ WT and KO cells confined or not. We chose to focus this experiment on the effector Arp2/3 rather than upstream nucleation promoting factor WASp as this would have required generating double cPLA₂-WASp KO mice, delaying the submission of our revised manuscript of at least nine more months.

Bulk RNAseq analysis showed that inhibition of Arp2/3 activity compromised the upregulation of all cPLA₂-dependent genes in response to confinement at 3 μm-height. Among these genes are the 103 genes that behave as CCR7 and include *Ikkβ*, as well as the two subunits of the Arp2/3. In contrast, Arp2/3 inhibition had a very minor effect on the transcriptional reprogramming of non-confined or LPS-treated DCs, as observed for cPLA₂ deficiency (fig. 6e, f, Supplementary fig. S5a, S6 and pages 11,13). Interestingly, we observed that while the genes whose expression was modified in response to confinement in a cPLA₂-dependent manner also required Arp2/3 activity, the opposite was not true: few hundred genes were upregulated in response to confinement in an Arp2/3-dependent but cPLA₂-independent manner. These results confirm that Arp2/3 activity is needed for cPLA₂-dependent reprogramming of DCs in response to shape sensing and further suggest that this actin nucleation complex also controls additional transcriptional changes in confined DCs. Of note, as our manuscript focus on the cPLA₂ shape-sensing pathway that controls CCR7 expression, we decided not to comment further on these genes to limit manuscript length.

3. A key message of the work is that the identified shape sensing mechanism is crucial for the homeostatic migration of DCs to the lymph node. However, the data shown in Fig. 5 do not provide any definitive evidence for the proposed mechanisms of shape sensing and the associated upregulation of CCR7 by mechanical constraints *in vivo*, as the corresponding knockouts have far-reaching consequences for cellular motility. For example, it is known that WASp-deficient cells migrate to the lymph node at a slower rate despite a CCR7 expression comparable to WT (<https://pubmed.ncbi.nlm.nih.gov/15494425/>).

Authors: thank you to the referee for raising this discussion point that we had not included. Indeed, a recent study to which we participated showed that in confined spaces such as microchannels or pillar forests, WASp deficiency led to an increased, rather than decreased motility of DCs (Oliveira et al. *Leukocyte Biology*, 2022, doi: 10.1002/JLB.1AB0821-013RR). In addition, *in vivo* experiments performed in this study using the Aldara inflammatory agent to induce DC maturation, showed that WASp KO DCs reached LNs in similar numbers in response to inflammation, although they displayed a distinct sub-tissular localization within LNs as compared to their WT counterpart. Similarly, we have characterized, as shown below (fig. 1), the migration of Arpin WT and KO DCs in microchannels with constrictions of 1 and 2 μm, and in collagen gels with a CCL21 gradient. We did not find any significant difference in the migration velocity of Arpin WT and KO cells in the channels and the speed of both phenotypes was comparable in collagen gels, indicating that ARPIN does not control DC motility *per se*. We did not include these data in the manuscript but could do so if the referee thinks this is necessary.

Nonetheless, having read the referee's comment, we understand that in the case of WASp, there are conflicting results in the literature, as the article she/he cites describes a defect of WASp KO DCs in migration to LNs in response to FITC sensitization procedure. Of note, this article does not show that the defect in WASp KO DCs migration to LNs is due to a defect in their motility *per se*, as this *in vivo* phenotype could also result, for example, from impaired recognition of CCL19/21 gradients or

retention within lymphatic vessels. To our knowledge, there is no result in the literature showing that leukocytes deficient for WASp or Arp2/3 have intrinsic motility defects, these amoeboid cells being on the contrary usually faster in the absence of branched actin (Vargas et al. and Leithner A. Nature Cell Biology 2016), although they sometime exhibit altered trajectories when evolving in complex environments. We therefore decided to respond to this referee's request by stressing that "Although WASp deficiency was not found to decrease DC motility in confinement *ex vivo* (Oliveira et al. Leukocyte Biology, 2022), we cannot exclude that *in vivo* it could contribute to DC migration to lymph nodes by other means than activating cPLA₂" (page 10).

Figure 1: Arpin WT and KO DCs migrate with comparable velocities *ex vivo*

A: Migration speed of immature DCs from Arpin WT and KO in channels with constrictions of 1 μm before and after crossing these constrictions. N=3, representative of one experiment n=16 cells in WT before, n= 17 cells in WT after, n=6 cells in KO before and after.

B. Migration speed of immature DCs from Arpin WT and KO in channels with constrictions of 2 μm before and after crossing these constrictions. N=3, representative of one experiment n=16 cells in WT before, n= 17 cells in WT after, n=5 cells in KO before and after.

C. Migration speed of LPS-activated DCs from Arpin WT and KO in collagen gels with chemokine gradient of CCL21. The speed is quantified at the zone close and far from the gradient. N=3, n=200 cells in WT far from gradient, n= 200 cells in WT close to the gradient, n= 288 cells in KO far from gradient, n= 265 cells in KO close to gradient.

4. The authors show convincingly that the transcriptional profile of mechanically stimulated DCs differs from that of DCs stimulated with LPS and that these cells are less efficient in activating T cells. Based on these data, they hypothesize that mechanical stimulation is responsible for immunoregulatory properties of tissue-patrolling DCs in the steady state which allows them to transport self-antigen to the lymph node in a tolerogenic state. Unfortunately, no functional *in vivo* data are shown to support this innovative hypothesis.

Authors: We thank the referee for bringing this point. We had considered that showing that cPLA₂ is needed for activation of the I κ k β /NF κ B pathway, which is known to drive cDC2 migration to LNs at steady-state and maintain peripheral tolerance (Baratin et al. Immunity 2015), was enough to reach our conclusion on shape sensing allowing antigen transport by DCs to lymph nodes in a tolerogenic state. Consistent with this previous finding, the new RNAseq analysis performed on CK666-treated DCs in response to referee's request #2 allowed us extracting a list of 467 genes that were highly upregulated in confined DCs and depended on both Arp2/3 and cPLA₂. Strikingly, pathway analysis of

these genes revealed that signaling of IL-10, a well-described tolerogenic cytokine, is the predominant pathway induced by shape sensing (Supplementary figS.6, results page 13, discussion page 15-second paragraph).

Nonetheless we agree with the referee that these results, although already quite compelling, are indirect evidence for shape-sensing protecting from autoimmunity. Therefore, we initiated an *in vivo* study where the susceptibility of cPLA₂ WT and KO mice to develop Experimental Autoimmune Encephalomyelitis (EAE) in response to immunization with Myelin Oligodendrocyte Glycoprotein (MOG) was compared. The results obtained in a first experiment displayed below show that mice whose CD11c⁺ compartment is KO for cPLA₂ develop a more severe phenotype than their wild-type counterpart upon MOG immunization, confirming the increased susceptibility of these animals to autoimmunity and supporting the role of cPLA₂ in making DCs tolerogenic. However, we have decided not to include them in the revised article for two reasons: 1) we could not do enough repeats of the experiment in a reasonable time for the revision of the article as obtaining animals with the right phenotypes is really tricky due to germline deleted mice that need to be excluded, and 2) this experiment calls for an entire study of the *in vivo* characterization of cPLA₂ KO mice in various immune cell subsets and tissues, which will take several years to complete and is out of the scope of this article. Therefore, to take into account this referee's concern, we have rephrased our conclusion to explicitly state that the evidence is so far indirect and to clarify the limitation of our study and the need for further *in vivo* study of cPLA₂ KO phenotypes (page 15-second paragraph).

Figure 2: cPLA₂ KO mice are more susceptible to autoimmune disease (EAE model).

Experimental autoimmune encephalomyelitis (EAE) was induced in both cPLA₂ WT (f/f CD11c cre-) and cPLA₂ KO (f/f CD11c cre+) male littermates of 8 weeks. The mice were observed over 2 weeks and scored for the severity of the clinical development of the disease (score from 1 to 4 with 4 having severe phenotype). N = 1 n = 4 WT and 5 KO mice. **A.** EAE median score for cPLA₂ WT and KO mice over the period of the study. **B.** EAE cumulative score of each mouse over the period of the study.

Minor points:

1. The intravital microscopic images are of poor quality and do not allow any conclusions to be drawn about the morphology of the DCs and their nuclei. What is the physiological range of nucleus confinement *in vivo*?

Authors: It is technically very challenging to get a better resolution when assessing DC migration *in vivo* due to the limitations of two-photon imaging. Nonetheless, to address this referee's concern, we now provide new movies and quantifications, as described above. These results show that dermal DCs

spend approximately a third of their time with their cell body constricted to minimal diameters ranging between 2 and 4 μm (fig. 1a, page 4), further validating the confinement heights used in our experiments. These results are very much in agreement with previous quantifications we made analyzing the minimal diameter of the nucleus of DCs patrolling the mouse ear skin (Raab et al. Science 2016). They are also in lines with several reports made by Michael Sixt's group (Renkawitz et al. Nature 2019). They are presented in figure 1a.

2. The cell morphology of the representative images is very heterogeneous. For example, the WASp-KO cell in Figure 3e is very small and round and shows no protrusions typical of DCs. Are these cells still alive.

Authors: The recent publication mentioned above (Major Point 3, Oliveira et al. Journal of Leukocyte Biology 2022, <https://doi.org/10.1002/JLB.1AB0821-013RR>) shows indeed that WASp KO DCs have a round morphology when evolving in confined environments. We now provide a movie of live DCs from WASp WT and KO under confinement of 3 μm height highlighting that WASp KO DCs maintain a round morphology upon confinement as compared to their WT counterpart (results page 8, supplementary movie S5), but are still actively migrating, showing that they are alive.

3. Please provide more than 2 biological replicates. E.g. in Fig. 1g; 3f,h

Authors: All missing replicates have been added to the corresponding figures.

Decision Letter, first revision:

7th Feb 2024

Dear Dr Lennon-Dumenil,

Your Article, "A Shape Sensing Mechanism driven by Arp2/3 and cPLA2 licenses Dendritic Cells for Migration to Lymph Nodes in Homeostasis" has now been seen by 4 referees.

You might recall that during the appeal process we noted that we would need new reviewers to cover the DC biology if we were to consider your appeal. So reviewer 1 and 2 here are the original reviewers and reviewer 3 and 4 are new. As you will see, all the reviewers are very positive about the paper.

We are very interested in the possibility of publishing your study in Nature Immunology, but would like to consider your response to these concerns in the form of a revised manuscript before we make a final decision on publication.

We therefore invite you to revise your manuscript taking into account all reviewer and editor comments. Please highlight all changes in the manuscript text file in Microsoft Word format.

- * Include a "Response to referees" document detailing, point-by-point, how you addressed each referee comment. If no action was taken to address a point, you must provide a compelling argument. This response will be sent back to the referees along with the revised manuscript.
- * If you have not done so already please begin to revise your manuscript so that it conforms to our Article format instructions at <http://www.nature.com/ni/authors/index.html>. Refer also to any guidelines provided in this letter.
- * Please include a revised version of any required reporting checklist. It will be available to referees to aid in their evaluation of the manuscript goes back for peer review. They are available here:

Reporting summary:

When submitting the revised version of your manuscript, please pay close attention to our [href="https://www.nature.com/nature-portfolio/editorial-policies/image-integrity">Digital Image Integrity Guidelines](https://www.nature.com/nature-portfolio/editorial-policies/image-integrity). and to the following points below:

-- that unprocessed scans are clearly labelled and match the gels and western blots presented in figures.

- that control panels for gels and western blots are appropriately described as loading on sample processing controls
- all images in the paper are checked for duplication of panels and for splicing of gel lanes.

[REDACTED]

We hope to receive your revised manuscript within two weeks. If you cannot send it within this time, please let us know. We will be happy to consider your revision so long as nothing similar has been accepted for publication at Nature Immunology or published elsewhere.

Nature Immunology is committed to improving transparency in authorship. As part of our efforts in this direction, we are now requesting that all authors identified as 'corresponding author' on published papers create and link their Open Researcher and Contributor Identifier (ORCID) with their account on the Manuscript Tracking System (MTS), prior to acceptance. ORCID helps the scientific community achieve unambiguous attribution of all scholarly contributions. You can create and link your ORCID from the home page of the MTS by clicking on 'Modify my Springer Nature account'. For more information please visit please visit www.springernature.com/orcid.

Sincerely,

Nick Bernard, PhD
Senior Editor
Nature Immunology

Reviewers' Comments:

Reviewer #1:

Remarks to the Author:

The authors have appropriately answered all my requests. This is a inspiring paper that breaks new

ground on the interphase of immunology and mechanobiology. It should have broad impact.

Reviewer #2:

Remarks to the Author:

The revised version of the manuscript has addressed all my criticisms and contains substantial new data that further support the authors' hypothesis. They present a very innovative concept that provides a comprehensive explanation of how the physical environment in tissues can influence the homeostatic behavior and function of immune cells. This multidisciplinary work has far-reaching implications for the fields of cell biology, biophysics and immunology and is therefore of great interest to the broad readership of Nature Immunology. I highly recommend publication.

Reviewer #3:

Remarks to the Author:

Overall this is a novel and interesting study, which would be of broad interest to the readership of the journal. I have several comments but all of which are minor and should be easily addressed without further experiments.

The data showing CCR7 regulation in DCs through confinement in vitro are convincing and robust, using a combination of genetic reporters and immunofluorescence staining of endogenous protein levels. Levels of CCR7 are functionally linked to migration speed and directional migration in response to soluble CCL19. However to link to migration within tissues, migration in response to CCL21 would be more relevant. cPLA2 is linked to CCR7 upregulation, specifically under confinement conditions, by knock down and knock out experiments. Nuclear localization of cPLA2 is linked to ARP2/3 activity using pharmacological perturbations and WASp and Arpin KO studies, but the in vivo validations (fig 5) are less convincing of a clear role for ARP2/3 as a key regulator of cPLA2 activity. The transcriptional data however very convincingly link cPLA2 activity to CCR7 upregulation along with a very interesting set of other immunoregulatory genes, which are shown to be distinct from gene expression induced in DCs by LPS. Finally the authors show that PGE2 production downstream of cPLA2 activation is a major mediator for CCR7 upregulation.

Minor points

Figure 1 (results section lines 148-150) I feel this is slightly over stated. What is shown clearly is the upregulation of CCR7 in response to confinement, but this is not directly linked to cell morphology/shape in the experiments shown. Further, confinement could alter motility via CCR7 independent mechanisms. For this last sentence to be more than speculative (lines 148-150), the authors would need to inhibit or knockdown CCR7 and show no change in migration rates at 2,3,4 micron confinement height.

In vivo steady state migration assays to draining LNs show only moderate changes in DC numbers, is this small change consistent with the hypothesis the WASp-Arpin dependent cPLA2 activation really 'licences' DCs for migration in the absence of inflammation? These data suggest a more subtle tuning by WASp or Arpin – perhaps the wording needs editing to reflect this. Further, since these pathways are likely to contribute to DC migration directly through their cytoskeletal functions for protrusion

formation, these subtle changes are hard to interpret. The additional text in lines 298/299 do not really address this satisfactorily. The data in 5b where cPLA2 is directly targeted is more convincing. The explanation of fig 5 data should also include description of the scale of the changes along with their statistical significance. Figure 5 would also be better presented if example/representative flow cytometry plots were presented alongside the cell number quantification. The DC numbers in the skin could be removed to a supplementary figure.

DC cultures with GM-CSF and now understood to generate a mixture of macrophages and cells like cDCs (as the authors state in their methods) but that a better method for generating DCs from bone marrow is now considered to be the use of Flt3L, which generates cDCs more similar to their in vivo counterparts. Can the authors comment on why GM-CSF cultures were used and in preference to Flt3L, and whether these in vitro derived cells are similar enough to the CD11c+ cells they are compared to in vivo?

Can the authors comment on how CCR7 is regulated differently by PPR activation versus confinement? This is mentioned as a control but I think this is a very interesting point.

Figure 1a labels – CD11C should be CD11c

Throughout Dapi should be DAPI

I found the flow diagrams in 7d and 7g non-intuitive to read, are there alternative presentations of these data?

Reviewer #4:

Remarks to the Author:

The manuscript by Alraies et al. provides novel evidence for a mechanism by which dendritic cells in the periphery can upregulate CCR7 and other migratory functions in the absence of inflammatory stimuli based on sensing of mechanical stress. The authors show in a series of convincing experiments that when dendritic cells move through a confinement of 3 μm , the nuclear envelope undergoes tension forces that trigger cPLA2 translocation to the nucleus in a manner dependent on Arp2/3. Consequently, NF κ B signaling is activated and CCR7 is upregulated. This study is well-performed and thoroughly investigates the various elements of the pathway to describe a cohesive story. The figures in the manuscript are put together nicely and the manuscript itself is clearly written. A few additional details would benefit the study, but the manuscript is overall strong and of high quality.

Major Comments:

1. The authors describe in the study that DCs subjected to confinement show some hallmarks of canonical maturation but not others. In particular, CCR7 is upregulated, while MHC-II and co-stimulatory markers are not. Antigen presentation to CD4 T cells is dampened compared to stimulation with LPS. It would be important to characterize other ways in which confinement-induced "maturation" differs from traditional activation with microbial compounds like LPS. For example, upon confinement, is the lifespan of DCs similar to that of DCs which have been treated with LPS? Or does the lack of inflammatory stimuli allow the DCs to survive in lymph nodes for a longer period of time? In addition to MHC-II, is MHC-I expression affected?

2. Previous work on cPLA2 nuclear translocation describes that this pathway is dependent on intracellular calcium signaling (doi: 10.1016/j.cell.2016.04.016). It would benefit the study to include some basic experiments to confirm that calcium is released from intracellular stores during 3 μ m confinement, and potentially that this pathway is inhibited when calcium is chelated. Additionally, the mechanism by which PGE2 upregulates CCR7 remains unclear. Presumably this is via NF κ B activation, but it would be good to confirm.

3. The finding that shape change induces DC activation of a unique type is a powerful finding. More work is needed in future papers to dissect out how exactly this affects T cells. The few experiments evaluating T cells and molecules implicated though RNASeq are superficial and non-additive to the paper. Using peptide for OTII proliferation is OK but does not adequately evaluate the DC function as an APC (Fig 7). The authors propose that the mechano-sensing pathway dependent on cPLA2 and Arp2/3 facilitates tolerogenic DC responses to self-antigens and thus the development of peripheral tolerance. Among the evidence for this are RNAseq data showing enrichment of IL-10 signaling in confined DCs, as well as Luminex analysis of secreted cytokines. DCs activated by PAMPs can make IL-10, so this alone is not a clear indicator of "tolerogenic" capacity. Further, it is inappropriate to focus on "differences" in co-stimulatory molecules between DCs activated by LPS vs compression if they are not statistically significant by flow cytometry (Fig S5d vs line 386). I do not think additional experiments are needed, but the conclusions about tolerance should be toned down, including the last sentence of the abstract – homeostatic migration is sufficient. This is not a paper about tolerance, but rather the cell biology of DCs.

Minor points:

- A model figure would enhance the clarity of this pathway for a broad audience.
- The sentence on page 11 beginning on line 352 is confusing: "Our results show that neither Arp2/3 nor cPLA2 were required for CCR7 upregulation in response to confinement..." Shouldn't it be that they are required?
- There is a typo in the last sentence of the abstract "an"
- If MHCII is used to gate DCs (fig. 5D) how are DCs from the KOs (presumably with different levels of MHCII) identified?

Author Rebuttal, first revision:

Reviewers' Comments:

Dear referees,

Thank you for your enthusiasm on our revised manuscript. We have addressed all your comments as described below, they significantly improved its quality.

All changes are shown in yellow in the revised article.

Reviewer #1:

Remarks to the Author:

The authors have appropriately answered all my requests. This is an inspiring paper that breaks new ground on the interphase of immunology and mechanobiology. It should have broad impact.

We thank the referee for his/her very positive assessment of our manuscript.

Reviewer #2:

Remarks to the Author:

The revised version of the manuscript has addressed all my criticisms and contains substantial new data that further support the authors' hypothesis. They present a very innovative concept that provides a comprehensive explanation of how the physical environment in tissues can influence the homeostatic behavior and function of immune cells. This multidisciplinary work has far-reaching implications for the fields of cell biology, biophysics and immunology and is therefore of great interest to the broad readership of Nature Immunology. I highly recommend publication.

Our response: We thank the referee for his/her very positive assessment of our manuscript.

Reviewer #3:

Remarks to the Author:

Overall this is a novel and interesting study, which would be of broad interest to the readership of the journal. I have several comments but all of which are minor and should be easily addressed without further experiments.

Our response: We thank the referee for his/her global positive assessment of our manuscript.

The data showing CCR7 regulation in DCs through confinement in vitro are convincing and robust, using a combination of genetic reporters and immunofluorescence staining of endogenous protein levels. Levels of CCR7 are functionally linked to migration speed and directional migration in response to soluble CCL19. However to link to migration within tissues, migration in response to CCL21 would be more relevant. cPLA2 is linked to CCR7 upregulation, specifically under confinement conditions, by knock down and knock out experiments.

Our response: we fully agreed with the referee, and we had initially tried to do this experiment with both CCL21 and CCL19. For a reason that we do not understand, only CCL19 formed a gradient successfully within our confinement device. As this experiment was meant to show that the CCR7 induced by confinement was on the cell surface and functional to guide dendritic cells, we considered that using CCL19 was enough for us to make this point.

Nuclear localization of cPLA2 is linked to ARP2/3 activity using pharmacological perturbations and WASp and Arpin KO studies, but the in vivo validations (fig 5) are less convincing of a clear role for ARP2/3 as a key regulator of cPLA2 activity.

Our response: We have tune down our conclusion on the role of Arp2/3 and WASp, suggesting a “tuning” role rather than a “licensing” one all along the manuscript.

The transcriptional data however very convincingly link cPLA2 activity to CCR7 upregulation along with a very interesting set of other immunoregulatory genes, which are shown to be distinct from gene expression induced in DCs by LPS. Finally the authors show that PGE2 production downstream of cPLA2 activation is a major mediator for CCR7 upregulation.

Our response: We thank the referee for his/her positive assessment of these important aspects of our work.

Minor points

Figure 1 (results section lines 148-150) I feel this is slightly over stated. What is shown clearly is the upregulation of CCR7 in response to confinement, but this is not directly linked to cell morphology/shape in the experiments shown. Further, confinement could alter motility via CCR7 independent mechanisms. For this last sentence to be more than speculative (lines 148-150), the authors would need to inhibit or knockdown CCR7 and show no change in migration rates at 2,3,4 micron confinement height.

Our response: We have modified the text according to the referee’s request.

In vivo steady state migration assays to draining LNs show only moderate changes in DC numbers, is this small change consistent with the hypothesis the WASp-Arp2/3 dependent cPLA2 activation really ‘licences’ DCs for migration in the absence of inflammation? These data suggest a more subtle tuning by WASp or Arpin – perhaps the wording needs editing to reflect this. Further, since these pathways are likely to contribute to DC migration directly through their cytoskeletal functions for protrusion formation, these subtle changes are hard to interpret. The additional text in lines 298/299 do not really address this satisfactorily. The data in 5b where cPLA2 is directly targeted is more convincing. The explanation of fig 5 data should also include description of the scale of the changes along with their statistical significance. Figure 5 would also be better presented if example/representative flow cytometry plots were presented alongside the cell number quantification. The DC numbers in the skin could be removed to a supplementary figure.

Our response: we have reorganized the figure as requested by the referee and tuned down our conclusion on the role of Arp2/3 and WASp being a “tuning” role rather than a “licensing” one all along the text (yellow highlights).

DC cultures with GMCSF and now understood to generate a mixture of macrophages and cells like cDCs (as the authors state in their methods) but that a better method for generating DCs from bone marrow is now considered to be the use of Flt3L, which generates cDCs more similar to their in vivo counterparts. Can the authors comment on why GMCSF cultures were used and in preference to Flt3L, and whether these in vitro derived cells are similar enough to the CD11c+ cells they are compared to in vivo?

Our response: we used a protocol developed by the team of Sebastian Amigorena in collaboration with Paola Ricciardi-Castagnoli (They et al. JCB 1999; Winzler et al. JEM 1997) to generate DCs from mouse bone-marrow. This protocol uses a supernatant of J558 cells transfected with GMCSF. It is different from the one that uses recombinant GMCSF, which as stated by the referee, indeed leads to very heterogenous cultures in terms of CD11c expression and maturation stage (as described in Helft et al. Immunity 2015). Our DC differentiation protocol leads to a DC population where >90% of the cells express CD11c after 10 days of culture and <15% of DCs are mature. This is now explained in the materials and methods (page 17).

Can the authors comments on how CCR7 is regulated differently by PPR activation versus confinement? This is mentioned as a control but I think this is very interesting point.

Our response: We followed the referee's suggestion by adding the following sentence to the first paragraph discussion : "Interestingly, while this mechanical pathway shares signaling players such as NFkB with the pathway induced by microbial sensing, it does not involve the direct engagement of DC receptors such as TLRs and NLRs and leads to an overlapping but distinct transcriptional program in these cells" (page 14).

Figure 1a labels – CD11C should be CD11c

Our response: This has been corrected.

Throughout Dapi should be DAPI

Our response: This has been corrected.

I found the flow diagrams in 7d and 7g non-intuitive to read, are there alternative presentations of these data?

Our response: We understand this point, however, we discussed it with our bioinformatics colleagues, who believe that this is the best representation for the case where four distinct samples must be compared. If the referee has something else in mind, we are fully open to his suggestions.

Reviewer #4:

Remarks to the Author:

The manuscript by Alraies et al. provides novel evidence for a mechanism by which dendritic cells in the

periphery can upregulate CCR7 and other migratory functions in the absence of inflammatory stimuli based on sensing of mechanical stress. The authors show in a series of convincing experiments that when dendritic cells move through a confinement of 3 μm , the nuclear envelope undergoes tension forces that trigger cPLA2 translocation to the nucleus in a manner dependent on Arp2/3. Consequently, NF κ B signaling is activated and CCR7 is upregulated. This study is well-performed and thoroughly investigates the various elements of the pathway to describe a cohesive story. The figures in the manuscript are put together nicely and the manuscript itself is clearly written. A few additional details would benefit the study, but the manuscript is overall strong and of high quality.

Our response: We thank the referee for his/her global positive assessment of our manuscript and for his/her suggestions to improve it.

Major Comments:

1. The authors describe in the study that DCs subjected to confinement show some hallmarks of canonical maturation but not others. In particular, CCR7 is upregulated, while MHC-II and co-stimulatory markers are not. Antigen presentation to CD4 T cells is dampened compared to stimulation with LPS. It would be important to characterize other ways in which confinement-induced “maturation” differs from traditional activation with microbial compounds like LPS. For example, upon confinement, is the lifespan of DCs similar to that of DCs which have been treated with LPS? Or does the lack of inflammatory stimuli allow the DCs to survive in lymph nodes for a longer period of time? In addition to MHC-II, is MHC-I expression affected?

Our response: These are very interesting questions. Regarding the cell lifespan, we found that it was similar in bone marrow-derived DCs confined at 3 \$\mu\text{m}\$ -height or treated with LPS (fig. S5f), this is commented in the text (page 13). Regarding the expression of MHC-I genes, we observed following the referee’s suggestion that they are upregulated upon confinement of DCs at 3 \$\mu\text{m}\$ -height but to a lesser extent than with LPS. We have added these results to fig. S5c (page 12). Regarding the *in vivo* experiments, we are waiting for cPLA₂ knock out mice to assess these differences in lymph node DCs using flow cytometry and transcriptomics analyses, which should be part of a follow up study.

2. Previous work on cPLA2 nuclear translocation describes that this pathway is dependent on intracellular calcium signaling (doi: 10.1016/j.cell.2016.04.016). It would benefit the study to include some basic experiments to confirm that calcium is released from intracellular stores during 3 μm confinement, and potentially that this pathway is inhibited when calcium is chelated.

Our response: We have shown in our previous manuscript (Lomakin et al. 2020) that calcium is indeed required to increase actomyosin contractility upon nuclear stretching and cPLA₂ activation in HeLa cells, suggesting that it might be required for DCs as well. However, our past experience working on calcium

dynamics in DCs is that using calcium probes is not impossible but very difficult as these cells tend to rapidly get rid of the probes, for reasons we do not understand. We have thus decided to address this question in DCs using Salsa mice, which express an endogenous calcium sensor, and which we have recently transferred to our animal facility. Indeed, we have interesting preliminary data suggesting that lysosomal calcium channel TRPML1 might be involved in the response of DCs to confinement. This is the starting point for the project of a recently recruited PhD student. We therefore chose not to discuss calcium involvement in the present manuscript.

Additionally, the mechanism by which PGE₂ upregulates CCR7 remains unclear. Presumably this is via NFκB activation, but it would be good to confirm.

Our response: We have followed the referee's suggestion by investigating the role of NFκB in upregulation of CCR7 expression by PGE₂ in cPLA₂ KO confined DCs. As hypothesized by the referee, we found that IKKβ inhibition impairs upregulation of CCR7 expression by PGE₂ (see new fig.7c). Accordingly, we also observed that addition of PGE₂ leads to NFκB nuclear translocation in DCs in an IKKβ-dependent manner (see new fig. 7c).

3. The finding that shape change induces DC activation of a unique type is a powerful finding. More work is needed in future papers to dissect out how exactly this affects T cells. The few experiments evaluating T cells and molecules implicated though RNASeq are superficial and non-additive to the paper. Using peptide for OTII proliferation is OK but does not adequately evaluate the DC function as an APC (Fig 7). The authors propose that the mechano-sensing pathway dependent on cPLA₂ and Arp2/3 facilitates tolerogenic DC responses to self-antigens and thus the development of peripheral tolerance. Among the evidence for this are RNAseq data showing enrichment of IL-10 signaling in confined DCs, as well as Luminex analysis of secreted cytokines. DCs activated by PAMPs can make IL-10, so this alone is not a clear indicator of "tolerogenic" capacity. Further, it is inappropriate to focus on "differences" in co-stimulatory molecules between DCs activated by LPS vs compression if they are not statistically significant by flow cytometry (Fig S5d vs line 386). I do not think additional experiments are needed, but the conclusions about tolerance should be toned down, including the last sentence of the abstract – homeostatic migration is sufficient. This is not a paper about tolerance, but rather the cell biology of DCs

Our response: This referee's comment is totally valid and fair. Accordingly, we have now tempered our conclusions regarding the tolerogenic potential of these cells throughout the entire text.

Minor points:

- A model figure would enhance the clarity of this pathway for a broad audience.

Our response: Thank you for this very good suggestion, this model figure is now provided (fig. S7).

- The sentence on page 11 beginning on line 352 is confusing: “Our results show that neither Arp2/3 nor cPLA2 were required for CCR7 upregulation in response to confinement...” Shouldn’t it be that they are required?

Our response: Sorry for this mistake, this has been rephrased properly (page 11).

- There is a typo in the last sentence of the abstract “an”

Our response: This has been fixed.

- If MHCII is used to gate DCs (fig. 5D) how are DCs from the KOs (presumably with different levels of MHCII) identified?

Our response: We did not observe any significant difference in MHC class II expression at the surface of lymph node DCs expressing or not cPLA₂ or WASp. There are several potential non-exclusive explanations for this result. It could be related to the kinetics of MHC-II upregulation: all *ex vivo* experiments were performed 4h upon DC treatment (confinement or LPS), which might be short to reach the fully mature MHC-II^{high} phenotype of lymph node DCs. It is also possible that the DCs that manage to migrate to lymph nodes, despite these genes being deleted, are equipped with some compensatory mechanism(s) and/or are responding to a distinct endogenous environmental stimulus than physical confinement. We are currently digging into these questions by carefully comparing the transcriptomic profiles of lymph node migratory DCs in WT and KO animals, which should be part of a follow-up study.

Decision Letter, second revision:

25th Mar 2024

Dear Dr. Lennon-Duménil,

Thank you for submitting your revised manuscript "Cell Shape Sensing Licenses Dendritic Cells for Homeostatic Migration to Lymph Nodes" (NI-A35561B). It has now been seen by the original referees and their comments are below. The reviewers find that the paper has improved in revision, and therefore we'll be happy in principle to publish it in Nature Immunology, pending minor revisions to comply with our editorial and formatting guidelines.

We will now perform detailed checks on your paper and will send you a checklist detailing our editorial and formatting requirements in about a week. Please do not upload the final materials and make any revisions until you receive this additional information from us.

If you had not uploaded a Word file for the current version of the manuscript, we will need one before beginning the editing process; please email that to immunology@us.nature.com at your earliest convenience.

Thank you again for your interest in Nature Immunology Please do not hesitate to contact me if you have any questions.

Sincerely,

Nick Bernard, PhD
Senior Editor
Nature Immunology

Reviewer #4 (Remarks to the Author):

The authors have adequately addressed my concerns with the most recent revisions, in particular the issues on statements regarding tolerance. The additional data in Fig 7 strengthens the overall story. This is a well-designed and executed study that will enhance our understanding of dendritic cell biology.

Author rebuttal, second revision:

Reviewers' Comments:

Dear referees,

Thank you for your enthusiasm on our revised manuscript. We have addressed all your comments as described below, they significantly improved its quality.

Reviewer #4:

Remarks to the Author:

The authors have adequately addressed my concerns with the most recent revisions, in particular the issues on statements regarding tolerance. The additional data in Fig 7 strengthens the overall story. This is a well-designed and executed study that will enhance our understanding of dendritic cell biology.

We thank the referee for her/his positive assessment of our revised manuscript, whose quality had indeed been improved by her/his suggestions.

Final Decision Letter:

Dear Dr. Lennon Duménil,

I am delighted to accept your manuscript entitled "Cell shape sensing licenses dendritic cells for homeostatic migration to lymph nodes" for publication in an upcoming issue of Nature Immunology.

Over the next few weeks, your paper will be copyedited to ensure that it conforms to Nature Immunology style. Once your paper is typeset, you will receive an email with a link to choose the appropriate publishing options for your paper and our Author Services team will be in touch regarding any additional information that may be required.

Please note that *Nature Immunology* is a Transformative Journal (TJ). Authors may publish their research with us through the traditional subscription access route or make their paper immediately open access through payment of an article-processing charge (APC). Authors will not be required to make a final decision about access to their article until it has been accepted. Find out more about Transformative Journals.

Your paper will be published online soon after we receive your corrections and will appear in print in the next available issue.

You may wish to make your media relations office aware of your accepted publication, in case they

consider it appropriate to organize some internal or external publicity. Once your paper has been scheduled you will receive an email confirming the publication details. This is normally 3-4 working days in advance of publication. If you need additional notice of the date and time of publication, please let the production team know when you receive the proof of your article to ensure there is sufficient time to coordinate. Further information on our embargo policies can be found here: <https://www.nature.com/authors/policies/embargo.html>

Also, if you have any spectacular or outstanding figures or graphics associated with your manuscript - though not necessarily included with your submission - we'd be delighted to consider them as candidates for our cover. Simply send an electronic version (accompanied by a hard copy) to us with a possible cover caption enclosed.

If you have not already done so, we strongly recommend that you upload the step-by-step protocols used in this manuscript to the Protocol Exchange. Protocol Exchange is an open online resource that allows researchers to share their detailed experimental know-how. All uploaded protocols are made freely available, assigned DOIs for ease of citation and fully searchable through nature.com. Protocols can be linked to any publications in which they are used and will be linked to from your article. You can also establish a dedicated page to collect all your lab Protocols. By uploading your Protocols to Protocol Exchange, you are enabling researchers to more readily reproduce or adapt the methodology you use, as well as increasing the visibility of your protocols and papers. Upload your Protocols at www.nature.com/protocolexchange/. Further information can be found at www.nature.com/protocolexchange/about .

Please note that we encourage the authors to self-archive their manuscript (the accepted version before copy editing) in their institutional repository, and in their funders' archives, six months after publication. Nature Portfolio recognizes the efforts of funding bodies to increase access of the research they fund, and strongly encourages authors to participate in such efforts. For information about our editorial policy, including license agreement and author copyright, please visit www.nature.com/ni/about/ed_policies/index.html

Sincerely,

Nick Bernard, PhD
Senior Editor
Nature Immunology